# Untangling the systematic dilemma behind the roughskin spurdog *Cirrhigaleus asper* (Merrett, 1973) (Chondrichthyes: Squaliformes), with phylogeny of Squalidae and a key to *Cirrhigaleus* species

**Sarah Viana** [1,2,3]*, **Karla D. A. Soares** [4]

**1** Department of Ichthyology and Fisheries Science, Rhodes University, Makhanda (Grahamstown), South Africa, **2** NRF–South African Institute for Aquatic Biodiversity, Makhanda (Grahamstown), South Africa, **3** Departamento de Zoologia, Instituto de Biociências, Universidade de São Paulo, São Paulo, Brazil, **4** Departamento de Zoologia, Instituto de Biologia, Universidade Federal do Rio de Janeiro, Rio de Janeiro, Brazil

* stviana@gmail.com

**Data Availability Statement:** All relevant data are within the paper and its Supporting Information files.

## Abstract

*Cirrhigaleus* comprises a small genus of rare barbel-bearing dogfish sharks with distributions in limited regions of all oceans. Generic validity and taxonomic status of some species are upon controversies by morphological and molecular evidence that often suggest reallocation of *Cirrhigaleus* species into the genus *Squalus*. Particularly, the roughskin spurdog *C. asper* exhibits intermediary morphological characteristics within Squalidae that requires clarification. In the present study, a phylogenetic approach was undertaken to test the correct generic placement of *C. asper* using novel and revised morphological characters. We performed maximum parsimony analysis of 51 morphological characters of the internal (e.g., neurocranium, clasper cartilages, pectoral and pelvic girdles) and external anatomy applied to 13 terminal taxa. *Cirrhigaleus* represents a valid genus and it is supported by eight synapomorphies: high number of monospondylous vertebrae; medial nasal lobe supported by fleshy core and innervated by the buccopharyngeal branch of the facial nerve; neurocranium with greatest width across nasal capsules; one facet and one condyle in the puboischiadic bar for articulating with the basipterygium; two intermediate segments between the basipterygium of the pelvic fin and the axial cartilage of the claspers; five terminal clasper cartilages; and posterior medial process of the puboischiadic bar absent. *Cirrhigaleus asper* is sister-species to a small clade comprising *C. barbifer* and *C. australis* which is supported by one synapomorphy, presence of conspicuous cusplets in the dermal denticles. *Cirrhigaleus barbifer*, *C. asper* and *C. australis* are redescribed herein and the neotype of *C. barbifer* is designated. A key to *Cirrhigaleus* species is also given and the inner relationships within *Squalus* is tentatively discussed.

**Funding:** This study was funded by the following: Fundação de Amparo à Pesquisa do Estado de São Paulo (FAPESP 2011/18861–7; 2013/11621–6; 2014/26503–1), STFLV, http://www.fapesp.br/en/; Conselho Nacional de Desenvolvimento Científico e Tecnológico (CNPq 158773/2011–0), STFLV, http://cnpq.br/; Geddes Collection Visiting Fellowship (2013), Australian Museum, STFLV. The funders had no role in study design, data collection and analysis, decision to publish, or preparation of the manuscript.

**Competing interests:** The authors have declared that no competing interests exist.

## Introduction

*Cirrhigaleus* Tanaka, 1912 [1] is a genus comprised by three deep-sea barbel-bearing dogfish sharks: the mandarin dogfish *Cirrhigaleus barbifer* Tanaka, 1912 [1] (type-species); the rough-skin spurdog *C. asper* (Merrett, 1973) [2]; and the Southern mandarin dogfish *C. australis* White, Last and Stevens, 2007 [3]. *Cirrhigaleus barbifer* is originally described from Japanese waters and exhibits records in the coasts of Indonesia, New Caledonia and Western Australia [1, 3–7]. *Cirrhigaleus asper* is recognized in the Western Indian and Atlantic Oceans, and in the Hawaiian Islands [2, 6, 8–11]. *Cirrhigaleus australis* is known from the South-western Pacific Ocean, including Australia (except Western Australia) and New Zealand [3, 10], and in the Indian Ocean [12].

Originally the genus *Cirrhigaleus* was described as monotypic and the main diagnostic characteristic attributed to its members was the presence of elongate moustache-like nasal barbels [1]. The distinctive nasal barbel (short or elongate) is an extension from the medial lobe of the anterior nasal flap of the nostrils, and it is believed to be the only efficient external diagnostic character of the genus. Due to morphological similarities between species of *Cirrhigaleus* and *Squalus* Linnaeus, 1758 [13], the validity of *Cirrhigaleus* as a separate genus within the family Squalidae as well as the correct generic allocation of particularly *C. asper* are upon incessant taxonomic discussions (e.g., [3, 4, 6, 8, 14, 15]). According to [16, 17], *Cirrhigaleus* and *Squalus* share similar characteristics regarding to the shape of dorsal-fin spines, spiracles, dermal denticles, dentition and shape of dorsal and caudal fins. Thus, these authors considered *Cirrhigaleus* as subgenus of *Squalus*. For [18, 19] the nasal barbels along with absence of upper and lower precaudal pits, and absence of precaudal keels in *C. barbifer* are sufficient for considering it as a valid and separate genus. Additional differences with *Squalus* regarding length of snout and head (shorter in *Cirrhigaleus*), length and height of dorsal fins (shorter and higher in *Cirrhigaleus*), size of dermal denticles (about twice in *Cirrhigaleus*), and teeth and vertebrae counts were noticed in [4] that supported its validity even though for these authors the lateral precaudal keel is also present in *Cirrhigaleus*.

*Cirrhigaleus asper* described from the Seychelles in the Western Indian Ocean is the only species within the genus that bears anterior margin of nostrils elongate but non moustache-like, a condition similar to those of species of *Squalus*. So, the species has been often misidentified with other *Squalus* species. Similarities of dentition, terminal cartilages of claspers, vertebral counts and morphometrics with *S. acanthias* Linnaeus, 1758 [13] and *S. blainvillei* (Risso, 1827) [20] were noticed in [2]. Thus, allocating the nominal species "*asper*" into the genus *Squalus* but without comparative analysis with *Cirrhigaleus*. The author further stated that *C. asper* has intermediate characteristics between these two species of *Squalus* (except for size of dermal denticles) when considering the definition of groups of species of *Squalus* proposed in [21]. Bass [8] was the pioneer on noticing morphological similarities between *S. asper* (= *Cirrhigaleus asper*) and *C. barbifer* such as the shape of the nasal barbels and height of dorsal fins. For these authors, *Squalus* comprises four group of species [8]: the *S. acanthias* group, the *S. mistukurii* group, the *S. megalops* group and the *S. asper* group. The latter group is represented by both *C. asper* and *C. barbifer* which are characterized by absence (or weakly evident) of upper precaudal pit, large dermal denticles, and head and snout small and blunt. Compagno (1984) [14] suggested the allocation of the nominal species "*asper*" within *Squalus* and maintaining *C. barbifer* in a separate genus but later on [6] recognized both nominal species as belonging to the genus *Cirrhigaleus* following the evidence provided in [22]. Difficulties on providing diagnostic characters for separating *Cirrhigaleus* species were also noticed in [3] when describing *C. australis*.

Cladistics analysis of internal and external morphological characters in [22] supported the monophyly of *Cirrhigaleus* comprising both *C. barbifer* and *C. asper* and as sister-group of

*Squalus*. Synapomorphies supporting the phylogenetic relationships between these genera included innervation of the nasal barbel, presence of supraethmoidal processes in the neurocranium and precaudal pit. Molecular systematics of Squalidae in [15] supported *Cirrhigaleus* (*C. australis* + *C. asper*) as monophyletic comprising a sister-group to a small clade representing the *S. acanthias* group (comprising the species *S. acanthias* and *S. suckleyi* (Girard, 1855) [23] and all other *Squalus* species forming another separate clade. These authors then suggested *Cirrhigaleus* as a potential junior synonym of *Squalus*. Controversies of morphological and molecular evidence have raised the debate regarding the taxonomic dilemma behind *C. asper* and the validity of *Cirrhigaleus*, requiring re-investigation. The present study thus aimed to elucidate the correct generic allocation of *C. asper* and discuss the phylogenetic relationships of *Cirrhigaleus* within Squalidae through a phylogenetic approach.

## Material and methods

### External morphology

Preserved specimens in 70% ethanol were examined for collecting external morphological data. Teeth samples from upper and lower jaws (three lateral teeth from the first series of teeth, right side) were taken and investigated using a stereoscopic microscope. Skin samples measuring 1x1 centimeters were taken from below the first dorsal fin (right side) and analysis of squamation was undertaken using a stereoscopic microscope and Scanning Electron Microscope (SEM) at the Instituto de Biociências, Universidade de São Paulo (IBUSP) and the Electron Microscope Unity, Rhodes University (RU). Terminology for external morphology and colour pattern follows [24]. Colouration is described from preserved specimens unless otherwise noted in text. Nomenclature for dentition is according to [25]. Terminology for dermal denticles is according to [26]. External morphology of the claspers is described following [27].

### Internal morphology

Dissections from preserved specimens in 70% ethanol were undertaken for analysis of the skeletal anatomy, particularly, neurocranium and associated nerves, pectoral and pelvic apparatus, and cartilages of the clasper. Fresh specimens were exposed to hot water at 60˚C for approximately 10min for separation of the flesh and skeleton. Later, skeletal structures were cleaned up using surgical instruments and preserved in 70% ethanol. Clearing and staining preparation were done for a few specimens, following [28, 29]. Micro-CT scan of the head of the holotype of *C. asper* was done at the Natural History Museum, London. Analyses of CT Scan datasets were undertaken using 3D Slicer software [30]. Terminology for neurocranium and cranial nerves follows [22], for pectoral fins and girdle as well as pelvic fins and girdle follow [22, 31, 32], and for cartilages of the claspers are according to [27]. Full descriptions of skeletal structures are provided under the section 'Taxonomic account'. Descriptions are first given for the type-species, *C. barbifer*, followed by comparisons with its congeners.

### Meristic data

Tooth counts follow [33]. Vertebral counts were obtained from radiographs (digital and printed film) and tooth row counts from preserved material. Vertebral counts follow [34].

### Morphometric data

Measurements were obtained using digital calipers with a 0.1 millimeter (mm) precision and/ or a metric tape (for measurements greater than 150 mm). External measurements were taken according to [24] as provided in [35] for *Squalus*, except for length of anterior nasal flap that

follows [3]. Cranial measurements were obtained from dissected specimens and follow [36, 37] with modifications as in [35]. Subethmoidal ridge length is defined here as the longitudinal distance between its anteriormost edge at the rostral carina and its posteriormost edge between the subnasal fenestrae. External and cranial measurements are expressed, respectively, as percentages of total length (%TL) and of the neurocranium total length (%CL). In tables, mean and standard deviation are given to include all specimens in which data were taken, except for *C. asper* whose range values of measurements taken from fresh specimens are given separately.

## Morphological phylogenetic analysis

Ingroup terminal taxa include the three species of *Cirrhigaleus. Dalatias licha* (Bonaterre, 1788) [38] and *Isistius brasiliensis* (Quoy and Gaimard, 1824) [39] (Dalatiidae) and species of *Squalus* (*S. acanthias*, *S. albifrons* Last, White and Stevens, 2007 [40], *S. brevirostris* Tanaka, 1917 [41], *S. japonicus* Ishikawa, 1908 [42], *S. megalops* (Macleay, 1881) [43], *S. mitsukurii* Jordan and Snyder, 1903 [44], *S. montalbani* Whitley, 1931 [45] and *S. suckleyi*) were analysed as outgroup taxa. *Dalatias licha* was chosen to root the cladogram based on the phylogenetic hypothesis of Squaliformes from [15]. Character polarity was made by outgroup comparison following [46]. Data for *D. licha* and *I. brasiliensis* were obtained from literature review if available (including [22, 32, 34, 47–52]) and/or examination of specimens (see comparative material). Character matrix was divided into two datasets: S1 Table, quantitative characters with absolute and normalized values for each terminal are presented; S2 Table, only qualitative characters are included.

Phylogenetic analysis was performed under parsimony using TNT 1.1 [53, 54] and the 'Traditional Search' option, using the TBR (Tree Bisection Reconnection) algorithm. Quantitative (1–4) and multistate qualitative (15, 30, 32 and 51) characters were analyzed as ordered. Missing entries were coded as '?' and used when appropriate study material was unavailable or character state was inapplicable for a given character. We used implied weighting herein and tested three different values of concavity constant, $k$ (1, 3 and 5). CI (consistency index) and RI (retention index) values and synapomorphies of nodes were obtained from the set of equally most-parsimonious trees. The relative degree of support for each node in the trees obtained with implied weights was assessed with branch support indices [55] and symmetric resampling [53]. Relative Bremer support was calculated using TBR and retaining suboptimal trees by seven steps. GC values (differences of frequencies "Group present/Contradicted" [53]) were calculated using the strict consensus and 2000 replicates. Tree edition was performed with the aid of FigTree v1.4.3 and Adobe Photoshop CS6.

The statement of each character and its states is followed by its CI and RI. A list of synapomorphies is given below and begins in Squalidae and progressively continues to less inclusive clades within the family. For each clade, only non-ambiguous synapomorphies are listed (see S1 File). After each synapomorphy, the number of the referred character and its state changes are shown in brackets. A complete list of character transformations is presented in S2 File. Complete data matrix including quantitative and qualitative characters analyzed are given in S3 File and S4 File.

## Taxonomic validation

Species recognized herein are redescribed under the section 'Taxonomic account'. Synonyms for species include authorship, date, pages and figures whenever possible. Diagnosis and key to species are based on adult specimens only (unless otherwise stated in text) due to ontogenetic and regional plasticity of phenotypic characteristics. In the descriptions, single values for morphometric and meristic data are for holotype whereas ranges represent values for all material in which data were taken (except when otherwise mentioned in text).

## Species illustrations

Specimens were photographed with a digital camera in different views and specific body structures (e.g., claspers). Photographs of teeth and dermal denticles of specimens were taken using a high-resolution digital camera attached to stereoscope microscope Leica DFC295. Skeletal structures and individual parts were illustrated using the ink on paper technique or schematic drawing using digital photography.

## Species distribution

QGIS 2.18 Las Palmas [56] and Google Earth 7.1.5.1557 were run to create maps of geographical distribution. Coordinates of each specimen examined were obtained from collecting event data available in the ichthyological databases. For specimens without accurate coordinates, the nearest locality data was considered for plotting maps.

## Abbreviations

Institutional abbreviations follow [57].

Anatomical abbreviations are as follows: abc: condyle for the anterior pelvic basal; abv: anterior pelvic basal; aoc: antorbital cartilage; ap: apopyle; ax: axial cartilage; bp: basipterygium; bp: basipterygium; bse: barrel-shaped elements; btp: basitrabecular process; buVII: buccopharyngeal branch of facial nerve; b1: first intermediate segment; b2: second intermediate segment; cbp: condyle for the basipterygium; cg: clasper groove; cnab: fleshy core; co: coracoid bar; df: diazonal foramen; ec: ethmoidal canal of *ophthalmicus superficialis* nerve; elf: endolymphatic foramen; ep: epiphysial pit; fopp: *profundus* canal for the *ophthalmicus profundus* nerve; fpb: facet for the basipterygium; fpr: facet for propterygium; fvn: foramen for ventral fin nerve; hp: hypopyle; lpp: lateral prepelvic process; lra: lateral rostral appendage; mes: mesopterygium; mnl: medial nasal lobe supported by cartilage; mp: mesial process; mra: medial rostral appendage; mrp: median rostral prominence; msc: mesocondyle; mtc: metacondyle; mtp: metapterygium; nab: nasal barbel; nc: nasal capsule; oc: otic capsule; pc: procondyle; pcf: pectoral fin; pcr: pectoral fin radials; pep: preorbital process; plf: perilymphatic foramen; plp: posterior-lateral process; poc: preorbital canal of superficial ophthalmic nerve; pop: postorbital process; pro: propterygium; ptp: posterior triangular process; pub: puboischiadic bar; p2: pelvic fin; r: rostrum; rd: dorsal marginal cartilage; rh: rhipidion; rl: pelvic radials; rk: rostral keel; rv: ventral marginal cartilage; scl: scapula; snf: subnasal fenestra; scp: scapular process; sec: subethmoid chamber; sep: supraethmoidal process; snf: subnasal fenestra; td: dorsal terminal cartilage; td2: dorsal terminal 2 cartilage; tv: ventral terminal cartilage; tv2: ventral terminal 2 cartilage; t3: accessory terminal 3 cartilage; β: beta cartilage.

## Comparative material

A list of examined preserved material is provided in S5 File and include species of the genera *Squalus*, *Dalatias* and *Isistius* that were used for comparative purposes. Preserved examined material of *Cirrhigaleus* species are given in full under the section 'Taxonomic account'. Skeletal material for character analysis and polarity is given for both ingroups and outgroups of the current analysis.

## Results

### Character description and polarity

Meristics (**S1 Table**).

1) Monospondylous vertebrae counts: minimum = 37, maximum = 53 (CI = 64; RI = 69).

Monospondylous vertebrae are defined as those vertebrae whose centrum corresponds to a single intermuscular septum and a pair of neural spines. Usually, these vertebrae are located between the occipital centrum in the neurocranium to over or posterior to the pelvic girdle [58]. This character has been used to separate taxonomically species of sharks as in [34] and not less obviously in Squalidae (e.g., [59]). In *Dalatias* and *Isistius*, total counts of monospondylous vertebrae vary from 37 (minimum) to 44 (maximum). In S*qualus* species, it varies from 37 (minimum) to 50 (maximum) whereas in *Cirrhigaleus* species it is 47–53.

2) Diplospondylous vertebrae counts: minimum = 35, maximum = 77 (CI = 100; RI = 100).

Diplospondylous vertebrae are defined as those vertebrae whose centrum corresponds to two subsequent intermuscular septa and a pair of neural spines. These vertebrae are located from the pelvic girdle to the caudal tip [58]. In *Dalatias* and *Isistius*, total counts of diplospondylous vertebrae vary from 35 (minimum) to 47 (maximum). In S*qualus* species, it varies from 60 (minimum) to 77 (maximum) whereas in *Cirrhigaleus* species the range from 60 to 69.

3) Upper teeth rows: minimum = 16, maximum = 39 (CI = 0; RI = 5).

4) Lower teeth rows: minimum = 15, maximum = 31 (CI = 0; RI = 3).

Total number of tooth rows in the upper and lower jaws are often expressed as a dental formula in taxonomic descriptions. These formulae vary widely within Squaliformes as seen in [60] and rarely are applied in phylogenetic analysis of sharks (e.g., [61, 62]). In *Dalatias* and *Isistius*, upper tooth rows range from 16 (minimum) to 39 (maximum) and lower tooth rows varies from 15 (minimum) to 31 (maximum). In *Squalus* species, the range is 21–28 for upper tooth and 17–25 for lower tooth rows. *Cirrhigaleus* species show a minimum of 23 and maximum of 30 upper tooth rows and 18–27 for lower tooth rows.

**Neurocranium and associated nerves.**

5) Innervation of the medial nasal lobe by the buccopharyngeal branch of the facial nerve (VII): character state 0 –absent; character state 1 –present (CI = 100; RI = 100).

In *Cirrhigaleus*, anterior margin of nostrils has a medial nasal lobe that bears fleshy core and it is innervated by the buccopharyngeal branch of facial nerve (VII) (5,1). The nerve buccopharyngeal branch of facial nerve (VII) arise from the foramen prooticum (V-VII) at the interorbital wall and runs aside the suborbital shelf of the neurocranium and over *suborbitalis* muscle in the palatoquadrate till it reaches the antorbital cartilage (Fig 1). Then, the branch buccopharyngeal runs ventrally in the subethmoidal region and reaches the nasal capsule in the neurocranium where it splits into two different small branches. The left branch is thicker than the right one and it runs to the origin of the nasal barbel (medial nasal lobe) at the fleshy core. The second branch runs straight to the rostral tip and aside the rostral keel. In *S. acanthias*, *S. suckleyi*, *S. megalops*, *S. brevirostris*, *S. albifrons*, *S. mitsukurii*, *S. montalbani* and *S. japonicus*, buccopharyngeal nerve runs exclusively to the tip of the rostrum with small innervations running to the nasal capsules, although these do not reach the anterior margin of nostrils. Thus, anterior margin of nostrils is not innervated in these species and the medial nasal lobe is internally supported by a thin nasal cartilage only, lacking fleshy core. In *Dalatias*, *Isistius* and species of *Squalus* such innervation is absent (5,0) [22]. Proposed this character as a synapomorphy of *Cirrhigaleus*.

6) Fleshy core at the anterior margin of nostrils: character state 0 –absent; character state 1 – present (CI = 100; RI = 100).

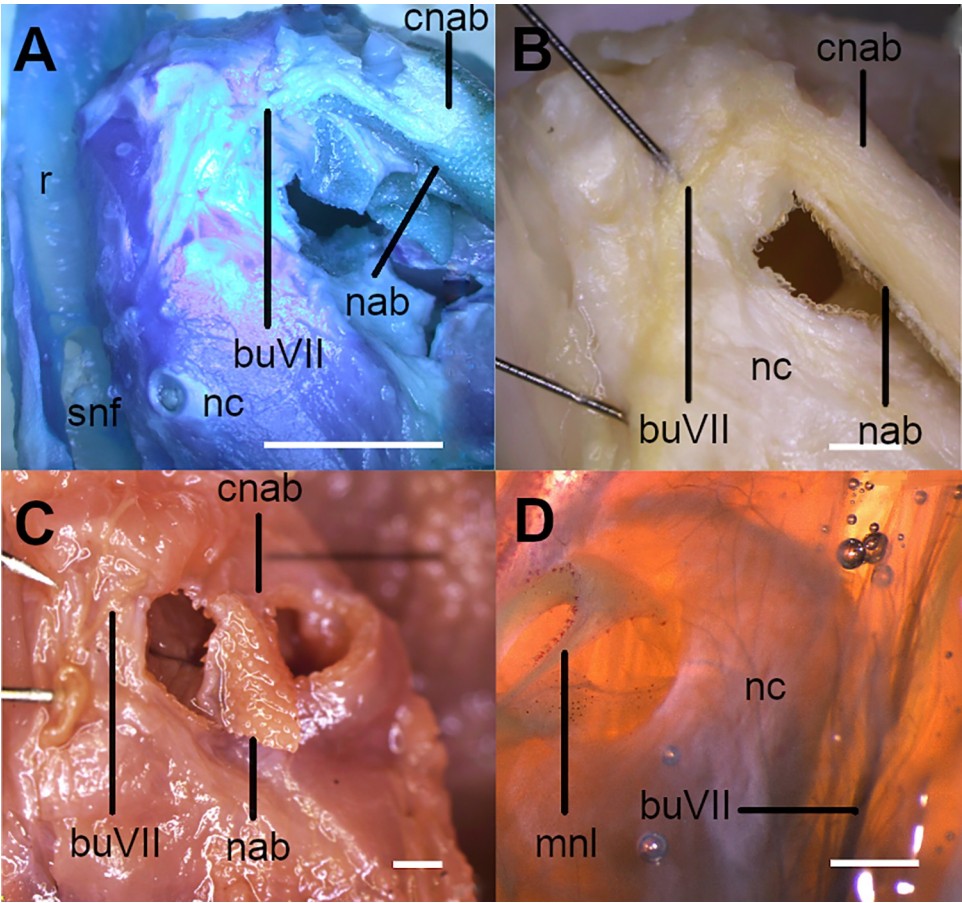

**Fig 1.** Anterior margin of nostrils in *Cirrhigaleus* (A–C, left side) and *Squalus* (D–E, right side), showing its innervation and internal support. A: *C. barbifer*, HUMZ 95177, juvenile female, 584 mm TL; B: *C. australis*, CSIRO H 7042–04, juvenile female, 605 mm TL; C: *C. asper*, SAIAB 6092, neonate female, 275 mm TL; D: *Squalus suckleyi*, CAS 21971 (cleared and stained), neonate female, 290 mm TL. Abbreviations: buVII: buccopharyngeal branch of facial nerve; cnab: fleshy core; mnl: medial nasal lobe supported by cartilage (in blue); nab: nasal barbel; nc: nasal capsule; r: rostrum; snf: subnasal fenestra. Scale bars: 5 mm (A); 1 mm (B–D).

Medial nasal lobe at the anterior margin of nostrils is supported by a fleshy core in *Cirrhigaleus* (6,1) (Fig 1). This condition is absent in *Dalatias*, *Isistius* and *Squalus* (6,0).

7) Supraethmoidal process in the neurocranium: character state 0 –absent; character state 1 – present (CI = 100; RI = 100).

Species of *Squalus* present a paired short and cylindrical process located at the dorsoposterior edge of the prefrontal fontanelle in the neurocranium (7,1) (Fig 2C–2E; Fig 3C–3F). These processes are absent in *Dalatias*, *Isistius* and in both *C. barbifer* and *C. australis* (7,0) (Fig 2A and 2B; Fig 3A and 3B). In *C. asper*, these processes are absent in the Indian Ocean population but present in the Western Atlantic population, and thus this terminal was coded as [1].

8) Rostrum shape: character state 0 –rostrum reduced and not trough-shaped; character state 1 –developed and trough-shaped (CI = 100; RI = 100).

Ethmoidal region of the neurocranium exhibits an anterior process, the rostral cartilage, which varies in shape and length. The rostrum may be developed and trough-shaped. This

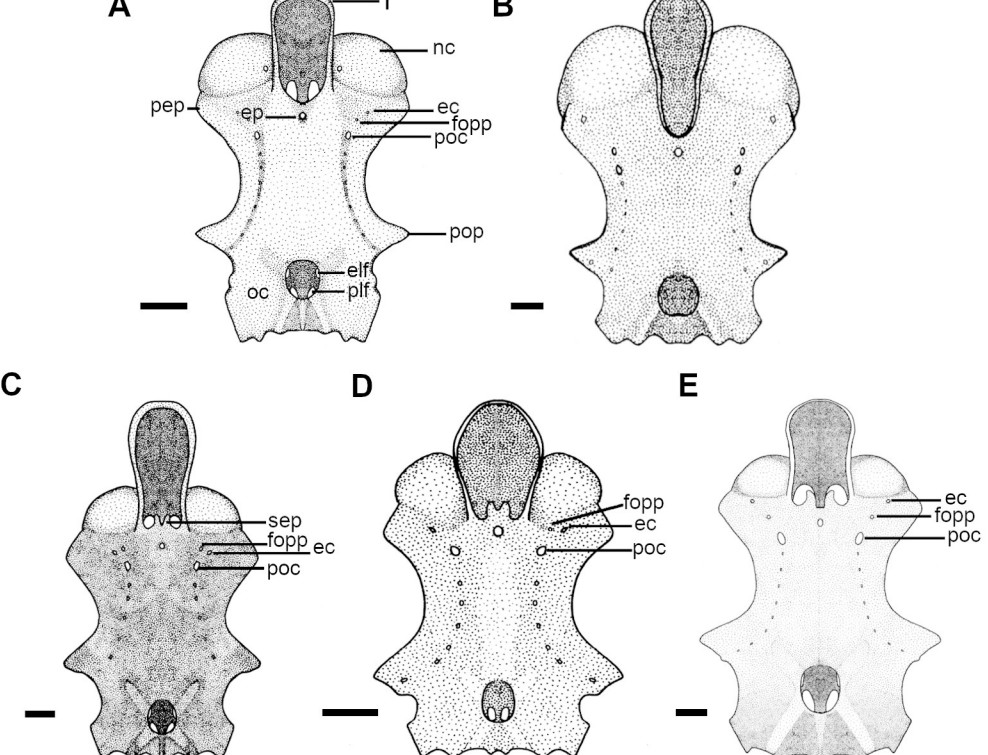

**Fig 2. Neurocranium of species of *Cirrhigaleus* and *Squalus* in dorsal view.** A: *C. barbifer*, HUMZ 95177, juvenile female, 584 mm TL; B: *C. australis*, CSIRO H7042-04, juvenile female, 605 mm TL; C: *S. suckleyi*, HUMZ 87643, adult male, 665mm TL; D: *S. megalops*, AMS I46093-001, adult male, 650 mm TL; E: *S. mitsukurii*, NSMT P-44381, juvenile male, 770 mm TL. Abbreviations: ec: ethmoidal canal of ophthalmicus superficialis nerve; elf: endolymphatic foramen; ep: epiphysial pit; foop: profundus canal for the ophthalmicus profundus nerve; nc: nasal capsule; oc: otic capsule; pep: preorbital process; plf: perilymphatic foramen; poc: preorbital canal of superficial ophthalmic nerve; pop: postorbital process; r: rostrum; sep: supraethmoidal process. Scale bars: 10mm.

condition is observed in species of *Cirrhigaleus* and *Squalus* (8,1). *Dalatias* and *Isistius* show rostrum reduced and not trough-shaped (8,0; see [52]).

9) Rostral keel in the neurocranium: character state 0 –absent; character state 1 –present (CI = 100; RI = 100).

Rostral keel is a ventral expansion of the rostrum, located between the nasal capsules. *Dalatias* and *Isistius* lack rostral keel in the neurocranium (9,0) whereas in all other taxa examined this keel is present (9,1).

10) Rostral keel extension: character state 0 –short, never transcending anterior margin of nasal capsules; character state 1 –elongate, always transcending anterior margin of nasal capsules (CI = 100; RI = 100).

Although *Squalus* and *Cirrhigaleus* present a rostral keel, this structure differs in length (Fig 4). *Cirrhigaleus* has short rostral keel, never transcending anteriorly the nasal capsules (10,0). *Squalus* has rostral keel very elongate, transcending anteriorly the nasal capsules (10,1).

11) Rostral appendages in the neurocranium: character state 0 –absent; character state 1 –present (CI = 100; RI = 100).

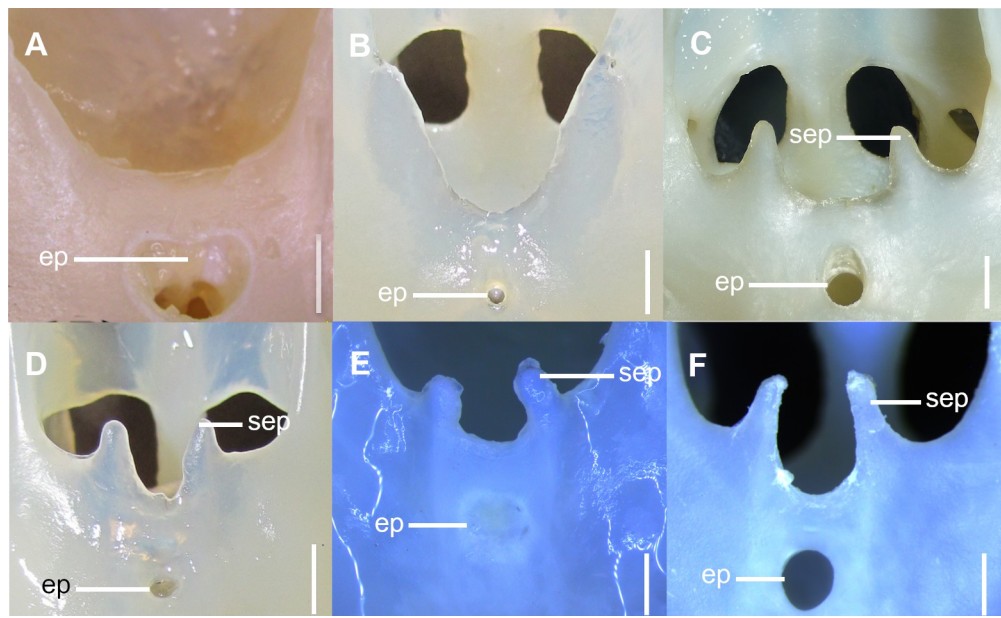

**Fig 3. Detail of ethmoidal region of the neurocranium in anterior dorsal view.** A: *C. australis*, CSIRO H7042-04, juvenile female, 605 mm TL; B: *C. asper*, SAM 38269, adult female, 1045 mm TL; C: *C. asper*, UFPB 11864, juvenile female, 970 mm TL; D: *S. suckleyi*, SAM 38346, adult female, 850 mm TL; E: *S. albifrons*, MZUSP 121272, adult male, 760 mm TL; F: *S. montalbani*, MZUSP121270, adult male, 713 mm TL. Abbreviations: ep: epiphyseal pit; sep: supraethmoidal process. Scale bars: 2mm (A,E,F), 5mm (B–D).

Rostral appendages may be present laterally and medially as hook-like projections for connecting anterolaterally the nasal capsules to the ventral base of the rostrum by ligaments. *Dalatias* and *Isistius* have no rostral appendages. These are also absent in species of *Cirrhigaleus* (11,0) but present in all *Squalus* species examined (11,1).

12) Number of rostral appendages: character state 0 –one pair of lateral rostral appendages; character state 1 –one pair of lateral rostral appendages plus one medial rostral appendage (CI = 100; RI = 100).

In *S. acanthias*, *S. suckleyi* (Fig 4C), *S. megalops* (Fig 4D), *S. brevirostris*, *S. albifrons*, one pair of hook-like rostral appendages is present laterally at the tip of the rostrum (12,0). In *S. mitsukurii* (Fig 4E), *S. montalbani* (Fig 5A) and *S. japonicus*, one pair of rostral appendages is present and additionally one single hook-like medial rostral appendage is present (12,1). *S. acanthias*, *S. suckleyi*, *S. megalops*, *S. brevirostris*, *S. albifrons* (Fig 5B) exhibit a similar medial rostral appendage, herein called median rostral prominence, but this structure is not hook-like.

13) Dorsal aperture of *profundus* canal: character state 0 –absent; character 1 –present (CI = 100; RI = 100).

In *Cirrhigaleus* and *Squalus*, *profundus* canal has one ventral aperture placed in the interorbital wall and a second aperture that opens up to the ethmoidal region, called herein dorsal aperture (13,1) [22, 63]. Stated that the *profundus* nerve runs together with superficial nerves in the preorbital canal in *Dalatias* and *Isistius*, thus, *profundus* canal is absent (13,0).

14) Position of the dorsal aperture of *profundus* canal: character state 0 –aside the ethmoidal canal; character state 1 –between ethmoidal canal and preorbital canal (CI = 50; RI = 75).

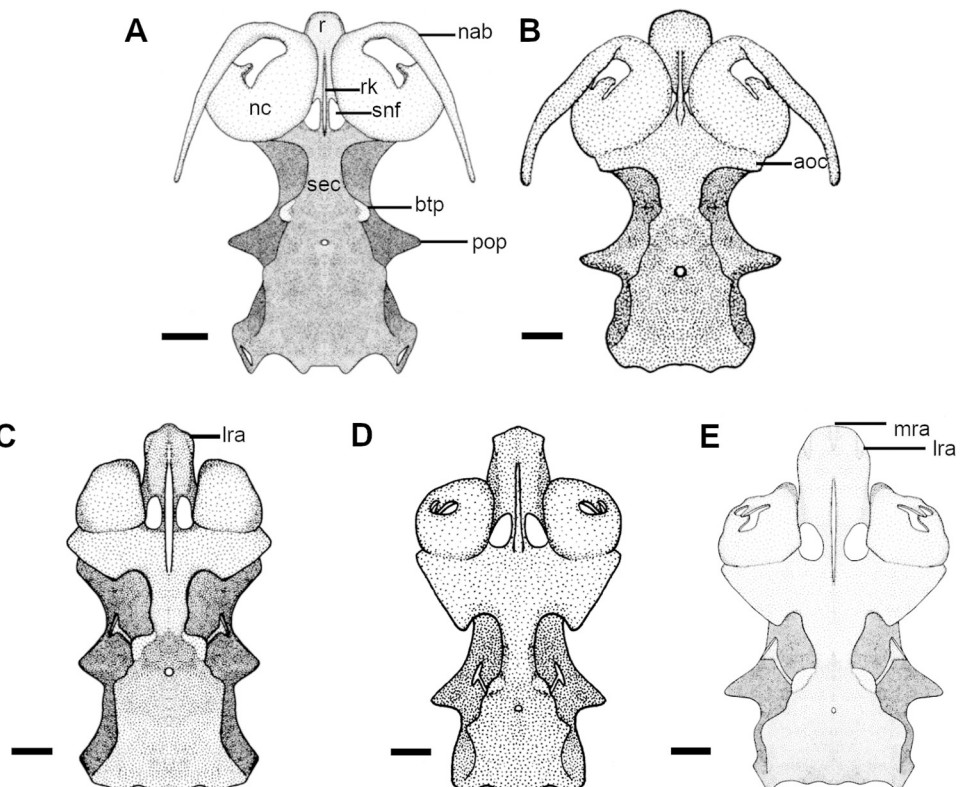

**Fig 4. Neurocranium of species of *Cirrhigaleus* and *Squalus* in ventral view.** A: *C. barbifer*, HUMZ 95177, juvenile female, 584 mm TL; B: *C. australis*, CSIRO H7042-04, juvenile female, 605 mm TL; C: *S. suckleyi*, HUMZ 87643, adult male, 665mm TL; D: *S. megalops*, AMS I46093-001, adult male, 650 mm TL; E: *S. mitsukurii*, NSMT P-44381, juvenile male, 770 mm TL. Abbreviations: aoc: antorbital cartilagem; btp: basitrabecular process; nab: nasal barbel; nc: nasal capsule; pop: postorbital process; r: rostrum; rk: rostral keel: sec: subethmoid chamber; snf: subnasal fenestra; lra: lateral rostral appendage; mra: medial rostral appendage. Scale bars: 10mm.

In *S. acanthias*, *S. suckleyi*, *S. megalops*, *S. brevirostris*, and *S. albifrons profundus* canal has its dorsal aperture placed aside the ethmoidal canal, at the base of the nasal capsule (14,0). In *Cirrhigaleus* and *S. mitsukurii*, *S. montalbani*, and *S. japonicus* the dorsal aperture is between the ethmoidal canal and the preorbital canal at the longitudinal sulcus (14,1).

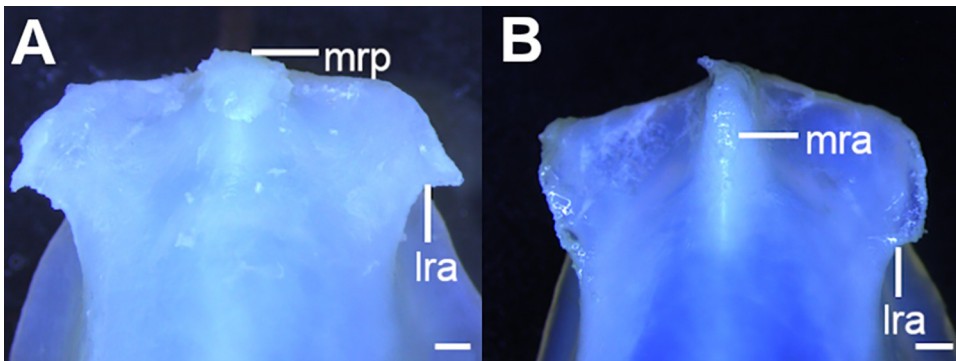

**Fig 5. Detail of rostral process of *Squalus* species in ventral view, showing rostral appendages.** A: *S. albifrons*, MZUSP 121272, adult male, 760 mm TL; B: *S. montalbani*, MZUSP121270, adult male, 713 mm TL. Abbreviations: lra: lateral rostral appendage; mra: medial rostral appendage; mrp: median rostral prominence. Scale bars: 1 mm.

15) Relative width between postorbital processes and nasal capsules of the neurocranium: character state 0 –width of the across postorbital processes greater than to nasal capsules; character state 1 –neurocranium equally wide across nasal capsules and postorbital processes; character state 2 –neurocranium with greatest width across nasal capsules (CI = 50; RI = 71).

Neurocranium has its greatest width at nasal capsules and/or across postorbital processes. In *Dalatias* and *Isistius*, neurocranium is wider across postorbital processes (15.0) and the same condition is observed for *S. acanthias*, *S. suckleyi*, *S. megalops*, *S. brevirostris*, *S. albifrons* and *S. mitsukurii* (15,1). *Squalus japonicus* has neurocranium equally wide at nasal capsules and postorbital processes. *Squalus montalbani* has its greatest width across nasal capsules and the same is noticed for species of *Cirrhigaleus* (15,2) (Fig 2A and 2B).

16) Antorbital cartilage in the neurocranium: character state 0 –vestigial; character state 1 – developed and expanded posteriorly (CI = 100; RI = 100).

Antorbital cartilage is an extension of the subethmoidal region that is triangular and directed posteriorly, forming the ventral base of the preorbital wall. It is vestigial in *Dalatias* and *Isistius* as well as in *Cirrhigaleus* (16,0). In *Squalus* species, the antorbital cartilage is well developed and expanded posteriorly (16,1) (Fig 4). No homology between this character and the one similarly described for non-squalean sharks and batoids in Shirai (1992) is implied in the present study as this requires further investigation. Character codification follows Shirai (1992, 1996).

**Pectoral and pelvic skeleton.**

17) Number of pectoral basals: character state 0 –three distinct pectoral basals; character state 1 –one single pectoral basal (CI = 100; RI = 100).

Pectoral fins are supported internally by pectoral basal and radials that vary in shape and name across Squaliformes. *Dalatias* and *Isistius* have one single basal in the pectoral fin (17,1). *Squalus* and *Cirrhigaleus* have three separated pectoral fin basals (17.0) (Fig 6).

18) Number of regions for pectoral articulation: character state 0 –one articular region; character state 1 –two articular regions (CI = 100; RI = 100).

In elasmobranchs, pectoral articular regions between the pectoral basals of the pectoral fin and pectoral girdle vary in number of articular surfaces and the types (condyle or facet) of articular surfaces [32]. This character refers to the number of surfaces in the scapula for articulation between the pectoral basals and pectoral girdle. As *Dalatias* and *Isistius* have one pectoral basal the pectoral fins articulate through a single articular region (18,0) while *Squalus* and *Cirrhigaleus* have two separate articular regions (18,1).

19) Articular region of pectoral propterygium: character state 0 –condyle; character state 1 – facet (CI = 100; RI = 100).

A condyle for articulation with the pectoral propterygium is found in *Cirrhigaleus* species (19,0) (Fig 7). In *Squalus*, a facet articulates with the propterygium (19,1). Since *Dalatias* and *Isistius* present one single pectoral basal and thus not homologous to the condition observed in *Cirrhigaleus* and *Squalus*, this character and also character 20 were coded as inapplicable for both taxa.

20) Articular region of pectoral mesopterygium and metapterygium: character state 0 –one condyle for articulation with mesopterygium only; character state 1 –one condyle for articulation with both mesopterygium and metapterygium (CI = 100; RI = 100).

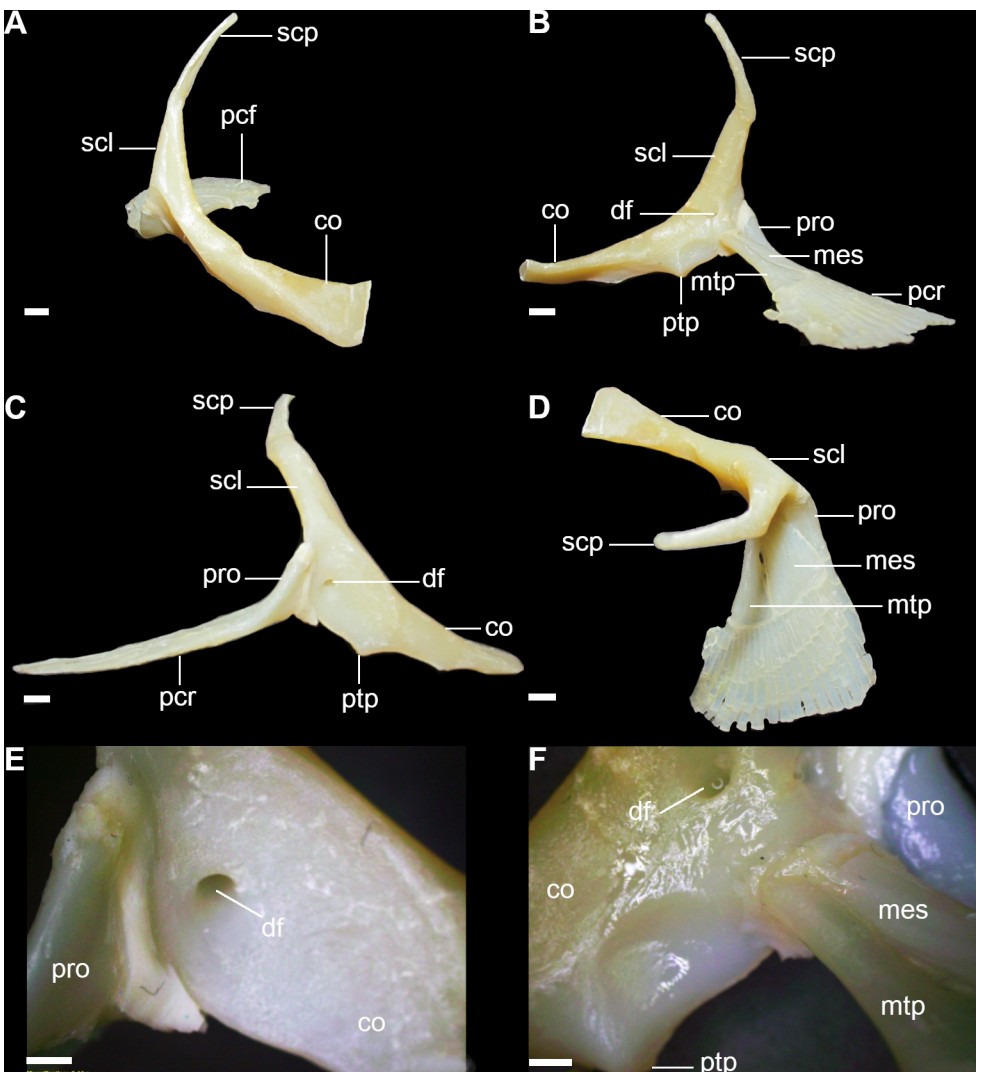

**Fig 6. Pectoral skeleton of *C. australis* (CSIRO H7042-04, juvenile female, 605 mm TL), showing pectoral girdle and pectoral basals.** A: anterior view; B: posterior view; C: lateral view; D: dorsal view. Abbreviations: co: coracoid bar; df: diazonal foramen; mes: mesopterygium; mtp: metapterygium; pcf: pectoral fin; pcr: pectoral fin radials; pro: propterygium; ptp: posterior triangular process; scl: scapula; scp: scapular process. Scale bars: 5mm (A–D); 2mm (E,F).

*Squalus acanthias* and *S. suckleyi* have a condyle for articulation with mesopterygium and the metapterygium does not have a direct articulation with the scapula in these two species but it articulates lateral-proximally to the mesopterygium (20,0). In *Cirrhigaleus*, *S. megalops*, *S. brevirostris*, *S. albifrons*, *S. mitsukurii*, *S. montalbani*, and *S. japonicus*, one condyle articulates with both mesopterygium and metapterygium (20,1) (Fig 7).

21) Dorsal ridge of scapula: character state 0 –smooth ridge; character state 1 –segmented ridge (CI = 100; RI = 100).

The *depressor pectoralis* fossa exhibits a smooth and cylindrical dorsal ridge in the scapula in *Dalatias*, *Isistius* and *Cirrhigaleus* (21,0) (Fig 7A–7F). Species of *Squalus* also show cylindrical ridge in this region with a few exceptions. *Squalus acanthias* (Fig 7G–7I) and *S. suckleyi* have scapula with segmented ridge comprised by small barrel-shaped units located lateral-

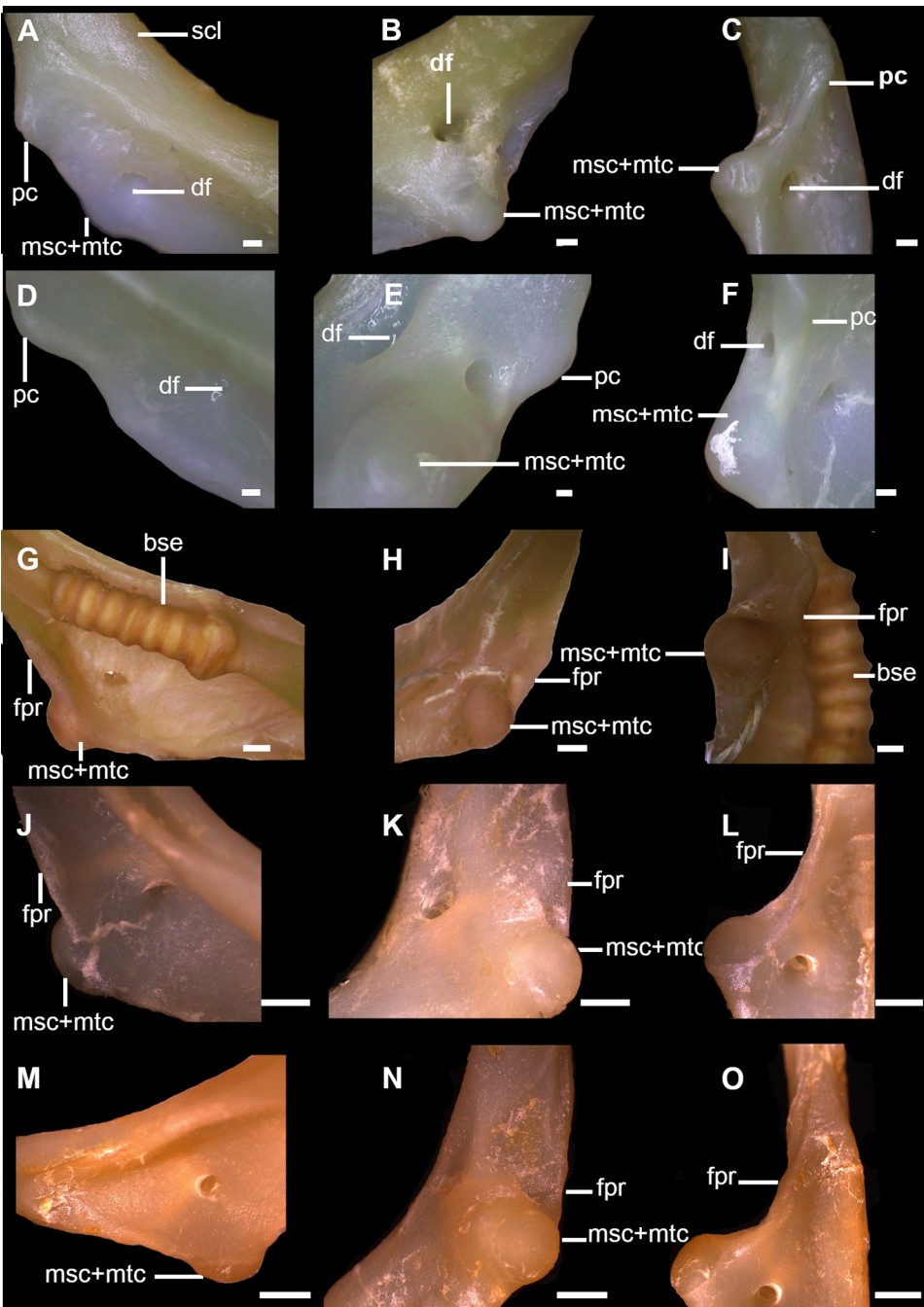

**Fig 7.** Detail of coracoid bar of *Cirrhigaleus* and *Squalus* species in anterior (A,D,G,J,M), posterior (B,E,H,K,N) and lateral views (C,F,I,L,O), showing the pectoral articular region between the pectoral girdle and pectoral basals. A–C: *C. australis*, CSIRO H7042-04, juvenile female, 605 mm TL; D–F: *C. asper*, SAM 38269, adult female, 1045 mm TL; G–I: *S. acanthias*, SAM 38276, adult male, 670mmTL; J–L: *S. albifrons*, MZUSP 121272, adult male, 760 mm TL; M–O: *S. montalbani*, MZUSP121270, adult male, 713 mm TL. Abbreviations: bse: barrel-shaped elements; df: diazonal foramen; fpr: facet for propterygium; msc: mesocondyle; mtc: metacondyle; pc: procondyle; scl: scapula. Scale bars: 2 mm.

dorsally to the *depressor pectoralis* fossa (21,1). These barrel-shaped units were observed in [64, 65] for *S. acanthias*. [66] suggested that this character may be an autapomorphy for this species but our analysis shows that the barrel-shaped elements may or may not be present in *S. suckleyi*.

22) Articular region for the basipterygium: character state 0 –one facet articulating with the basipterygium; character state 1 –one facet and one condyle articulating with basipterygium (CI = 100; RI = 100).

Puboischiadic bar of the pelvic girdle articulates with the pelvic fin through two different regions, one articular region for the articulation with the anterior pelvic basal (= first enlarged radial as in [66] of the pelvic fin and a second one with the basipterygium of the pelvic fin. The type of surfaces for the articulation with the pelvic fin may vary between facet or condyle. In *Dalatias* and *Isistius*, puboischiadic bar has one pelvic condyle posterolaterally on each side articulating with the anterior pelvic basal, and an inner facet posterolaterally on each side and medially to the pelvic condyle for articulating with the basipterygium. *Squalus* species also exhibit this pattern of pelvic articulation (25,0) as noticed in [66]. *Cirrhigaleus* species have the same pattern of pelvic articulation between the puboischiadic bar and anterior pelvic basal but the articular region for the basipterygium has two articular surfaces, an inner facet and a condyle (25,1) (Fig 8A–8F). Pelvic facet is placed ventromedially and the condyle more dorsolaterally in relation to the facet. This condition was not noticed for *C. barbifer* in [66] possibly due to analysis limitations of the dissected specimen.

23) Number of pelvic foramina: character state 0 –one single foramen; character state 1 –two foramina (CI = 33; RI = 60).

Pubosichiadic bar at the pelvic girdle has one or two foramina for the pelvic nerve, located laterally on each side of the lateral prepelvic process. *Dalatias* and *Isistius* have a single pelvic foramen on each side (26,0). *Squalus acanthias*, *S. suckleyi*, *S. megalops* and *S. brevirostris* present a single foramen as well. *Squalus albifrons*, *S. mitsukurii*, *S. montalbani*, *S. japonicus* and species of *Cirrhigaleus* have two pelvic foramina (Fig 8; 26,1).

24) Posterior medial process of the puboischiadic bar: character state 0 –present; character state 1 –absent (CI = 100; RI = 100).

Puboschiadic bar may be convex medially at the posterior margin, herein named posterior medial process as in [65] or posteriormedian projection as in [66]. This process is present in most taxa examined (27,0) except *Cirrhigaleus* species (27,1).

25) Shape of the posterior medial process of the puboischiadic bar: character state 0 –conspicuously elongate; character state 1 –posterior medial process short (CI = 0; RI = 0).

In *Isistius*, posterior medial process is conspicuously elongate, extending backwards as a triangular projection (28,0). In *Dalatias* and *Squalus* species, posterior medial process is short, not extending backwards as a triangular projection (28,1).

**Clasper morphology.**

26) Number of intermediate segments between the basipterygium of the pelvic fin and the axial cartilage of the claspers: character state 0 –one single; character state 1 –two (CI = 67; RI = 67).

*Dalatias* and *Isistius* have a single intermediate segment articulating the basipterygium of the pelvic fin and the axial cartilage of the claspers (29,0). *Squalus acanthias*, *S. suckleyi*, *S. megalops*, *S. brevirostris*, *Squalus albifrons*, *S. mitsukurii* and *S. japonicus* also show a single

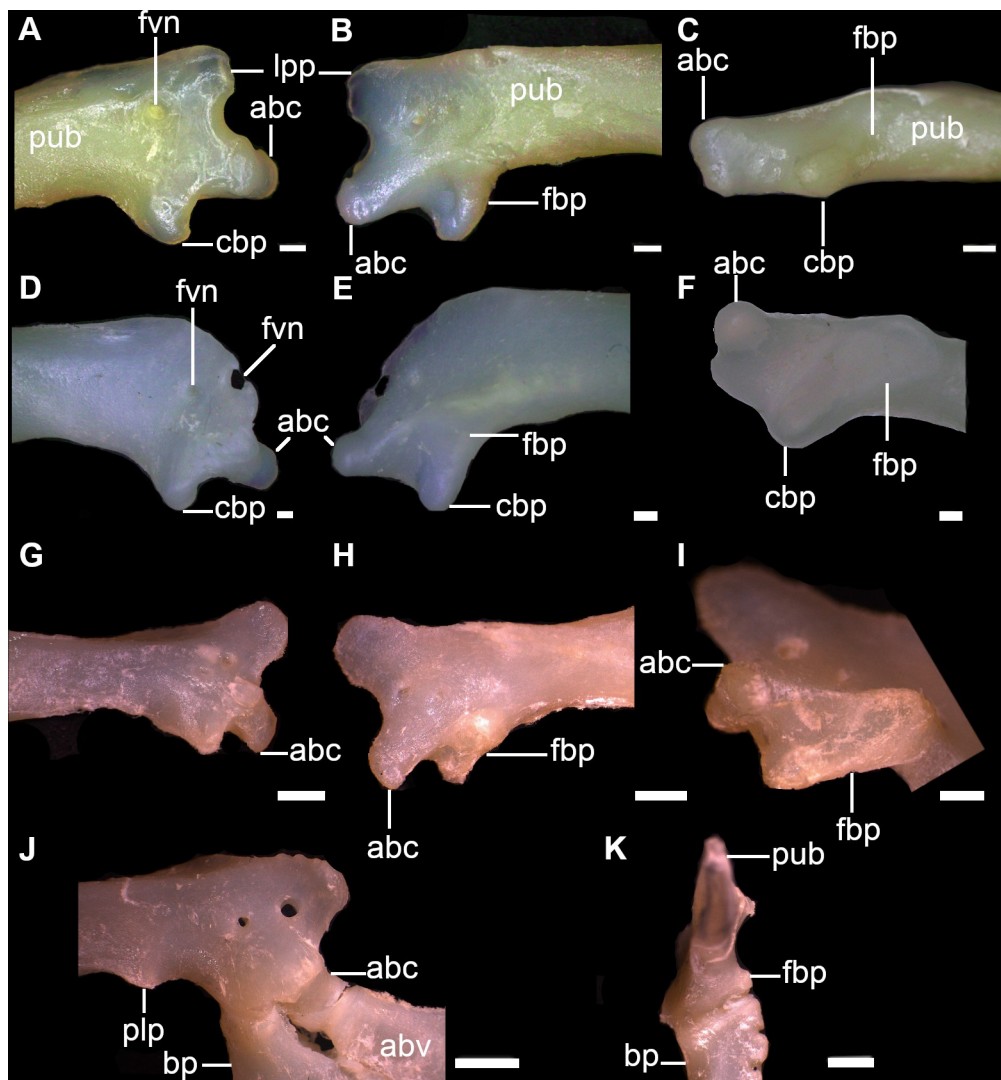

**Fig 8.** Puboischiadic bar (right side) in dorsal (A,D,G,J), ventral (B,E,H), posterior (C,F,I) and medial (K) views showing the pelvic articular region with the pelvic fin. A–C: *C. australis*, CSIRO H7042-04, juvenile female, 605 mm TL; D–F: *C. asper*, SAM 38269, adult female, 1045 mm TL; G–I: *S. montalbani*, MZUSP121270, adult male, 713 mm TL; J,K: *S. albifrons*, MZUSP 121272, adult male, 760 mm TL. Abbreviations: abc: condyle for the anterior pelvic basal; abv: anterior pelvic basal; bp: basipterygium; cbp: condyle for the basipterygium; fpb: facet for the basipterygium; fvn: foramen for ventral fin nerve; lpp: lateral prepelvic proces; plp: posterior-lateral process; pub: puboischiadic bar. Scale bars: 1mm.

intermediate segment (Fig 9). *S. montalbani* and *Cirrhigaleus* have two intermediate segments (26,1) (Fig 10).

27) Position of beta cartilage in claspers: character state 0 –beta cartilage placed over the intermediate segment and at the proximal edge of the axial cartilage; character state 1 –beta cartilage placed over the distal edge of basipterygium, the first intermediate segment and the proximal edge of the axial cartilage (CI = 33; RI = 60).

Beta cartilage comprises a cylindrical element that connects dorsally the basipetrygium to the axial cartilage. Relative extension of the beta cartilage varies within Squalidae. *Dalatias* has one beta cartilage placed over the intermediate segment and partially at proximal edge of the

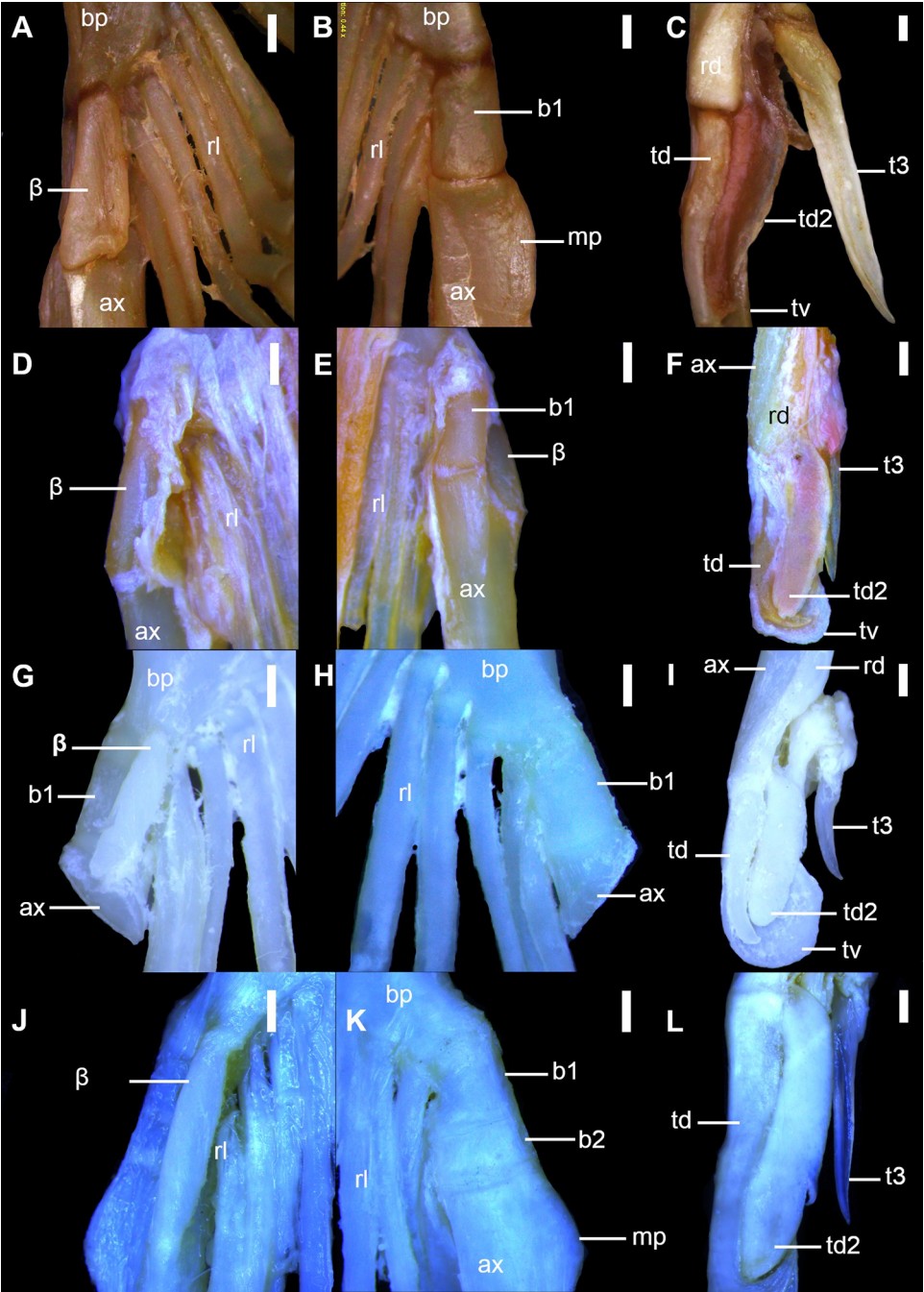

**Fig 9.** Detail of cartilages of the clasper in *Squalus* species in dorsal (A,D,G,J), ventral (B,E,H,K) and terminal dorsal (C,F,I,L) views. A–C: *S. acanthias*, SAM 38276, adult male, 670mmTL; D–F: *S. brevirostris*, HUMZ 189762, adult male, 433 mm TL; G–I: *S. albifrons*, MZUSP 121272, adult male, 760 mm TL; J–L: *S. montalbani*, MZUSP121270, adult male, 713 mm TL. Abbreviations: ax: axial cartilage; bp: basipterygium; b1: first intermediate segment; b2: second intermediate segment; mp: mesial process; rd: dorsal marginal cartilage; rl: pelvic radials; td: dorsal terminal cartilage; td2: dorsal terminal 2 cartilage; tv: ventral terminal cartilage; t3: accessory terminal 3 cartilage; β: beta cartilage. Scale bars: 1 mm.

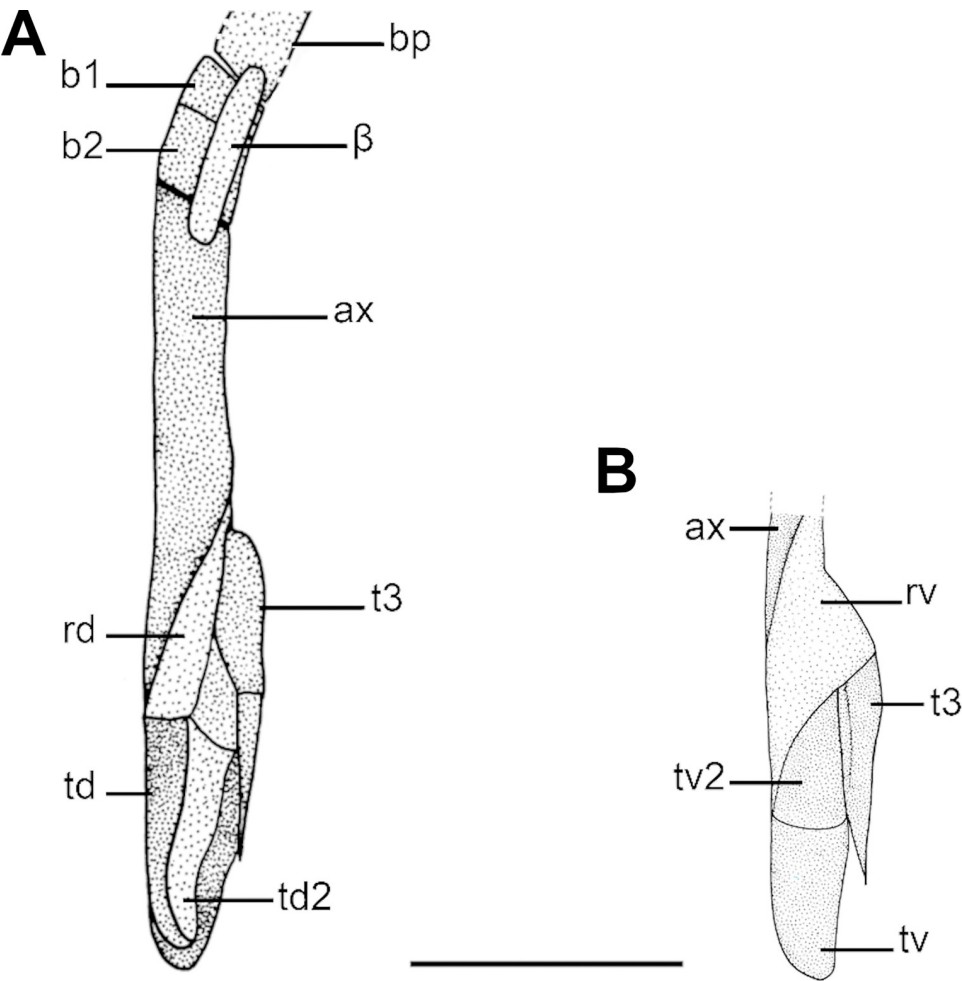

**Fig 10.** Cartilages of the clasper (left side) of *C. australis*, NMNZ P 38074, adult male, 1020 mm TL in dorsal view (A) and detail of terminal cartilages in ventral view (B). Abbreviations: ax: axial cartilage; b1: first intermediate segment; b2: second intermediate segment; rd: dorsal marginal cartilage; rv: ventral marginal cartilage; td: dorsal terminal cartilage; td2: dorsal terminal 2 cartilage; tv: ventral terminal cartilage; tv2: ventral terminal 2 cartilage; t3: accessory terminal 3 cartilage; β: beta cartilage. Scale bar: 10 mm.

axial cartilage of the clasper (27,0). *Squalus acanthias*, *S. suckleyi*, *S. megalops*, *S. brevirostris* and *S. mitsukurii* exhibit the same pattern (Fig 10). *Isistius*, *Squalus albifrons*, *S. montalbani*, and *S. japonicus* as well as species of *Cirrhigaleus* (Fig 10) show beta cartilage placed over the distal edge of basipterygium of the pelvic fin, intermediate segment and the proximal edge of the axial cartilage (27,1).

28) Number of terminal cartilages of the clasper: character state 0 –three terminal cartilages; character state 1 –four terminal cartilages; character state 2 –five terminal cartilages (CI = 100; RI = 100).

Terminal cartilages vary in number within Squaliformes [27]. *Dalatias* and *Isistius* have three terminal cartilages (30,0): accessory terminal 3 cartilage, ventral terminal cartilage and dorsal terminal cartilage. *Squalus* species show four terminal cartilages (28,1) which include these three cartilages plus dorsal terminal 2 cartilage. *Cirrhigaleus* species have five terminal

cartilages (28,2): accessory terminal 3 cartilage, dorsal terminal cartilage, dorsal terminal 2 cartilage, ventral terminal cartilage, and ventral terminal 2 cartilage (Fig 10).

29) Mesial process in the axial cartilage of the claspers: character state 0 –absent; character state 1 –present (CI = 50; RI = 75).

In claspers, the axial cartilage may be straight or exhibit some sinuousity and/or a mesial expansion at the proximal edge, herein called mesial process. In *Dalatias* and *Isistius*, mesial process is absent (29,0). *Squalus megalops*, *S. brevirostris* and *S. albifrons* share the same condition. *Squalus acanthias*, *S. suckleyi*, *S. mitsukurii*, *S. montalbani*, *S. japonicus* as well as species of *Cirrhigaleus* exhibit a conspicuous mesial process in the axial cartilage (29,1).

30) Relative length of ventral terminal cartilage of the claspers: character state 0 –ventral terminal cartilage shorter than one-third the length of axial cartilage; character state 1 –ventral terminal cartilage greater than one-third the length of the axial cartilage; character state 2 – ventral terminal cartilage with length equal to the length of the axial cartilage (CI = 67; RI = 80).

Ventral terminal cartilage of the claspers varies in length in relation to the length of the axial cartilage. *Isistius and Dalatias* have short ventral terminal cartilage whose length is smaller than one-third the length of the axial cartilage (30,0). *Squalus megalops*, *S. brevirostris*, *S. albifrons*, and *S. japonicus* show the same condition. *Squalus mitsukurii*, *S. montalbani* and species of *Cirrhigaleus* (Fig 10) have length of ventral terminal cartilage greater than one-third the length of the axial cartilage (30,1). *Squalus acanthias* and *S. suckleyi* have elongate ventral terminal cartilage that is equal in length to the length of the axial cartilage (30,2).

31) Accessory terminal 3 cartilage (t3 or spur): character state 0 –blade-like; character state 1 – pin-like (CI = 100; RI = 100).

The accessory terminal 3 cartilage (or spur) is blade-like in *Isistius* (31,0). The same condition is observed for *Squalus megalops*, *S. brevirostris*, *S. albifrons*, *S. mitsukurii*, *S. montalbani* and *S. japonicus* as well as species of *Cirrhigaleus*. In *S. acanthias* and *S. suckleyi*, accessory terminal 3 cartilage (or spur) is pin-like (31,1).

**External morphology.**

32) Dentition: character state 0 –dignathic heterodonty; character state 1 –dignathic homodonty (CI = 100; RI = 100).

*Dalatias* and *Isistius* have dignathic heterodonty (32,0). *Squalus* and *Cirrhigaleus* have dignathic homodonty (32,1). This was proposed as synapomorphy of Squalidae in [22].

33) Dorsal fin spines: character state 0 –absent; character state 1 –present (CI = 100; RI = 100).

Dorsal-fin spines may be present or not at the basal cartilage of the dorsal fin. These spines may be located internally thus retained underneath the skin or externally arising above the skin. In *Dalatias* and *Isistius* dorsal-fin spines are absent (33,0). In *Cirrhigaleus* and *Squalus*, dorsal-fin spines are present (33,1).

34) Subterminal notch of caudal fin: character state 0 –present; character state 1 –absent (CI = 100; RI = 100).

Caudal fin may have a subterminal notch at the postventral caudal margin in Squaliformes. *Dalatias* and *Isistius* do have a subterminal notch (34,0). Species of *Cirrhigaleus* and *Squalus* do not have subterminal notch (34,1).

35) Support of the anterior margin of nostrils: character state 0 –nasal lobes supported by flesh only; character state 1 –nasal lobes supported by cartilage (CI = 100; RI = 100).

Anterior margin of nostrils has nasal lobes that are supported internally by flesh or cartilage. In *Dalatias and Isistius*, nasal lobes are supported by flesh (35,0). In *Cirrhigaleus*, the same condition is observed. In *Squalus*, nasal lobes are supported internally by thin cartilages (35,1).

36) Number of nasal lobes at anterior margin of nostrils: character state 0 –two; character state 1 –one (CI = 100; RI = 100).

Anterior margin of nostrils may have one or more nasal lobes, one laterally and a second lobe medially. *Dalatias and Isistius* have a single nasal lobe (36,1). The same condition is observed for *S. acanthias* and *S. suckleyi* but a medial nasal lobe may be present in neonates. *Cirrhigaleus* species have two nasal lobes (36,0) as well as *S. megalops, S. brevirostris, S. albifrons, S. mitsukurii, S. montalbani* and *S. japonicus*.

37) Shape of nasal barbels: character state 0 –non moustache-like; character state 1 –moustache-like (CI = 100; RI = 100).

*Cirrhigaleus barbifer* (Fig 11A) and *C. australis* (Fig 11B) have nasal barbels conspicuously moustache-like and conspicuously elongate, with its tips reaching the mouth (37,1). In *C. asper* (Fig 11C), the nasal barbel is non moustache-like and very short, with its tips never reaching the mouth (37,0). Nasal barbels are absent in *Squalus, Isistius* and *Dalatia* and thus these terminal taxa were coded as "?" for this character.

38) Upper labial furrow: character state 0 –absent; character state 1 –present (CI = 100; RI = 100).

Upper labial furrow is a lateral sulcus formed as a space between the labial cartilage and the quadrate plate. Upper labial furrow is absent in *Dalatias* and *Isistius* (38,0) but present in *Cirrhigaleus* and *Squalus* (38,1).

39) Relative length of upper and lower labial furrows: character state 0 –short; character state 1 –elongate (CI = 100; RI = 100).

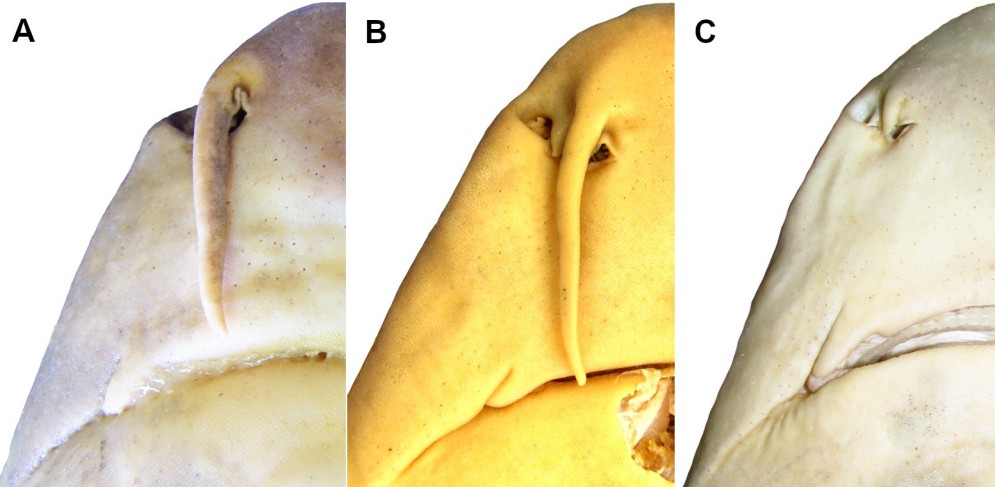

**Fig 11. Ventral view of the head (right side) in *Cirrhigaleus* species showing nasal barbels and upper labial furrows.** A: *C. barbifer*, HUMZ 197852 (neotype), adult female, 870 mm TL; B: *C. australis*, AMS I 27022–001 (paratype), adult female, 1205 mm TL; C: *C. asper*, uncatalogued specimen, adult female, 1068 mm TL.

Upper labial furrow length is much shorter than lower labial furrow in *Cirrhigaleus* species (39,0) whereas upper labial furrow is as elongate as the lower labial furrow in species of *Squalus* (39,1).

40) Dorsal fin shape: character state 0 –dorsal fins similar in shape; character state 1 –dorsal fins dissimilar in shape (CI = 100; RI = 100).

First and second dorsal fins are similar in shape in *Dalatias* and *Isistius* (40,0). *Cirrhigaleus* and *Squalus* have dissimilar dorsal fins (40,1).

41) Dorsal fin length: character state 0 –dorsal fins similar in length; character state 1 –dorsal fins dissimilar in length (CI = 50; RI = 80).

First and second dorsal fins may be similar in length. This condition is observed in *Dalatias*, *Isistius*, *Cirrhigaleus*, *S. acanthias* and *S. suckleyi* (41,0). *Squalus megalops*, *S brevirostris*, *S. albifrons*, *S. mitsukurii*, *S. montalbani* and *S. japonicus* have first dorsal fin greater in length than second dorsal fin (41,1).

42) Proportional length of dorsal-fin inner margin and height: character state 0 –dorsal-fin inner margin length equal to dorsal fin height; character state 1 –dorsal-fin inner margin length smaller than dorsal fin height (CI = 50; RI = 67).

Inner margin of dorsal fins may be similar in length to the dorsal fin height. This condition is present in *Dalatias*, *Isistius*, *S. acanthias* and *S. suckleyi* (42,0). *Cirrhigaleus* species and *Squalus megalops*, *S. brevirostris*, *S. albifrons*, *S. mitsukurii*, *S. montalbani* and *S. japonicus* have dorsal-fin inner margin much smaller than dorsal fin height (42,1).

43) Proportional length between pectoral-pelvic fins and pelvic-caudal fins: character state 0 –pectoral-pelvic space equal or greater pelvic-caudal space; character state 1 –pectoral-pelvic space smaller than pelvic-caudal space (CI = 50; RI = 80).

Distance between pectoral and pelvic fins is equal or greater than the distance between pelvic and caudal fins in *Dalatias*, *Isistius*, *Cirrhigaleus* species, *S. acanthias* and *S. suckleyi* (43,0). In *S. megalops*, *S. brevirostris*, *S. albifrons*, *S. mitsukurii*, *S. montalbani* and *S. japonicus*, pectoral-pelvic space is smaller than pelvic-caudal space (43,1).

44) Precaudal pits: character state 0 –absent; character state 1 –present (CI = 100; RI = 100).

The caudal penduncle may exhibit an upper precaudal pit which is a small notch or concavity placed just prior the origin of the caudal fin. In *Dalatias and Isistius*, precaudal pits are absent (44,0). The same condition is observed for *Cirrhigaleus*. Species of *Squalus* have precaudal pits (44,1) as it was noticed in [22] for the genus.

45) Precaudal keel: character state 0 –absent; character state 1 –present (CI = 0; RI = 0).

A longitudinal keel may be present laterally at the caudal peduncle, extending beyond the origin of the lower caudal lobe. Precaudal keel is absent in *Dalatias* (45,0) but present in *Isistius*, *Cirrhigaleus* and species of *Squalus* (45,1).

46) Caudal fin shape: character state 0 –rectangular; character state 1 –triangular (CI = 0; RI = 0).

Dorsal caudal margin is usually more elongate than the ventral caudal margin which confer a rectangular shape to the caudal fin. Sometimes the ventral caudal lobe is as elongate as the dorsal caudal margin and thus the caudal fin is triangular in shape. *Isistius* has this pattern of

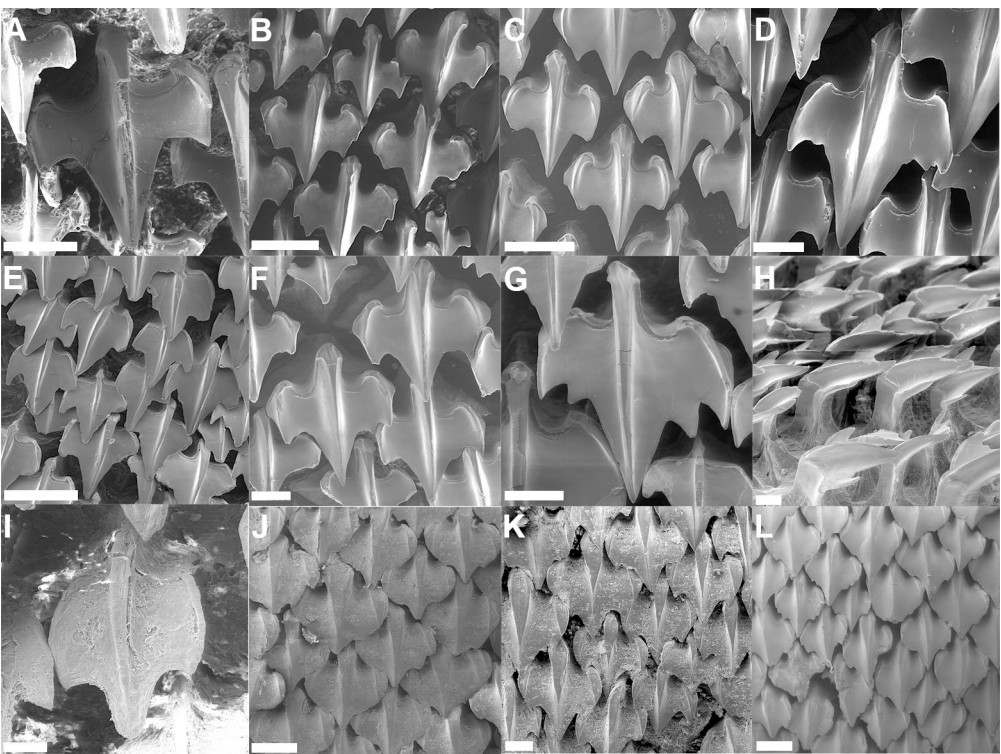

**Fig 12.** Scanning electron microscopy of dermal denticles of species of *Cirrhigaleus* in coronal (A–L, except H) and lateral (H) views. *C. barbifer* (A–C): A: HUMZ 197852 (neotype), adult female, 870 mm TL; B: NMW 98257, adult female, 960 mm TL; C: CSIRO H 5875–09, adult female, 978 mm TL; *C. australis* (D–H): D: NMNZ P 38074, adult male, 1020 mm TL; E: NMNZ P 42489, juvenile male, 710 mm TL; F: CSIRO H 7048–01, adult male, 993 mm TL; G–H AMS I 45670–001, juvenile male, 630 mm TL. *C. asper* (I–L): I: SAIAB, 6092, juvenile female, 275 mm TL; J: SAIAB 31890, adult female, 1090 mm TL; K: SAM 39879, adult female, 1023 mm TL; L: NUPEC uncatalogued, adult female, 1270 mm TL. Scale bars: 100 μm (I); 200 μm (A,D,G,J–L); 500 μm (B,C,E,F,H).

caudal fin shape (46,1) whereas *Dalatias*, *Cirrhigaleus* and *Squalus* have rectangular caudal fin (46,0).

**Squamation.**

47) Cusps at the crown: character state 0 –absent; character state 1 –present (CI = 100; RI = 100).

Crown of the dermal denticles may exhibit medial and lateral cusps. In *Dalatias* and *Isistius*, dermal denticles have no cusps (47,0) while in *Cirrhigaleus* (Fig 12) and *Squalus*, cusps are present (47,1).

48) Cusplets at the crown: character state 0 –inconspicuous; character state 1 –conspicuous (CI = 100; RI = 100).

Cusplets are structures that are accessory to the main cusp and it is usually smaller in length. The cusplets are located medially to the lateral cusp on either side of the posterior margin of the crown base. In *C. asper* (Fig 12I–12L) these cusplets are inconspicuous (48,0) as in *Squalus*. In *C. barbifer* and *C. australis* (Fig 12A–12H) these cusplets are conspicuous (48,1).

49) Ridges on the crown surface: character state 0 –absent; character state 1 –present (CI = 100; RI = 100).

Dermal denticles present medial and lateral ridges at the dorsal surface of the crown in most taxa examined (49,1). *Dalatias* and *Isistius* do not have medial and lateral ridges (49,0).

50) Number of ridges in the dermal denticles: character state 0 –three ridges; character state 1 –single ridge (CI = 100; RI = 100).

*Cirrhigaleus*, *Squalus megalops*, *S. brevirostris*, *S. albifrons*, *S. mitsukurii*, *S. montalbani* and *S. japonicus* have three ridges, one medial and two lateral ridges (50,0). *Squalus acanthias* and *S. suckleyi* have a single medial ridge in the dermal denticle (50,1).

51) Dermal denticles shape: character state 0 –diamond-shaped; character state 1 –arrow-shaped; character state 2: club-shaped; character state 3: heart-shaped; (CI = 100; RI = 100).

The crown shape of dermal denticles vary in shape. *Dalatias* and *Isistius* have crown diamond-shaped (51,0). *Cirrhigaleus*, *S. mitsukurii*, *S. montalbani* and *S. japonicus* have heart-shaped dermal denticles (51,1). *Squalus acanthias* and *S. suckleyi* present dermal denticles arrow-shaped (51,2) whereas *S. megalops*, *S. brevirostris*, *S. albifrons* have club-shaped dermal denticles (51,3).

## Morphological phylogenetic reconstruction

Maximum parsimony analysis performed herein included 51 morphological characters (four quantitative and 47 qualitative) and 13 terminal taxa. We chose the cladistic analysis using implied weighting with a concavity constant equal to 1 as the preferred hypothesis, considering its stronger support and greatest internal resolution of clades. This analysis resulted in two equally most-parsimonious trees with 69.9 steps, CI = 0.80 and RI = 0.88. The character matrix was divided into quantitative characters with absolute and normalized values (S1 Table) and qualitative characters (S2 Table) for each terminal analyzed. The list of synapomorphies presented below refers to the family Squalidae, *Squalus* and *Cirrhigaleus*. Relative Bremer support and GC values are shown in the cladogram (Fig 13) below each node. Morphological evidence that supports each node is summarized below.

Clade A corresponds to the family Squalidae that is supported by the following synapomorphies: higher values of diplospondylous vertebrae (character 2, 0.190–0.286>0.690.0.786); rostrum developed and trough-shaped (ch. 8, 0>1); rostral keel present in the neurocranium (ch. 9, 0>1); presence of a dorsal aperture of *profundus* canal (ch. 13, 0>1); neurocranium equally wide across nasal capsules and postorbital processes (ch. 15, 0>1); three separate pectoral basals (ch. 17, 0>1); separate articular surfaces for the pectoral basals (ch. 18, 0>1); four terminal cartilages on claspers (ch. 28, 0>1); mesial process present in the axial cartilage of the claspers (ch. 29, 0>1); ventral terminal cartilage greater than one-third the length of the axial cartilage (ch. 30, 0>1); dignathic homodonty (ch. 32, 1>0); presence of a subterminal notch of caudal fin (ch. 374 0>1); two nasal lobes at anterior margin of nostrils (ch. 36, 1>0); upper labial furrow present (ch. 38, 0>1); dorsal fins dissimilar in shape (ch. 40, 0>1); presence of cusps at the crown denticle (ch. 47, 0>1); ridges present on the crown surface (ch. 49, 0>1); dermal denticles not diamond-shaped (ch 51, 0>1). Two clades, clades B and I, are hypothesized within Clade A.

Clade B consists of all examined species of *Squalus* and is supported by five synapomorphies: presence of supraethmoidal process in the neurocranium (ch. 7,0>1); rostral appendages present in the neurocranium (ch. 11, 0>1); antorbital cartilage developed and expanded posteriorly (ch. 16, 0>1); nasal lobes supported by cartilage (ch. 35, 0>1); presence of precaudal pits (ch. 44, 0>1). *Squalus acanthias* and *S. suckleyi* (Clade C) are placed at the base of *Squalus* and grouped by six synapomorphies: single condyle for articulation with

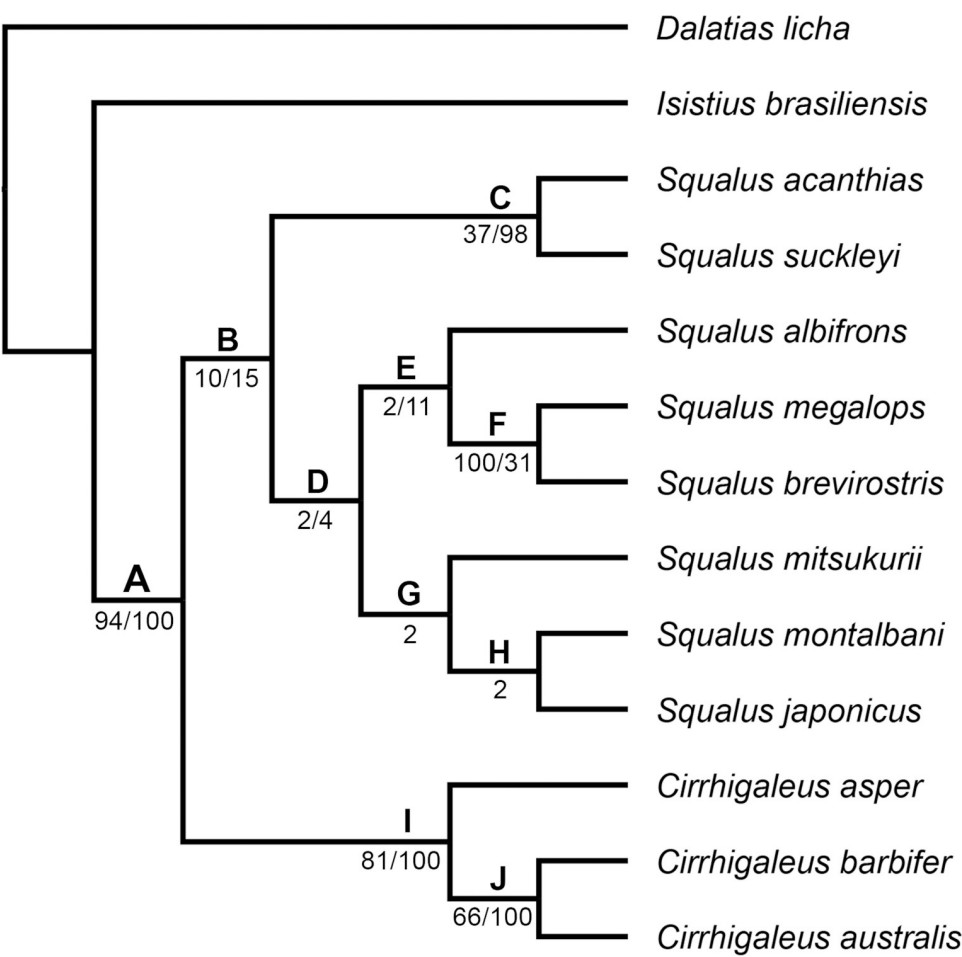

**Fig 13. Strict consensus cladogram of the two equally most-parsimonious cladograms of 69.9 steps resulting from the analysis of 51 morphological characters (CI = 0.80; RI = 0.88).** Relative Bremer support values are given below each node.

mesopterygium (ch. 20, 1>0), segmented dorsal ridge of scapula (ch. 21, 0>1), ventral terminal cartilage with length equal to the length of the axial cartilage (ch. 30, 1>2); one ridge in the dermal denticles (ch. 50, 0>1) and arrow-shaped denticles (ch. 51, 1>2). *Squalus acanthias* is characterized by higher values of monospondylous vertebrae (ch. 1, 0.375>0.563–0.813) whereas *S. suckleyi* has no autapomorphies.

Clade D is composed of all other *Squalus* species examined and supported by three synapomorphies: higher number of diplospondylous vertebrae counts (ch. 2, 0.69–0.79>0.88), dorsal fins dissimilar in length (ch. 41, 0>1) and pectoral-pelvic space smaller than pelvic-caudal space (ch. 43, 0>1).

Clade E (*S. albifrons*, *S. brevirostris* and *S. megalops*) is supported by two synapomorphies: mesial process absent in the axial cartilage of the claspers (ch. 29, 1>0) and ventral terminal cartilage greater than one-third the length of the axial cartilage (ch. 30, 1>0). *Squalus albifrons* is placed at the base of this clade and has no autapomorphies. *Squalus brevirostris* and *S. megalops* (Clade F) are hypothesized as closely related and this relationship is supported by lower values of monospondylous vertebrae (ch. 1, 0.375>0.250–0.313); no autapomorphies were found for both taxa.

Clade G consisting of *S. mitsukurii*, *S. japonicus* and *S. montalbani* is supported by the presence of one pair of lateral rostral appendages plus one medial rostral appendage (ch. 12, 0>1). *Squalus mitsukurii* has no autapomorphies. A close relationship between *S. japonicus* and *S. montalbani* (Clade H) is hypothesized based on equally wide nasal capsules and postorbital processes (ch. 15, 0>1). *Squalus japonicus* is characterized by a ventral terminal cartilage shorter than one-third the length of axial cartilage (ch. 30, 1>0) whereas *S. montalbani* is characterized by a neurocranium with greatest width across nasal capsules (ch. 15, 1>2) and two intermediate segments between the basipterygium of the pelvic fin and the axial cartilage of the claspers (ch. 26, 0>1).

Clade I consists of all species of *Cirrhigaleus* and its monophyly is supported by eight synapomorphies: higher values of monospondylous vertebrae (ch. 1, 0.375>0.750); presence of an innervation of the medial nasal lobe by the buccopharyngeal branch of the facial nerve (ch. 5, 0>1); fleshy core present at the anterior margin of nostrils (ch. 6, 0>1); neurocranium with greatest width across nasal capsules (ch. 15, 1>2); one facet and one condyle for articulating with basipterygium of the pelvic fin (ch. 22, 0>1); two intermediate segments between the basipterygium and the axial cartilage of the claspers (ch. 26, 0>1); five terminal cartilages of the claspers (ch. 28, 1>2); posterior medial process of the puboischiadic bar absent (ch. 24, 0>1). *Cirrhigaleus asper* is hypothesized as sister-group of the clade comprising *C. australis* and *C. barbifer* (Clade J) and this species has no autapomorphies. Clade J is supported by the presence of conspicuous cusplets at the crown of dermal denticles (ch. 48, 0>1); *C. australis* and *C. barbifer* have no autapomorphies.

## Discussion

### Systematics of Squalidae

Phylogenetic analysis of 51 morphological characters supports the monophyly of Squalidae which is congruent with previous hypothesis of [15, 22]. Out of the 18 synapomorphies for the clade, only one was previously proposed: dignathic homodonty in [22]. Presence of a facet on the pectoral articular region for propterygium was proposed as synapomorphy of Squalidae in [32] and again in [49, 51] which is in contrast to our current analysis. *Cirrhigaleus* share a condition similar to those observed for Etmopteridae (as described in [51]): a condyle for the propterygium and a condyle for the mesopterygium and metapterygium.

Some synapomorphies, however, require revision as other members within Squaliformes that were left out of the present analysis. Squalidae share the same conditions related to the number of pectoral basals with Etmopteridae and *Echinorhinus* [22, 37, 48, 67], and the number of pectoral articular regions with Chlamydoselachidae [32] and Etmopteridae [51]. Other characters, including rostral keel in the neurocranium, and number of terminal cartilages on claspers, number of nasal lobes at nostrils, shape of dermal denticles and number of vertebrae have been overlooked for Squaliformes. A more inclusive cladistic analysis of morphological characters for representatives of all families within Squaliformes is required in order to revise the synapomorphies proposed here for Squalidae and test the influence of homoplastic characters. As the scope of this study is to elucidate the taxonomic classification of *Cirrhigaleus* and to test its inner phylogeny we will no further discuss the relationships of Squalidae and its monophyly within the order.

### Monophyly of genera and inner relationships

Morphological phylogenetic reconstruction undertaken in this study supports the monophyly of *Cirrhigaleus*. The genus comprises a clade that is sister-group to a second clade, representing the genus *Squalus*. Thus, the current analysis supports the validity of *Cirrhigaleus* as suggested

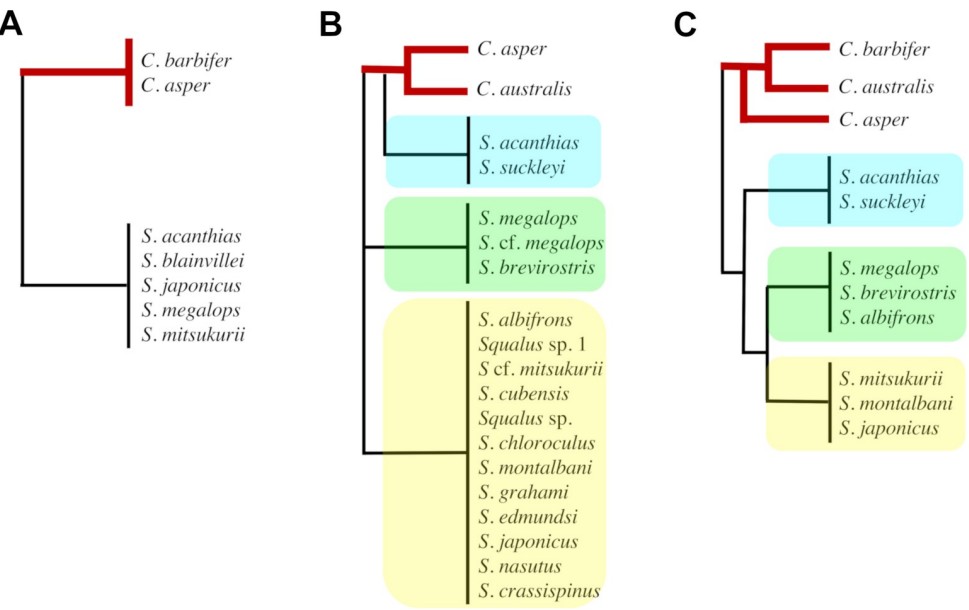

**Fig 14. Simplified cladograms of previous phylogenetic hypotheses for Squalidae, highlighting the relationships of *Cirrhigaleus* species (thick red branches).** Interrelationships between the three major groups of species within *Squalus* are also shown. A: Shirai [22]; B: Naylor et al. [15]; C: present study. Blue rectangle: *S. acanthias* group; Green rectangle: *S. megalops* group; Yellow rectangle: *S. mitsukurii* group.

in [18, 19, 22] but refute the molecular hypothesis using NADH2 gene of [15] in reallocating *Cirrhigaleus* species into the genus *Squalus* (Fig 14).

For *Cirrhigaleus*, besides the synapomorphy raised in [22] that is the presence of an innervation of the medial nasal lobe by the buccopharyngeal branch of the facial nerve (ch. 5 in the present analysis) other seven novel synapomorphies are hypothesized in the current analysis. *Cirrhigaleus* is furthermore defined as a genus of roughskin dogfish sharks with higher values of monospondylous vertebrae counts, medial nasal lobe supported by fleshy core and innervated by the buccopharyngeal branch of the facial nerve, neurocranium with greatest width across nasal capsules, one facet and one condyle in the puboischiadic bar for articulating with the basipterygium, two intermediate segments between the basipterygium of the pelvic fin and the axial cartilage of the claspers, five terminal cartilages of the claspers, and posterior medial process of the puboischiadic bar absent [22]. Described the fleshy core of the nostrils in *Cirrhigaleus* but it did not take into account in his analysis. The pelvic articular region described here for the genus is proposed for the first time and it is in contrast to the recent description of [66] observed for *C. barbifer*. According to these authors, the puboischiadic bar articulates with the basipterygium through a single enlarged and reniform facet located on each postero-lateral portion of the pelvic girdle and medially to the position of the condyle for articulation with the anterior pelvic element (*sensu* first enlarged radial in [66]). *Cirrhigaleus* species lack the posterior medial process of the puboischiadic bar (*sensu* posteromedian projection) as also noticed in [66]. Characters of the claspers (e.g., number of intermediate elements; number of terminal cartilages) are cladistically analyzed herein for the first time and should be taken into account in future phylogenetic analysis within Squaliformes. Descriptive anatomy of the claspers has been provided to discuss the interrelationships in sharks and batoids as seen in [68, 69] and more recently [70, 71]. [27, 47] treated clasper morphology within Squaliformes but since then no additional examination has been conducted, except for general species descriptions such as for *Isistius* in [35, 52].

The monophyly of *Squalus* is supported by rostral appendages present in the neurocranium, presence of paired supraethmoidal processes in the neurocranium, antorbital cartilage developed and expanded posteriorly, nasal lobes supported by cartilage, and presence of precaudal pits. With exception to the presence of precaudal pits proposed in [22], the characters raised here are all novel synapomorphies of the genus. Presence of supraethmoidal process in the ethmoidal region of the neurocranium is synapomorphic for the genus and thus in congruence with [22]. Plasticity of supraethmoidal processes in *C. asper*, however, have resulted it as uninformative character for *Cirrhigaleus*.

Furthermore, novel insights into interrelationships within *Squalus* are highlighted in the current analysis especially regarding the 'validity' of group/complex of species. In the resulting most-parsimonious morphological tree, all three groups of species of *Squalus* are monophyletic. *Squalus acanthias* group (Clade C) consisting of *S. acanthias* and S. *suckleyi* is sister-group of all other *Squalus* members. *Squalus megalops* group (Clade E), herein comprising of S. *albifrons*, *S. megalops* and *S. brevirostris*, is sister-group of the *S. mitsukurii* group (Clade G) consisting of *S. mitsukurii*, *S. montalbani* and *S. japonicus*. The topology provided here represents the first hypothesis of morphological interrelationships within the genus and it is partially congruent with the most recent molecular phylogenetic hypothesis (e.g. [15, 72, 73]) in supporting three separate clades within *Squalus* (Fig 14). Our analysis further provides a better resolution with regards to the validity of the complex/group of species as the inner and interrelationships between the *S. mitsukurii* and *S. megalops* groups have been thought to be unclear and unresolved. With exception to the *S. acanthias* group (Clade C) and a smaller clade consisting of *S. megalops* and *S. brevirostris* (Clade F) that exhibit Bremer support and GC value above 30%, other clades and phylogenetic arrangements between *Squalus* species reveal weak support possibly due to operational issues as only a few *Squalus* species were included in the analysis for outgroup inference. These results, however, are tentative as the focus of the current study is on *Cirrhigaleus*. Future investigations on the phylogenetic interrelationships within the genus *Squalus* may reveal a different scenario by increasing the number of characters and adding other terminal taxa to the analysis. Whether the validity of the complexes/groups of species is again supported the taxonomic classification for the whole genus must be revised. Since *S. acanthias* is the type-species of the genus, the clade comprising *S. acanthias* and *S. suckleyi* could retain the designation *Squalus*, and the remaining two clades should be given a new generic designation. *Flakeus* Whitley, 1939 [74] and *Koinga* Whitley, 1939 [74] are generic names currently in synonymy with *Squalus* and may be resurrected.

## Generic placement of C. asper

Phylogenetic analysis of morphological characters presented here supports that the nominal species *C. asper* is correctly assigned to the genus *Cirrhigaleus* and that this species is sister-group to a small clade consisting of *C. barbifer* and *C. australis*. These results are congruent to the evidence presented earlier in [3, 6, 22] that supported to be congeneric with other *Cirrhigaleus* species but against [2, 8, 15] on suggesting its generic allocation to *Squalus*. *Cirrhigaleus barbifer* and *C. australis* share a condition of the dermal denticles that is the presence of cusplets at the posterior margin of the crown which is absent in *C. asper*. This species thus has shown to share many meristics, inner and external morphological characteristics with *Cirrhigaleus* congeners that was not previously known including higher number of monospondylous vertebrae, medial nasal lobe (nasal barbels) supported by fleshy core and innervated by the buccopharyngeal branch of the facial nerve, neurocranium with greatest width across nasal capsules, pelvic articular region comprised by one facet and one condyle for articulating with the basipterygium, two intermediate segments between the basipterygium and the axial cartilage, and five terminal cartilages of the claspers.

External morphology, meristics and morphometric data of *C. asper* are much more similar to its congeners. It shares with *C. barbifer* and *C. australis* many characteristics, such as: body trihedral and markedly robust; dermal denticles tricuspid and conspicuously wide with cusplets; teeth unicuspid with apron markedly broad and cusp somewhat upright; first and second dorsal fins vertical and upright, equally tall; first and second dorsal-fin spines almost equal in length (second dorsal-fin spine often worn down but not broken); origin of second dorsal fin placed over a vertical line traced at pelvic-fin free rear tips; upper labial furrow markedly short with thick fold.

Morphological differences are more apparent between species of *Cirrhigaleus* and *Squalus* related to, for instance, upper labial furrow small with thick fold (*vs*. upper labial furrow large and thin in *Squalus*), spiracles above the eyes (*vs*. eyes placed laterally behind the eyes in *Squalus*), dermal denticles with cusplets at posterior margin of the crown (*vs*. cusplets absent in *Squalus)*, and second dorsal fin with its origin over pelvic free rear tips (*vs*. origin of second dorsal fin far behind pelvic free rear tips in *Squalus*) [2, 8]. Noticed morphological differences between *C. asper* and species of *Squalus*, that are also observed in this study, including: size of dermal denticles (three times larger in *C. asper* than in species of Squalidae); position of origin of first dorsal fin (behind free rear tips of pectoral fins in *C. asper vs*. prior or over free rear tips of pectoral fins in *Squalus*, except for *S. acanthias*); origin of pelvic fins (just prior origin of second dorsal fin in *C. asper vs*. conspicuously prior to second dorsal fin in *Squalus*); length of caudal peduncle (shorter in *C. asper* than in *Squalus*), and precaudal pit (absent in *C. asper vs*. present in *Squalus*). Similarities between *Cirrhigaleus* and *Squalus* regarding dentition, length of dorsal-fin spines, shape of dermal denticles, absence of nasal barbels, presence of precaudal keel, vertebral counts and morphology of terminal cartilages of the claspers led the misleading generic placement of *C. asper* and synonymy of *Cirrhigaleus* in [2, 8] as many of the characteristics pointed out in these studies can be applied to other species of *Cirrhigaleus* as well as some members of Squaliformes (e.g. *Centrophorus*).

In particular, presence or absence of nasal barbels has been applied as diagnostic character to separate species of *Squalus* and *Cirrhigaleus*. *Cirrhigaleus asper* and species of *Squalus* bear anterior margin of nostrils conspicuously short and thus the former species have been misidentified as *Squalus*. Nostrils internal anatomy is not homologous within Squalidae as so a precautionary approach has to be conducted to diagnose species. Nasal barbels are here defined as in [22] as an extension of the mesial nasal lobe that is supported internally by fleshy core and innervated by a buccopharyngeal branch of facial nerve (VII). Nasal barbels differ in length within *Cirrhigaleus* as *C. barbifer* and *C. australis* exhibit conspicuously elongated and moustache-like nasal barbels whereas *C. asper* shows short and non moustache-like barbels. Species of *Squalus* do not bear nasal barbels because the mesial nasal lobe is supported internally by a thin nasal cartilage with not associated innervations. Additionally, lateral nasal lobe is often slightly larger than medial lobe in species of *Squalus* as stated previously in [4] while *Cirrhigaleus* species have lateral nasal lobe much shorter than medial one, including *C. asper*.

**Taxonomic account.** Family Squalidae Blainville, 1816 [75]

Genus *Cirrhigaleus* Tanaka, 1912 [1]

*Cirrhigaleus* Tanaka, 1912: 151–154, 163; pl. XLI, Figs 156–162 (original description, illustrated; Sagami Sea, Japan; type species by original designation and monotypy) [1]; Herre 1936: 59 (cited; Japan) [76]; Bigelow and Schroeder 1948: 451 (cited; Japan) [18]. Bigelow and Schroeder 1957: 18, 19, 24, 37–38 (cited, revision, description; Japan) [19]; Garrick and Paul 1971: 1–13 (revision, description, illustrated; Japan, New Zealand) [4]; Bass et al. 1976: 9, 10 (cited; Japan) [8]; Fourmanoir and Rivaton 1979: 436 (listed; Japan, New Zealand, Vanuatu) [5]; Compagno 1984: 61–62 (description, revision; Japan, New Zealand, Australia, Vanuatu) [14]; Shirai 1992: 1–125 (cited, listed, described; Japan, South Africa) [22]; Compagno and

Niem 1998 (in part): 1203–1224 (listed, cited; Japan, New Zealand, Vanuatu, Australia) [77]; Compagno 1999: 472 (listed; West Indian, Central Pacific and North Atlantic Oceans) [78]; Yuanding and Qingwen 2001: 292–293 (listed; Northwest Pacific Ocean) [79]; Nakabo 2002: 155 (listed; Southern Japan, Ryukyu Islands, New Zealand) [80]; Compagno et al. 2005: 71–73 (revision; Pacific, Atlantic and Indian Oceans) [6]; Last et al. 2007: 1 (cited only) [24]; White et al. 2007: 19–30 (description; Japan, New Zealand, Australia, Taiwan, Vanuatu, Indonesia, Seychelles) [3]; Ebert 2013: 52–55 (cited, listed, revised; Japan, New Zealand, Australia, Hawaii, South Africa, Mozambique, Seychelles, St. Paul and Amsterdam Islands, Gulf of Mexico) [12]; Ebert et al. 2013: 74, 80–82 (cited, description; Pacific, Atlantic and Indian Oceans) [10]; Kempster et al. 2013: 1–4 (cited; Japan, Western and Southern Australia, Indonesia) [7]; Nakabo 2013: 194 (listed; Southern Japan, Ryukyu Islands, New Zealand) [81]; Ebert 2015: 54–56 (cited; South Africa) [11]; Duffy and Last 2015: 125–131 (description; New Zealand, Australia, Japan) [82]; Del Moral-Flores et al. 2015: 58 (listed, Seychelles, Mexico) [83].

*Squalus* (subgenus *Cirrhigaleus*): Garman 1913: 457 (description; Japan) [16]; Fowler 1941: 262 (description; Japan) [17].

*Phaenopogon* Herre, 1935: 121–124, Fig 1 (original description, illustrated; type species *P. barbulifer* by original designation and monotypy; unnecessary replacement name for *Cirrhigaleus*; Misaki Bay, Japan) [84]; Herre 1936: 59 (cited as synonym of *Cirrhigaleus*; Japan) [76].

*Squalus*: Fowler 1936: 69 (cited; Japan) [85]; Bass et al. 1976: 8–20 (revision, description; Japan, New Zealand, South Africa, Mozambique) [8]; Compagno 1984 (in part): 110, 114 (cited, description; Hawaiian Islands, Gulf of Mexico) [14].

*Type species*: *Cirrhigaleus barbifer* Tanaka, 1912 [1] by original designation and monotypy.

*Diagnosis*. A genus of the family Squalidae differing from *Squalus* by bearing nasal barbel in the anterior margin of nostrils that it is innervated by a branch buccopharyngeal of the facial nerve (VII) and internally supported by fleshy core. *Cirrhigaleus* is distinct from *Squalus* on lacking precaudal pits (*vs*. present in *Squalus*), spiracles placed dorsally above the eyes (*vs*. spiracles placed laterally behind the eyes), eyes with both anterior and posterior margins notched (*vs*. eyes with anterior margin concave and posterior margin notched), upper labial furrow very short with thick fold (*vs*. upper labial furrow large with thin fold), and origin of second dorsal fin over or just prior a vertical line traced at pelvic-fin free rear tips (*vs*. origin of second dorsal fin far behind a vertical line traced at pelvic-fin free rear tips). *Cirrhigaleus* also differs from *Squalus* by having dorsal-fin spines markedly elongate, transcending greatly dorsal-fin apexes (*vs*. dorsal-fin spine short, rarely transcending dorsal-fin apexes). Caudal fin of *Cirrhigaleus* exhibits continuous transition between upper and lower lobes at level of caudal fork, a condition distinct from *Squalus* (*vs*. discontinuous transition between upper and lower lobes). *Cirrhigaleus* body trihedral in cross-section, conspicuously arched dorsally and humped at belly, tail fusiform and short dorsal-caudal space (*vs*. body entirely fusiform from head to tail, inconspicuously arched dorsal and ventrally at belly, elongate dorsal-caudal space in *Squalus*). *Cirrhigaleus* species also show higher values of monospondylous vertebrae counts than *Squalus* species (47–53 for *Cirrhigaleus vs*. 37–50 for *Squalus*).

*Geographical distribution*. *Cirrhigaleus* occurs from the North to Southwestern Pacific Oceans, including waters off Japan, Indonesia, New Zealand and New Caledonia, and in the Western Indian Ocean from the Seychelles to South Africa and St. Paul Islands, as well as in the North and South Western Atlantic Oceans from the USA to Brazil (Fig 15).

*Remarks*. Three valid species of barbel-bearing dogfishes of the genus *Cirrhigaleus* are here recognized: *C. barbifer*, *C. asper* and *C. australis*. They comprise benthic species that are often found on or near bottom of the outer continental shelves and mid-continental slopes in depths between 91–1103 meters. According to [6, 12], species are rarely caught off bays and river mouths.

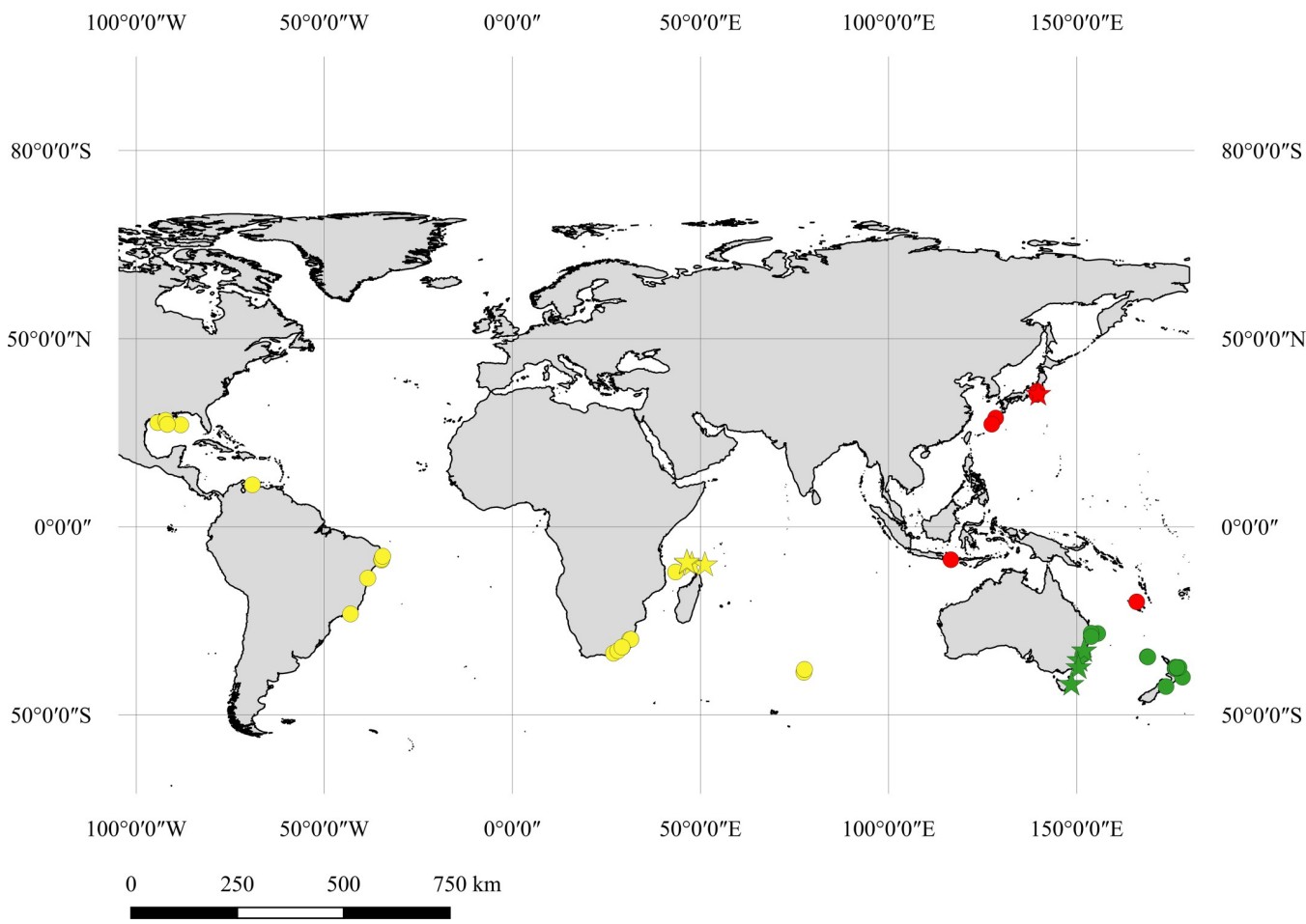

**Fig 15. Map of the geographical distribution of *Cirrhigaleus* species.** Stars: type material; dots: other material; red symbols: *C. barbifer*; green symbols: *C. australis*; yellow symbols: *C. asper*.

## *Cirrhigaleus barbifer* Tanaka, 1912 [1]

Mandarin dogfish; Hige-zune (Japanese)

*Cirrhigaleus barbifer* Tanaka, 1912: 151–154; pl. XLI, Figs 156–162 (original description, illustrated; holotype by monotypy, ZUMT 3397 (lost), adult male, 855 mm TL, collected at Tokyo Fish Market, Japan; type locality Sagami Sea, Japan) [1]; Bigelow and Schroeder 1948: 451 (cited; Japan) [18]; Bigelow and Schroeder 1957: 17–19, 24, 37–38 (cited, revision, description; Japan) [19]; Garrick and Paul 1971 (in part): 1–13 (revision, description, illustrated; Japan) [4]; Compagno 1984: 61–62 (description, illustrated; Japan, Vanuatu) [14]; Shirai 1992: 1–125 (cited, listed, described; Japan) [22]; Compagno and Niem 1998 (in part): 1203–1224 (listed, cited, illustrated; Japan, Vanuatu) [77]; Compagno 1999: 472 (listed) [78]; Yuanding and Qingwen 2001: 292–293 (listed; Japan) [79]; Nakabo 2002: 155 (listed; Southern Japan, Ryukyu Islands) [80]; Compagno et al. 2005 (in part): 73 (description; Japan) [6]; White et al. 2006: 66, 319 (cited, listed, illustrated; Japan, Indonesia, Vanuatu) [86]; White et al. 2007: 19–30 (description; Japan, Taiwan, Vanuatu, Indonesia) [3]; Ebert et al. 2013: 74–82 (cited, description; Western Pacific Ocean) [10]; Ebert et al. 2013: 284–285 (listed; Taiwan, Indonesia, Japan) [87]; Kempster et al. 2013: 1–4 (cited; Japan, Western and Southern Australia, Indonesia) [7]; Nakabo 2013: 194 (listed; Southern Japan, Ryukyu Islands) [81]; Weigmann 2016: 902

(listed; North-western Pacific, Eastern Indian Oceans) [88]; Miyazaki et al. 2019: 121–122, Fig 2L (listed; Sagami Sea) [89].

*Squalus barbifer* Garman, 1913: 457 (description; Japan) [16]; Fowler 1941: 262 (description; Japan) [17]; Bass et al. 1976: 9, 10 (cited; Japan) [8]; Fourmanoir and Rivaton 1979: 436 (listed; Japan, New Zealand, Vanuatu) [5].

*Phaenopogon barbulifer* Herre, 1935: 121–124, Fig 1 (original description, illustrated; holotype by original designation and monotypy, SU 13901, adult female, 730 mm TL, collected in Misaki Bay, Japan by A. Owston [84]; Herre 1936: 59 (cited as synonym of *C. barbifer*; Japan) [76].

*Neotype designation*. HUMZ 197852, adult female, 870 mm TL, off Kanaya, Tokyo Bay, Japan, 200 m depth, collected by Kenta Suda on 01 December 2006.

*Type locality*: off Kanaya, Gulf of Tokyo, Japan.

*Other material examined (8 specimens)*: CSIRO H 5875–09, adult female, 978 mm TL, Tanjung Luar, Indonesia, 08˚45'S,116˚35'E; HUMZ 95177, juvenile female, 584 mm TL, East China Sea, 28˚54.2'N,128˚29.3'E; HUMZ 101533, juvenile male, 650 mm TL, Okinawa, Japan, 27˚16.2'N,127˚27'E; HUMZ 231872 (photo only), juvenile female, 899mm TL, Pacific coast of Northern Japan, 320–350m depth; MNHN 1997–3568, adult female, 800 mm TL, New Caledonia; NMW 98257, adult female, 960 mm TL, precise locality unknown, Asia; SU 13901 (holotype of *Phaenopogon barbulifer* Herre, 1935 [84]), juvenile female, 730 mm TL, Misaki Bay, Japan, 35.138197˚N,139.617375˚E; SU 14171, adult female, 855 mm TL, Sagami Sea, Japan.

*Diagnosis*. *Cirrhigaleus barbifer* is distinguished from its congeners by having body dark grey dorsally (*vs*. light grey for *C. australis vs*. brown for *C. asper*), and teeth with apron conspicuously broad (*vs*. narrower in *C. australis* and *C. asper*). It is easily distinct from *C. asper* by having nasal barbels moustache-like and well elongate (5.4%–6.4%TL), often reaching the mouth (*vs*. nasal barbel non moustache-like and markedly short, its length 1.0%–1.5% TL, slightly transcending posterior margin of nostrils in *C. asper*), and dermal denticles bat-like (*vs*. heart-shaped) with median ridge very narrow (*vs*. broad) and lateral cusps conspicuous (*vs*. inconspicuous). *Cirrhigaleus barbifer* is separated from *C. australis* by having more elongate nasal barbels (5.4%–6.4%TL *vs*. 4.4%–6.2%TL, respectively) and dermal denticles with a two cusplets on each side (*vs*. one cusplet in *C. australis*) (Fig 16).

*Description*. Morphometric and meristic data are shown in Table 1 and S3 and S4 Tables. Single value is for neotype and ranges values are for other material.

*External morphology*. Body conspicuously robust and trihedral, markedly humped dorsal and ventrally throughout all its extension, turning slender caudally from pelvic fin insertion to caudal fin origin (Fig 16); body with its greatest depth at abdomen, its height 11.8%, 8.6%–17.1% TL); head height 1.1, 0.8–1.0 times trunk height and 0.9, 0.8–0.9 times abdomen height; body conspicuously broad from head to abdomen with its greatest width at head, corresponding to 1.0, 1.0–1.2 times trunk width and 1.5, 1.1–1.5 times abdomen width. Head small, its length 20.7% (18.8%–23.0%) TL, depressed and markedly convex dorsally between eyes and spiracles; head very narrow anteriorly, its width at nostrils 6.7% (6.4%–7.9%) TL and its width at mouth 12.7% (11.0%–13.2%) TL. Snout rounded at tip and conspicuously short (preorbital length 6.4%, 5.5%–6.8% TL); distance from nostrils to snout tip 0.9 (0.8–1.0) times its distance to upper labial furrow; internarial space 0.9 (0.7–1.4) times eye length, and 0.6 (0.5–0.6) times preoral length; anterior margin of nostrils bilobed with lateral lobe broad and large, although much smaller than medial lobe; medial lobe of nostrils conspicuously elongate and moustache-like as nasal barbels, often reaching anterior margin of mouth; anterior nasal flap length 5.4%, 5.4%–6.4% TL or fitting 3.8 (3.0–4.5) times in head length. Eyes oval and elongate, its length 2.0 (2.1–5.2) times greater than its height, and 0.8 (0.6–1.1) times prenarial length; eyes conspicuously concave on its anterior and lateral margins; posterior margin of eyes slightly

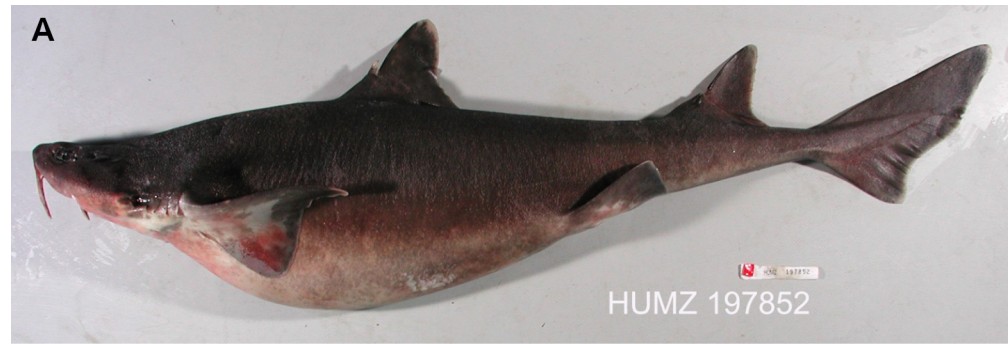

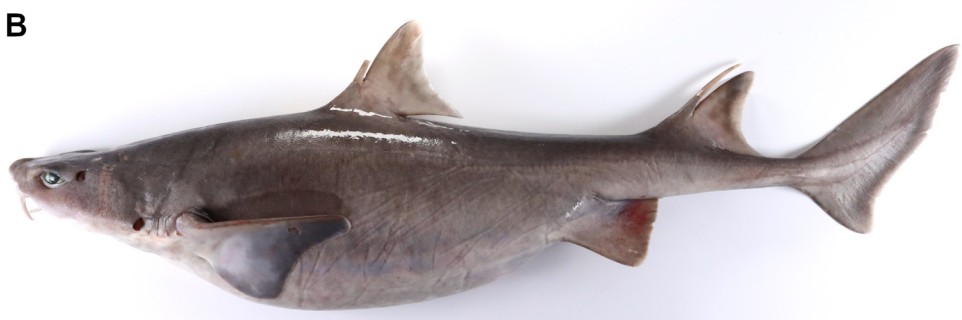

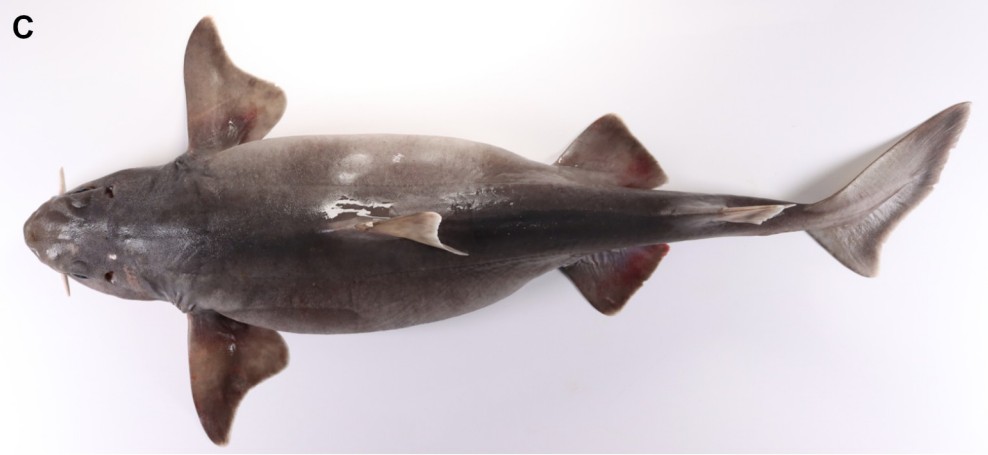

**Fig 16.** *Cirrhigaleus barbifer* in lateral (A,B) and dorsal (C) views. A: HUMZ 197852 (neotype of *C. barbifer*), adult female, 870 mm TL; B,C: HUMZ 231872, adult female, 899 mm TL (photo credits: Fisheries Science Center, The Hokkaido University Museum).

notched. Prespiracular length 0.5 times prepectoral length, and 1.6 (1.6–1.8) times preorbital length. Spiracles crescent and broad, its length 0.4 (0.3–0.4) times eye length, located lateral-posteriorly and above eyes. Prebranchial length 1.6 (1.5–1.7) times greater than prespiracular length. Gill slits vertical, somewhat concave, and very tall, with fifth gill slit height corresponding to 1.2 (1.1–1.5) times first gill slit height.

Preoral length 1.0 (0.9–1.1) times mouth width. Mouth conspicuously arched and broad, its width 2.3 (2.2–2.4) times internarial space and 1.9 (1.9–2.1) times prenarial length. Upper labial furrow markedly small, its length 1.1% (1.1%–1.5%) TL with fold very short and broad; lower labial furrow elongate, lacking fold. Teeth similar in both jaws, unicuspid, flattened labial-lingually and alternate (Fig 17); teeth markedly broad and low at crown with upper teeth

**Table 1. External measurements of *C. barbifer* expressed as percentage of total length (%TL).**

| | *Cirrhigaleus barbifer* | | | | | | | | | |
|---|---|---|---|---|---|---|---|---|---|---|
| | Japan | | | | | NC/Indonesia | | | | *x* | SD |
| | Neotype | N | | | | N | | | | | |
| TL (mm) | 870.0 | 3 | 650.0 | – | 960.0 | 2 | 800.0 | – | 978.0 | 831.3 | 129.4 |
| PCL | 79.9 | 3 | 76.7 | – | 79.2 | 2 | 78.9 | – | 84.4 | 79.7 | 2.5 |
| PD2 | 63.6 | 3 | 60.8 | – | 62.0 | 2 | 63.4 | – | 66.5 | 62.9 | 2.1 |
| PD1 | 33.3 | 3 | 31.8 | – | 32.9 | 2 | 31.9 | – | 35.3 | 32.9 | 1.3 |
| SVL | 54.4 | 3 | 52.5 | – | 56.3 | 2 | 54.8 | – | 55.7 | 54.8 | 1.3 |
| PP2 | 50.6 | 3 | 49.2 | – | 53.6 | 2 | 50.6 | – | 52.4 | 51.2 | 1.6 |
| PP1 | 20.5 | 3 | 18.1 | – | 21.0 | 2 | 20.4 | – | 20.9 | 19.9 | 1.3 |
| HDL | 20.7 | 3 | 18.8 | – | 19.5 | 2 | 23.0 | – | 23.0 | 20.7 | 1.9 |
| PG1 | 16.8 | 3 | 14.8 | – | 16.2 | 2 | 18.9 | – | 19.6 | 17.1 | 1.8 |
| PSP | 10.5 | 3 | 9.5 | – | 9.8 | 2 | 10.3 | – | 11.3 | 10.2 | 0.7 |
| POB | 6.4 | 3 | 5.5 | – | 6.2 | 2 | 6.3 | – | 6.8 | 6.1 | 0.5 |
| PRN | 4.3 | 3 | 3.8 | – | 4.3 | 2 | 4.2 | – | 4.2 | 4.1 | 0.2 |
| POR | 7.8 | 3 | 7.1 | – | 8.7 | 2 | 8.1 | – | 8.2 | 7.9 | 0.6 |
| INLF | 4.9 | 3 | 3.7 | – | 4.3 | 2 | 4.7 | – | 5.3 | 4.5 | 0.6 |
| MOW | 8.0 | 3 | 7.2 | – | 8.1 | 2 | 7.9 | – | 9.0 | 8.0 | 0.6 |
| ULA | 1.1 | 3 | 1.1 | – | 1.1 | 2 | 1.1 | – | 1.5 | 1.2 | 0.2 |
| INW | 3.4 | 3 | 3.0 | – | 3.8 | 2 | 3.6 | – | 3.8 | 3.5 | 0.3 |
| INO | 8.1 | 3 | 7.6 | – | 8.1 | 2 | 8.1 | – | 8.7 | 8.1 | 0.4 |
| EYL | 3.6 | 3 | 2.6 | – | 4.3 | 2 | 3.6 | – | 4.3 | 3.7 | 0.6 |
| EYH | 1.8 | 3 | 0.5 | – | 1.9 | 2 | 0.9 | – | 1.1 | 1.2 | 0.5 |
| SPL | 1.5 | 3 | 1.2 | – | 1.7 | 2 | 1.3 | – | 1.6 | 1.4 | 0.2 |
| GS1 | 2.0 | 3 | 1.8 | – | 2.2 | 2 | 2.2 | – | 2.2 | 2.1 | 0.2 |
| GS5 | 2.3 | 3 | 2.0 | – | 3.1 | 2 | 2.6 | – | 2.7 | 2.5 | 0.4 |
| IDS | 22.4 | 3 | 21.9 | – | 22.6 | 2 | 23.1 | – | 24.5 | 22.7 | 1.0 |
| DCS | 7.5 | 3 | 7.6 | – | 8.9 | 2 | 7.7 | – | 9.1 | 8.1 | 0.7 |
| PPS | 28.2 | 3 | 26.2 | – | 31.8 | 2 | 26.9 | – | 30.0 | 28.4 | 2.1 |
| PCA | 18.4 | 3 | 20.0 | – | 21.9 | 2 | 21.0 | – | 21.3 | 20.5 | 1.2 |
| D1L | 14.0 | 3 | 13.9 | – | 14.8 | 2 | 14.6 | – | 15.1 | 14.5 | 0.5 |
| D1A | 13.5 | 3 | 13.5 | – | 14.3 | 2 | 12.7 | – | 12.9 | 13.4 | 0.6 |
| D1B | 8.3 | 3 | 8.3 | – | 8.9 | 2 | 8.3 | – | 9.7 | 8.7 | 0.6 |
| D1H | 10.0 | 3 | 8.9 | – | 10.8 | 2 | 10.1 | – | 10.1 | 10.0 | 0.6 |
| D1I | 5.8 | 3 | 5.4 | – | 6.3 | 2 | 5.7 | – | 6.3 | 5.9 | 0.4 |
| D1P | 10.6 | 3 | 9.0 | – | 10.9 | 2 | 12.1 | – | 12.2 | 10.7 | 1.3 |
| D1ES | - | 2 | 5.2 | – | 6.0 | 2 | 5.3 | – | 8.3 | 6.2 | 1.4 |
| D1BS | 0.9 | 3 | 0.8 | – | 1.0 | 2 | 0.8 | – | 0.9 | 0.9 | 0.1 |
| D2L | 13.9 | 3 | 14.0 | – | 15.0 | 2 | 14.5 | – | 14.7 | 14.5 | 0.4 |
| D2A | 13.3 | 3 | 13.7 | – | 14.9 | 2 | 13.3 | – | 13.5 | 13.9 | 0.7 |
| D2B | 8.9 | 3 | 8.5 | – | 9.7 | 2 | 8.8 | – | 9.9 | 9.2 | 0.6 |
| D2H | 9.3 | 3 | 8.8 | – | 10.3 | 2 | 10.0 | – | 10.5 | 9.8 | 0.6 |
| D2I | 5.0 | 3 | 4.8 | – | 5.9 | 2 | 4.9 | – | 5.6 | 5.3 | 0.5 |
| D2P | 10.5 | 3 | 7.3 | – | 8.7 | 2 | 8.7 | – | 10.5 | 8.9 | 1.3 |
| D2ES | – | 3 | 7.1 | – | 7.6 | 2 | 7.9 | – | 8.0 | 7.6 | 0.4 |
| D2BS | – | 3 | 0.8 | – | 1.2 | 2 | 0.9 | – | 1.2 | 1.0 | 0.2 |
| P1A | 15.5 | 3 | 15.3 | – | 17.1 | 2 | 15.8 | – | 16.6 | 16.2 | 0.8 |
| P1I | 8.9 | 3 | 8.3 | – | 8.5 | 2 | 9.4 | – | 9.5 | 8.8 | 0.5 |

(*Continued*)

**Table 1.** (Continued)

| | Japan | | | | NC/Indonesia | | | $x$ | SD |
|---|---|---|---|---|---|---|---|---|---|
| | Neotype | N | | | N | | | | |
| P1B | 4.6 | 3 | 4.3 | – 5.4 | 2 | 4.5 | – 5.0 | 4.8 | 0.4 |
| P1P | 11.4 | 3 | 11.3 | – 13.1 | 2 | 11.8 | – 12.7 | 12.0 | 0.8 |
| P2L | 13.2 | 3 | 12.2 | – 13.3 | 2 | 13.5 | – 14.4 | 13.3 | 0.7 |
| P2I | 5.4 | 3 | 5.6 | – 6.2 | 2 | 6.5 | – 6.9 | 6.1 | 0.6 |
| CDM | 20.5 | 3 | 20.5 | – 23.3 | 2 | 15.0 | – 20.7 | 20.1 | 2.7 |
| CPV | 11.6 | 3 | 9.0 | – 11.0 | 2 | 10.3 | – 11.7 | 10.7 | 1.0 |
| CFW | 8.4 | 3 | 7.0 | – 8.2 | 2 | 5.5 | – 8.2 | 7.5 | 1.1 |
| HANW | 6.7 | 3 | 6.4 | – 7.6 | 2 | 7.8 | – 7.9 | 7.3 | 0.6 |
| HAMW | 12.7 | 3 | 11.0 | – 12.7 | 2 | 12.5 | – 13.2 | 12.3 | 0.9 |
| HDW | 15.7 | 3 | 13.7 | – 26.7 | 2 | 16.5 | – 16.9 | 17.3 | 4.7 |
| TRW | 14.9 | 3 | 11.2 | – 27.4 | 2 | 13.9 | – 17.2 | 16.3 | 5.8 |
| ABW | 10.3 | 3 | 9.3 | – 20.5 | 2 | 14.0 | – 14.1 | 13.5 | 3.9 |
| HDH | 10.7 | 3 | 7.2 | – 13.0 | 2 | 12.9 | – 13.1 | 11.6 | 2.3 |
| TRH | 10.1 | 3 | 7.6 | – 16.4 | 2 | 13.8 | – 14.9 | 12.7 | 3.3 |
| ABH | 11.8 | 3 | 8.6 | – 17.1 | 2 | 15.6 | – 15.7 | 13.8 | 3.1 |
| CLO | – | 1 | 1.7 | – 1.7 | | – | | 1.7 | – |
| CLI | – | 1 | 3.5 | – 3.5 | | – | | 3.5 | – |
| ANFL | 5.4 | 3 | 5.4 | – 6.4 | 2 | 5.9 | – 6.4 | 5.9 | 0.5 |

Range values for other material are provided by region for comparisons. TL is expressed in millimeter. n: number of specimens; *x*: mean; SD: standard deviation. NC: New Caledonia.

smaller than lower teeth; teeth with cusp short and somewhat cylindrical, oblique and directed laterally; mesial cutting edge slightly concave on upper jaw and straight on lower jaw; mesial heel conspicuously tapered and pointed; distal heel rounded; apron short and markedly broad;

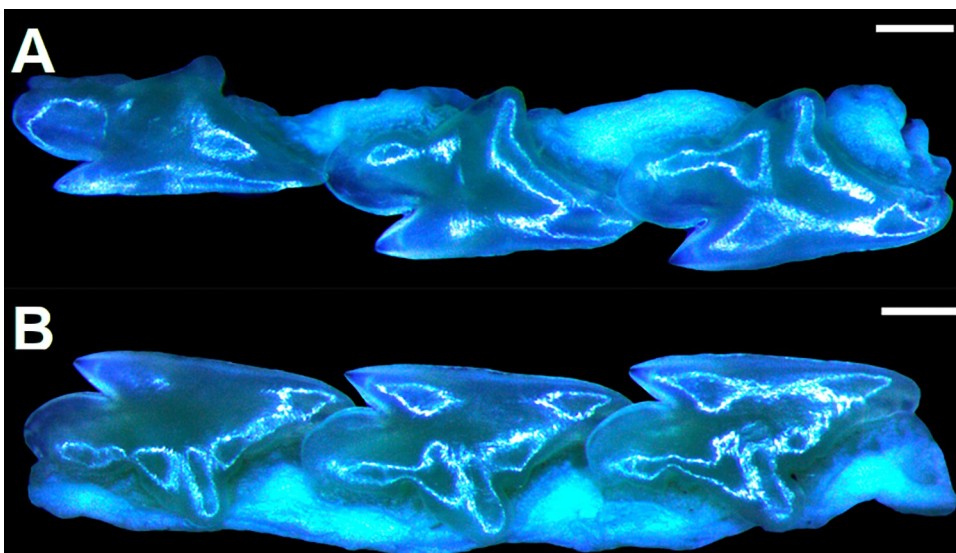

**Fig 17.** Upper (A) and lower (B) teeth of *C. barbifer*, NMW 98257, adult female, 960 mm TL (labial view). Scale bars: 1 mm.

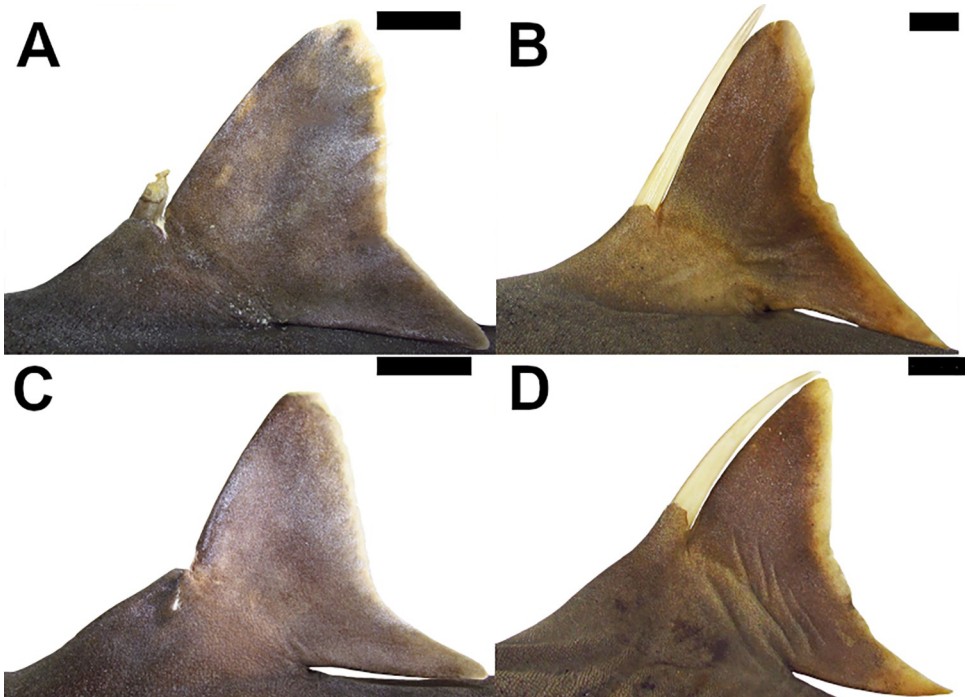

**Fig 18.** First (a, b) and second (c, d) dorsal fins of *C. barbifer*, showing intraspecific regional variations. a,c: HUMZ 197852 (neotype), adult female, 870 mm TL, Japan; b,d: CSIRO H5875-09, adult female, 978 mm TL, Indonesia. Scale bars: 20 mm.

apron narrower on lower jaw than upper jaw. Intermediate tooth on both jaws in some specimens, usually narrower at crown than the following teeth; Intermediate tooth with cusp vertical, mesial cutting edge convex, both mesial and labial heels rather pointed. Three series of functional teeth on both jaws for neotype; upper jaw with 15–0–14 (14–1–14) teeth rows; lower jaw with 11–0–10 (11–1–11) teeth rows.

Dorsal fins conspicuously upright and vertical, both equally large with length of first dorsal fin 1.0 (0.9–1.1) times second dorsal fin length (Fig 18); dorsal fins equally tall, with first dorsal fin height 1.1 (1.0) times second dorsal fin height. Pre-first dorsal length 1.6 (1.6–1.8) times prepectoral length. First dorsal fin markedly slender and rounded at the apex, broad at its base (base length 8.3%, 8.3%–9.7% TL); first dorsal-fin anterior margin concave, and posterior margin straight but concave near its free rear tip; first dorsal-fin inner margin short, its length 5.8% (5.4%–6.3%) TL; first dorsal fin height 1.7 (1.6–1.8) times greater than first dorsal-fin inner margin length. First dorsal-fin spine with its origin behind the vertical line traced at free rear tips of the pectoral fins in adults; first dorsal-fin spine concave, wide, and elongate, its length 5.2%–8.3% TL; first dorsal-fin spine not reaching the fin apex in neotype, its length corresponding to 0.5–0.8 times first dorsal-fin height. Interdorsal space 1.1 (1.1–1.2) times prepectoral length and 3.0 (2.5–3.0) times dorsal-caudal space. Pre-second dorsal length much greater than dorsal-caudal distance (63.6%, 60.8%–66.5% TL *vs.* 7.5%, 7.6%–9.1% TL, respectively). Pre-second dorsal length 3.1 (2.9–3.4) times prepectoral length. Origin of second dorsal fin prior to a vertical line traced at pelvic-fin free rear tips. Second dorsal fin slender, rounded and lobe-like at apex, broad at its base (base length 8.9%, 8.5%–9.9% TL); second dorsal-fin anterior margin concave; second dorsal-fin posterior margin straight, although concave on its lower half (slightly falcate in juveniles); second dorsal-fin height 1.9 (1.7–2.1) times greater than second dorsal-fin inner margin length; second dorsal-fin inner margin short, its length

5.0% (4.8%–5.9%) TL. Second dorsal-fin spine directed posteriorly, narrow and elongate, its length 7.1%–8.0% TL, corresponding to 0.7–0.8 times second dorsal fin height; second dorsal-fin spine length 1.0–1.5 times length of first dorsal-fin spine often transcending its second dorsal-fin apex (rarely in juveniles).

Pectoral fins conspicuously broad, although narrow anteriorly at its base (base length 4.6%, 4.3%–5.4% TL) and broad posteriorly (pectoral-fin posterior margin length 11.4%, 11.3%–13.1% TL); pectoral-fin anterior and inner margins strongly convex, and pectoral-fin posterior margin slightly concave (Fig 19A); pectoral-fin apex and free rear tips broadly rounded, although not lobe-like; pectoral-fin apex not transcending a horizontal line traced at pectoral-fin free rear tip; pectoral fin elongate with pectoral-fin anterior margin length 1.7 (1.7–2.0) times greater than inner margin length, and 1.4 (1.3–1.5) times pectoral-fin posterior margin length; pectoral-fin posterior margin length 1.1 (0.8–1.5) times trunk height. Pectoral-pelvic space 1.5 (1.3–1.5) times greater than pelvic-caudal space. Pelvic-caudal space 0.8 (0.9–1.0) times interdorsal space. Pelvic fins nearest second dorsal fin than first dorsal fin, with its free rear tips extending far behind a vertical line traced at origin of second dorsal fin; pelvic fin large, its length 1.7 (1.6–2.0) times in interdorsal space. Pelvic fins conspicuously broad, pentagonal in pair view with both pelvic-fin anterior and inner margins convex (Fig 19B); pelvic-fin apex broadly rounded; pelvic-fin free rear tips triangular, although rounded and lobe-like on males; pelvic fin length 1.1 (1.1–1.5) times length of preventral caudal margin. Claspers elongate and thick its outer length 1.7%–3.8% TL; clasper inner length 0.6 times pelvic-fin inner margin length; clasper groove markedly elongate and vertical, placed lateral-dorsally; apopyle and hypopyle very narrow, located at the opposite extremities of clasper groove; rhipidion flap-like, conspicuously tapered and elongate, placed laterally.

Caudal peduncle broad in cross-section and conspicuously short, dorsal-caudal space 7.5% (7.6%–9.1%) TL; lateral precaudal keel prominent laterally almost as a fold, since insertion of second dorsal fin to behind origin of caudal fin; upper and lower precaudal pits absent. Caudal fin (Fig 19C) with rectangular upper caudal lobe, posterior caudal tip rounded and broad; dorsal caudal margin straight; upper postventral caudal margin markedly concave, although straight proximally; lower postventral caudal margin convex and very short; dorsal-caudal margin elongate, its length 1.0 (0.7–1.2) times head length, and 1.8 (1.5–2.6) times larger than length of preventral caudal margin; preventral caudal margin straight and conspicuously small, its length 2.1 (1.5–1.9) times pelvic-fin inner margin length; ventral caudal tip rounded; caudal fork inconspicuous with transition between upper and lower caudal lobes somewhat continuous; caudal fork wide, its width 8.4% (5.5%–8.2%)TL.

*Dermal denticles*. Dermal denticles tricuspid and bat-like, somewhat imbricate, large and conspicuously broad at the crown (Fig 12A–12C); dermal denticles with its width as large as its length; cusps conspicuous, pointed and located posteriorly at posterior margin of the crown; median cusp much more elongate than lateral ones; median ridge elongate, thin, tall and straight, transcending anteriorly the crown base; lateral ridges slender, short, low and conspicuously convex proximally; median furrow narrow and shallow, placed anteriorly over median ridge; lateral furrows short and profound, located anteriorly aside each lateral ridge; two secondary cusps (or cusplets) often present on each side of posterior margin of the crown; cusplets much smaller than median cusp.

*Colouration*. In fresh, body brownish grey at the dorsal and at the upper lateral half, whitish ventrally (Fig 16). Nasal barbels white, sometimes greyish lateral and ventrally at its distal end. Dorsal fins grey to light grey with somewhat dark grey submarginal bar; dorsal-fin posterior margins broadly white from apex to its middle line; dorsal-fin free rear tips whitish. Dorsal-fin spines greyish brown, whitish at the tips. Pectoral fins brownish grey dorsally and whitish ventrally with large dark grey blotches throughout the edges; pectoral-fin posterior margin slightly

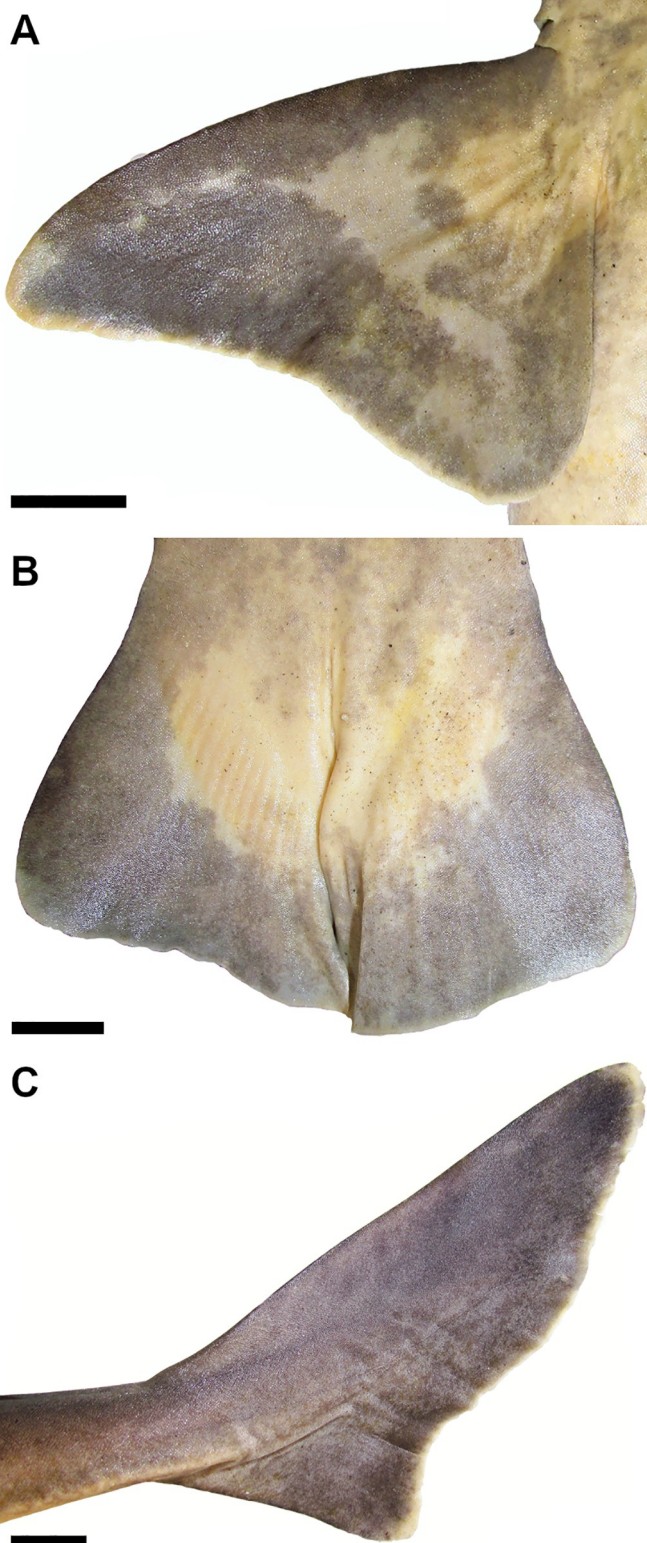

**Fig 19.** Pectoral (A), pelvic (B) and caudal (C) fins of the neotype of *C. barbifer*, HUMZ 197852, adult female, 870 mm TL. A,B: ventral view; C: lateral view. Scale bars: 20 mm.

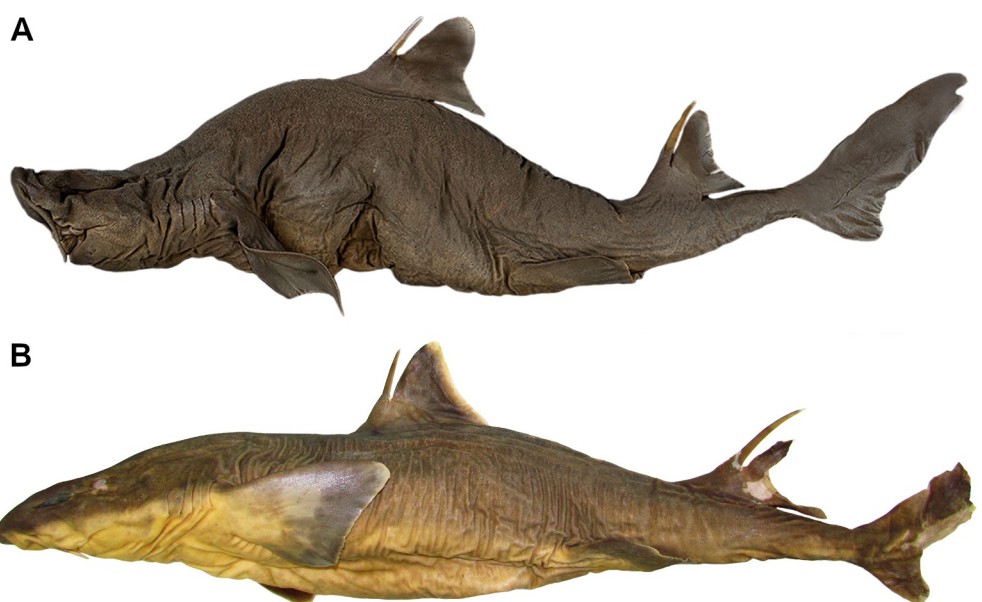

**Fig 20. Specimens of *C. barbifer* in lateral view.** A: SU 13901 (holotype of *Phaenopogon barbulifer* Herre, 1935), juvenile female, 730 mm TL, Japan; B: MNHN 1997–3568, adult female, 800 mm TL, New Caledonia.

white. Pelvic fins brownish grey dorsally and white ventrally with large light grey blotches throughout the edges; pelvic-fin posterior margin slightly white. Caudal fin brownish grey throughout all its extension; preventral caudal margin white, with blackish subterminal stripe; postventral caudal margins narrowly white with dark grey subterminal bar internally; black caudal stripe evident anteriorly. In preserved specimens (Fig 20), body dark grey to brownish dorsal and laterally, white ventrally. Nasal barbels white, greyish lateral and ventrally in its distal end. Dorsal fins grey, somewhat light grey posteriorly in the submarginal bar; dorsal-fin posterior margins broadly white from apex to its middle line; dorsal-fin free rear tips whitish. Dorsal-fin spines dark brown, whitish at the tips. Pectoral fins dark grey dorsally and whitish ventrally with large light grey blotches; pectoral-fin posterior margin uniformly white. Pelvic fins grey dorsally and white ventrally with large light grey blotches; pelvic-fin posterior margin white. Caudal fin dark grey throughout all its extension; postventral caudal margins narrowly white with blackish subterminal bar internally; black caudal stripe evident anteriorly.

*Vertebral counts.* Monospondylous vertebrae 50 for neotype (47–50 for other material); 88 (83–88) precaudal vertebrae; 115 (110–116) total vertebrae.

*Geographical distribution. Cirrhigaleus barbifer* occurs exclusively in the Pacific Ocean in waters from Japan, Indonesia, and New Caledonia as well as in Western Australia (Fig 15). It is caught between 200–760 meters depth.

**Remarks.** *Designation of the neotype of C. barbifer.* Tanaka (1912) [1] described both genus and species based in a single mature male collected at the Tokyo Fish Market in Japan that was designated originally as type and deposited under the catalogue number 3397 in the Science College Museum in Tokyo (currently ZUMT). It is possible that the holotype came from the fish collection of Mr. A. Owston because Dr. S. Tanaka used to work on specimens collected or donated by him as it is provided in [1, 90–93]. Some authors (e.g., [17, 19]) made references to the illustration of the holotype provided in [1] rather than to the specimen itself. According to Dr. K. Sakamoto (ZUMT), the holotype is lost. No other specimen of *C. barbifer* exists at the ZUMT fish collection, according to verifications achieved personally by the first author of this manuscript, suggesting that the holotype no longer exists. Interestingly, there is

no other specimen in the fish collections from Japan collected from the same locality as the type locality of *C. barbifer*, the Sagami Sea, provided in [1], which reject the possibility of the holotype to have been transferred to another collection in the past. A specimen of *C. barbifer* (SU 14171) that was also collected by Mr. A. Owston is available at CAS and it has same collecting data of those of the holotype of *C. barbifer* including collecting site and total length. However, this specimen is a mature female (not a mature male), according to D. Catania (CAS), which discards the possibility of being a missing holotype that ended up at CAS and received a new catalogue number by mistake on the accession and assignment of the specimen.

*Cirrhigaleus barbifer* has been regularly confused with *C. australis* in the South Pacific Ocean due to their morphological similarities regarding body size and shape, length of nasal barbels and dorsal-fin spines, and shape of dorsal and caudal fins as well as shape and size of dermal denticles. The characteristics provided in [1] for *C. barbifer* may also be applied to *C. australis*. [3] tentatively provided morphological diagnostic characters for separating these two nominal species but many of these (e.g., eye length, dorsal-caudal space, and vertebral counts) exhibited continuous range of values. These studies indicate clearly the difficulty to properly separate *C. barbifer* and *C. australis*. Moreover, instraspecific variations (ontogenetic and regional) observed here within *C. barbifer* reveal a more complex scenario behind the taxonomy of this nominal species because it obscures its morphological differentiation with congeners. Here, the neotype of *C. barbifer* (HUMZ 197852) is designated to clarify the taxonomic status of this species for better defining morphologically its local form and separating it effectively from *C. australis*. The neotype is held at Hokkaido University and it was collected off Kanaya, Gulf of Tokyo in Japan, which it is the locality nearest to the original type locality of *C. barbifer*. Following the requirements of International Code of Zoological Nomenclature (ICZN), diagnosis and description of this species are efficiently provided in our study, and the differentiation with closest related species (*C. australis*) are provided below based on characters of shape of dermal denticles, number of intermediate teeth, body colour, and length of nasal barbels.

*Synonymy of C. barbifer.* [84] described *Phaenopogon* Herre, 1935 [84] as a new genus and monotypic to include *P. barbulifer* Herre, 1935 [84]. This species was described based on a single specimen collected from Misaki Bay (Sagami Sea), Japan in 1906 by Mr. Alan Owston. This author did not take into account the previous study of [1] as he himself mentioned later in [76]. The characteristics provided in [84] are congruent with those observed for *C. barbifer*, including length and shape of nasal barbels, shape of dermal denticles, body colour, shape of fins and proportional external measurements. These observations indicate that *P. barbulifer* is junior synonym of *C. barbifer*, which is in agreement with [3, 14, 76].

*Intraspecific variations.* Origin of first dorsal fin is anterior or over a vertical line traced at pectoral-fin free rear tips in juveniles of *C. barbifer* while it is behind it in adults. First dorsal-fin spine length reaches up to half of first dorsal-fin height and never transcends first dorsal-fin apex in juveniles while first dorsal-fin spine length is over half of first dorsal-fin height and conspicuously transcends first-dorsal fin apex in adults. Second dorsal-fin spine is also smaller in juveniles (7.1%–7.5% TL) than in adults (7.6%–8.0% TL), reaching or mostly often transcending its second dorsal-fin apex in adults but not so often in juveniles. Nasal barbels exceed posterior margin of mouth in juveniles while in adults it reaches anterior margin of mouth or just before it, which it is in agreement with [4]. Height of first and second dorsal fins increases with growth (first dorsal-fin height 8.9%–10.0% TL in juveniles *vs*. 10.0%–10.8% TL in adults; second dorsal-fin height 8.8%–9.6% TL *vs*. 9.3%–10.5% TL, respectively). Other measurements that vary ontogenetically in the Japanese species are: pre-second dorsal length (60.8%–61.0% in juveniles *vs*. 62.0%–66.5% TL in adults); pre-vent length (52.5%–54.8% TL in juveniles *vs*.

54.4%–56.3% TL in adults; prespiracular length (9.5%–9.8% TL in juveniles *vs.* 9.7%–11.3% TL in adults); length of pelvic fin (12.2%–13.3% TL in juveniles *vs.* 13.1%–14.4% TL in adults).

External morphology of *C. barbifer* also varies regionally. Indonesian specimen has pectoral fins very narrow on its posterior margin (*vs.* broad in other specimens) and first dorsal fin conspicuously slender on its upper half (*vs.* conspicuously broad in other specimens). Indonesia/ NC specimens show first dorsal-fin posterior margin straight (*vs.* concave in other specimens). Specimens from Japan have dorsal fins with posterior margin markedly concave (*vs.* mostly straight in Indonesia/NC). Regional variations are also observed for: length of pectoral fin (pectoral-fin inner margin length 8.3%–8.5% TL in Japan *vs.* 9.4%–9.5% TL in Indonesia/NC); head width at nostrils (6.4%–7.6% TL in Japan *vs.* 7.8%–7.9% TL Indonesia/NC); pre-branchial length (14.8%–16.2% TL in Japan *vs.* 18.9%–19.6% TL in Indonesia/NC); length of first dorsal-fin posterior margin (9.0%–10.9% TL in Japan *vs.* 12.1%–12.2% TL in Indonesia/NC). Other morphometric variations are noticed between specimens from Japan and Indonesia/NC, respectively: interdorsal space (21.9%–22.6% TL *vs.* 23.1%–24.5% TL); head length (18.8%– 19.5% TL *vs.* 23.0% TL); length of first dorsal-fin anterior margin (13.5%–14.3% TL *vs.* 12.7%– 12.9% TL); length of second dorsal-fin spine (7.1%–7.6% TL *vs.* 7.9%–8.0% TL); pelvic-fin length (12.2%–13.3% TL *vs.* 13.5%–14.4% TL) and inner margin length (5.4%–6.2% TL *vs.* 6.5%–6.9% TL). These variations may be due to the preservation conditions of the Indonesia/ NC material but a more throughout taxonomic investigation should be considered in the future in order to clarify existing morphotypes in the region.

***Cirrhigaleus asper* (Merrett, 1973) [2].**   Roughskin spurdog

*Squalus asper* Merrett, 1973: 93–110, Figs 1–6, pl. Ib (original description, illustrated; type by original designation; Seychelles) [2]; Bass et al. 1976: 2, 9–11, 18–20, 65, 59, Figs 8 and 12 (description; South Africa, Mozambique) [8]; Compagno 1984: 110, 114 (description; Seychelles, South Africa, Mozambique, Gulf of Mexico, Hawaiian islands) [14]; Bass et al. 1986: 60–61 (cited; South Africa) [94]; Shirai 1992: 8, 35, 106, 121 (listed, cited; South Africa) [22]; Baranes 2003: 42, 46, 48 (cited, Comores) [95]; Manilo and Bogorodsky 2003: 93, 128 (listed, cited; Arabian Sea) [96].

*Cirrhigaleus asper* Shirai, 1992: 121–122 (cited, listed; South Africa) [22]; Compagno 1999: 472 (listed) [78]; Soto 2001: 66 (listed; Brazil) [97]; Compagno 2002: 381–382 (listed, cited; North and Central Atlantic, West Indian Ocean) [98]; Gadig and Gomes 2003: 27 (listed; West Indian and Western Atlantic Oceans) [99]; Kiraly et al. 2003: 2, 12 (listed, cited; North Carolina to Florida, U.S.A., Puerto Rico, Virgin islands) [100]; Heemstra and Heemstra 2004: 54 (cited; Western Indian Ocean) [101]; Soto and Mincarone 2004: 73 (listed; Brazil) [102]; Mundy 2005: 24, 103 (listed, cited; Hawaii, USA) [103]; Compagno et al. 2005: 72, pl. 2 (description; Pacific, Atlantic and Indian Oceans) [6]; Fischer et al. 2006: 495–501 (cited, reproductive biology; Brazil) [104]; Nunan and Senna 2007: 169 (listed; Brazil) [105]; White et al. 2007: 19, 27 (cited; Western Indian Ocean) [3]; Gomes et al. 2010: 42, 43 (cited, description; Brazil) [9]; Naylor et al. 2012: 59, 148 (cited, listed; off Florida, USA) [15]; Ebert 2013: 53–55, 243 (listed, cited, described; South-western Indian Ocean) [12]; Ebert et al. 2013: 74, 81 (cited, description; Indian and Western Atlantic Oceans) [10]; Rosa and Gadig 2014: 93 (listed; Brazil) [106]; Ebert 2015: 55, 56 (cited, listed; South-eastern Atlantic Ocean) [11]; Del Moral-Flores et al. 2015: 58 (listed, Seychelles, Mexico) [83]; Compagno 2016: 1157 (cited only; North Atlantic and Western Indian Oceans) [107]; Weigmann 2016: 902 (cited, Western Indian, North-eastern Pacific, Western Atlantic Oceans) [88]; Ehemann et al. 2019: 4,7,9,12,13, (cited; Venezuela) [108]; Ebert et al 2021: 20–21 (listed; South Africa and Mozambique) [109]; Blanco-Parra and Niño-Torres 2022: 155, 161 (listed; Mexican Caribbean) [110].

*Cyrrhigaleus asper*: Menezes 2011: 4 (listed, misspelling; Brazil) [111].

*Type material*: BMNH 1972.10.10.1 (holotype), adult male, 880 mm TL, off Aldabra Island, Seychelles, Western Indian Ocean, 09˚27'S,46˚23.5'E, 219 meters depth; Collected by Royal Society Indian Ocean Deep Slope Fishing Expedition, 1969 on 10 February, 1969; BMNH 1972.10.10.2 (paratype), adult male, 847 mm TL, off Astove Island, Seychelles, Western Indian Ocean, 10˚04'S,47˚43'E, 329 meters depth, collected on 19 January, 1969; BMNH 1972.10.10.3 (paratype),adult female, 865 mm TL, off Farquhar Island, Seychelles, Western Indian Ocean, 10˚10'S,51˚12'E, 274 meters depth, collected on 24 January, 1969; BMNH 1972.10.10.4 (paratype), adult female, 1002 mm TL, off Assumption Island, Seychelles, Western Indian Ocean, 09˚43'S,46˚29'E, 600 meters depth, collected on 9 January, 1969. Paratypes with collector data same as holotype.

*Type locality*: Aldabra Island, Seychelles, Western Indian Ocean.

*Other material examined (53 specimens)*: MNHN 1884–0149, two neonate males, 138–144 mm TL, St. Paul Island, 38˚40'1"S,77˚30'0"E; MNHN 1884–0151, embryos, less than 90 mm TL, same locality as MNHN 1884–0149; MNHN 1959–0067, embryo male, 107 mm TL, same locality as MNHN 1884–0149; MNHN 1964–0001, neonate male, 207 mm TL, Amsterdam Island, 37˚55'1"S,77˚40'1"E; MNHN 1964–0003, neonate male, 253 mm TL, same locality as MNHN 1964–0001; MNHN 1986–0722, adult female, 760 mm TL, Comores, 12˚0'0"S,43˚25'1"E; MNRJ 30227, adult male, 970 mm TL, off Bahia coast, Brazil, 13˚40'45"S,38˚25'36"W; MNRJ 30228, adult male, 970 mm TL, same locality as MNRJ 30227; NUPEC uncatalogued, two adult females, 1270–1300 mm TL, unknown locality, Brazil; SAIAB 6036, juvenile male, 390 mm TL, unknown locality, South Africa; SAIAB 6037, juvenile female, 380 mm TL, off Kwazulu-Natal, South Africa, 29˚53'36.58"S,31˚29'46.68"E; SAIAB 6038, juvenile female, 320 mm TL, same locality as SAIAB 6037; SAIAB 6040, adult female, 1120 mm TL, Amatikulu, South Africa, 29.04˚S,31.53˚E; SAIAB 6092, juvenile female, 275 mm TL; neonate male, 270 mm TL, unknown locality, South Africa; SAIAB 25423, adult female, 1110 mm TL, off Coffee Bay, South Africa, 31.98˚S,29.14˚E; SAIAB 27027, adult female, 1090 mm TL, Kowie River mouth, Port Alfred, South Africa, 33˚36'9.83"S,26˚54'6.37"E; SAIAB 31890, adult female, 1090 mm TL, off Gonubie, South Africa, 32˚57'25.29"S,28˚01'46.26"E; SAIAB 186460, juvenile female, 456 mm TL, off Durban, South Africa, 30.00˚S,31.15˚E; SAM 38268, nine neonate females, 132–145 mm TL, seven neonate males, 130–137 mm TL, off Coffee Bay, Transkei, South Africa, 31˚59'2.59"S,29˚09'42.48"E; SAM 39879, adult female, 1023 mm TL, Gulf of Mexico, 27˚6'16.12"N,88˚8'36.41"W; UERJ 1641, adult male, 990 mm TL, off Rio de Janeiro, Brazil, 23˚9'55.29"S,43˚1'16.34"W; USNM 220585, juvenile male, 245 mm TL, off Venezuela coast, 11˚9.8'N,69˚5'W; USNM 217364, adult female, 1000 mm TL, Texas, U.S.A., 27.7˚-N,94.27˚W; NMNZ P 39849, adult female, 1140 mm TL, Desoto Canyon, Louisiana, USA, 28˚17'N,92˚11'W; NMNZ P 39850, adult female, 1100 mm TL, Desoto Canyon, Louisiana, USA, 27˚13'N,91˚42'W; Uncatalogued specimen (field number 227CAS0218), adult female, 1068mm TL, Pernambuco coast, Brazil, 8˚10'19.4"S, 34˚33'57.7"W, 233m depth; Uncatalogued specimen (field number 161CAS1016), juvenile female, 578 mm TL, Pernambuco coast, Brazil, 8˚35'52.2"S, 34˚42'50.8"W, 411m depth; Uncatalogued specimen (field number 114CAS1115), juvenile male, 945 mm TL, Pernambuco coast, Brazil, 7˚49'30.2"S, 34˚27'33.0"W, 338m depth; Uncatalogued specimen (field number 121CAS0716), juvenile male, 880mm TL, Pernambuco coast, Brazil, 8˚51'15.6"S, 34˚46'45.0"W, 500m depth; Uncatalogued specimen (not retained, field number 230CAS0218), juvenile female, 851mm TL, Pernambuco coast, Brazil, 8˚10'9.7"S, 34˚33'54.1"W, 229m depth; Uncatalogued specimen (not retained, field number 210CAS0118), juvenile female, 880mm TL, Pernambuco coast, Brazil, 8˚12'20.9"S, 34˚34'42.4"W, 313m depth; Uncatalogued specimen (not retained, field number 213CAS0118), juvenile male, 842mm TL, Pernambuco coast, Brazil, 8˚12'20.9"S, 34˚34'42.4"W, 313m depth; Uncatalogued specimen (not retained, field number 229CAS0218), adult male, 964mm TL,

Pernambuco coast, Brazil, 8˚10'19.4"S, 34˚33'57.7"W, 233m depth; Uncatalogued specimen (not retained, field number 208CAS0118), adult male, 985mm TL, Pernambuco coast, Brazil, 8˚12'20.9"S, 34˚34'42.4"W, 313m depth.

*Diagnosis. Cirrhigaleus asper* is easily distinguished from its congeners by having anterior margin of nostrils conspicuously short, its length 1.0%–1.5% TL and non moustache-like, never reaching the mouth (*vs.* 5.4%–6.4% TL in *C. barbifer vs.* 4.4%–6.2% TL in *C. australis*, moustache-like and reaching the mouth), and body brown in colour when preserved (*vs.* dark grey in *C. barbifer vs.* light grey in *C. australis*). It differs from *C. barbifer* and *C. australis* by larger upper labial furrow, its length 1.4%–2.1% TL (*vs.* 1.1%-1.5% TL for *C. barbifer vs.* 0.9%–1.4% TL for *C. australis*), shorter pelvic fins, its length 10.8%–12.7% TL (*vs.* 13.1%–14.4% TL in *C. barbifer vs.* 11.9%–14.8% TL in *C. australis*), and lower second dorsal fin, its height 3.9%–9.3% TL (vs. 9.3%–10.5% TL in *C. barbifer vs.* 8.4%–9.9% TL for *C. australis*). It is also separated from its congeners by having dermal denticles heart-shaped (*vs.* bat-like), median ridge conspicuously broad (*vs.* median ridge very narrow) and lateral cusps inconspicuous at posterior margin of crown (*vs.* lateral cusps conspicuous) (Fig 21).

*Description.* Morphometric and meristic data are shown in Table 2 and S3 and S4 Tables.

*External morphology.* Body trihedral, stout and robust, conspicuously arched dorsally from anterior margin of spiracle to pelvic fin insertion, turning more slender to the tail (Fig 21); body with greatest width at head than at trunk and abdomen; head width 1.5 (0.6–1.5) times greater than trunk width and 1.8 (0.7–1.8) times wider than abdomen width; body equally deep from head to abdomen with head height 0.9 (0.7–1.2) times trunk height and 0.9 (0.6–1.3) times abdomen height. Head flattened anteriorly and elongate, its length 22.4% (19.7%–24.3%) TL; head narrower at nostrils than at mouth (its width at nostrils 7.7%, 6.0%–10.2% TL; width at mouth 12.5%, 10.8%–13.4% TL). Eyes markedly elliptical with anterior and posterior margins notched; eyes conspicuously large, its length 4.5% (2.0%–6.0%) TL, corresponding to 5.8 (1.7–6.0) times greater than its height. Prespiracular length 0.5 (0.5–07) times prepectoral length and 1.7 (1.5–1.8) times preorbital length. Spiracles crescent, located posteriorly and above the eyes; spiracles conspicuously large, its length 0.3 (0.2–0.7) times eye length. Gill slit somewhat concave, vertical and markedly tall with fifth gill slit 1.1 (0.8–1.4) times higher than first gill slit.

Snout rounded and short with preorbital length 6.5% (5.9%–8.4%) TL; nostrils equally distant to snout tip and mouth with prenarial length 0.9 (0.8–1.2) times distance from nostrils to upper labial furrow and 0.5 (0.5–0.6) times preoral length; nostrils with anterior margin bilobate and markedly broad; medial lobe of anterior margin larger than lateral one, although not moustache-like and elongate; anterior nasal flap with its length 1.5% (1.0%–2.4%) TL, corresponding to 14.7 (9.6–22.5) times smaller than head length. Mouth arched and broad (its width 8.1%, 7.4%–9.2% TL); upper labial furrow markedly small, its length 1.6% (1.0%–2.8%) TL with a small and thick fold; lower labial furrow also small, without fold. Teeth similar in both jaws, labial-lingually flattened, very broad but low at the crown (Fig 22); lower teeth larger than upper teeth; teeth unicuspid with cusp thick and large, somewhat upright and lateral; mesial cutting edge straight and diagonal; distal heel rounded; mesial heel pointed; apron very short and narrow. Three series of functional teeth on both jaws; upper jaw with 14–14 (13–13) teeth rows; lower jaw with 12–12 (12–11) teeth rows.

Dorsal fins conspicuously upright and vertical (Fig 23), both equally tall with first dorsal fin height 1.0 (0.9–2.1) times height of second dorsal fin; first dorsal fin as large as second dorsal fin, its length 1.0 (0.9–1.5) times length of second dorsal fin. Origin of the first dorsal fin located behind the vertical traced at pectoral free rear tips, although near to it in most specimens. First dorsal fin with anterior margin convex and large, its length 13.7% (10.8%–14.5%) TL; first dorsal-fin posterior margin straight and large, its length 10.1% (6.9%–11.8%) TL; first

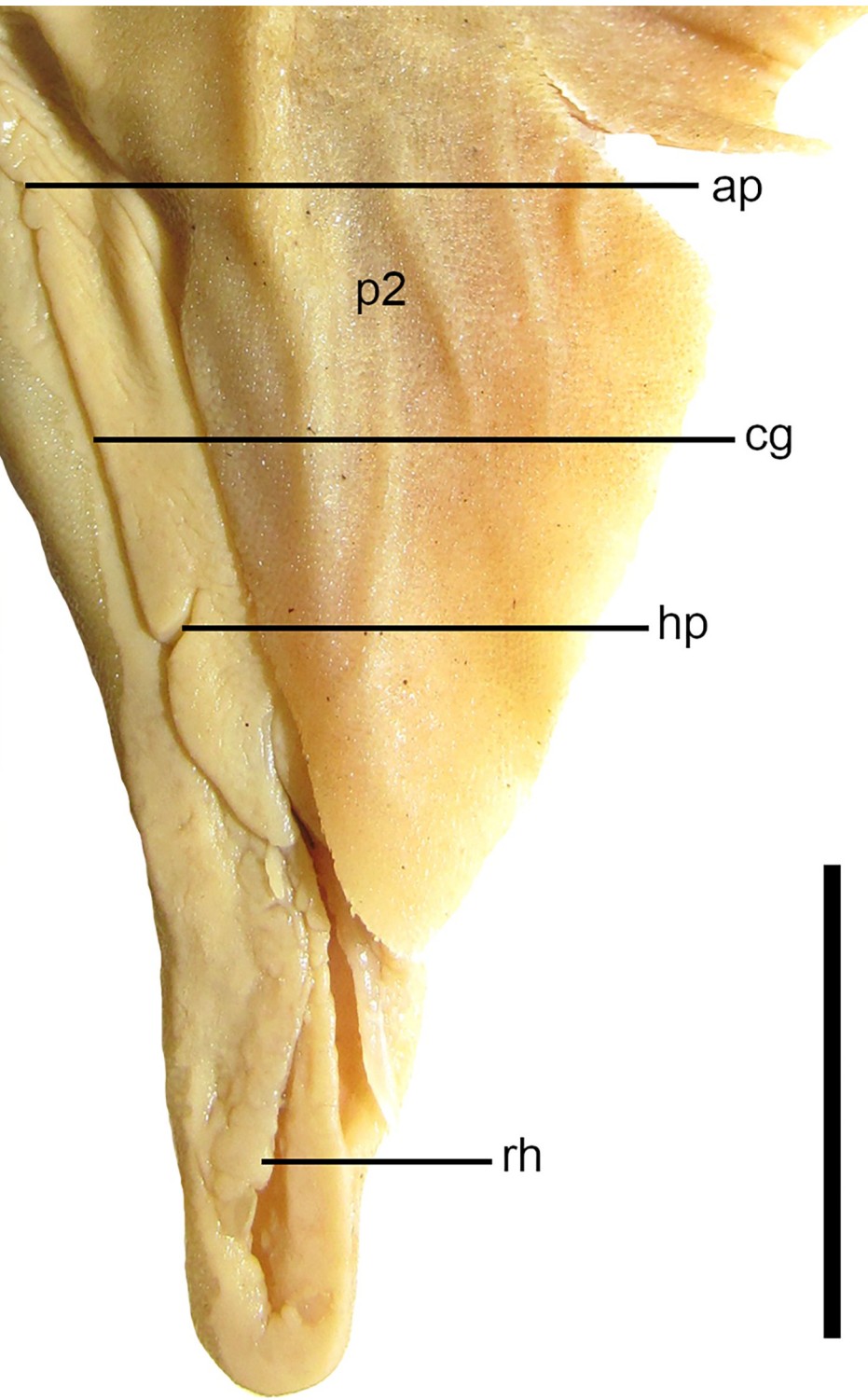

**Fig 21. Specimens of *C. asper* in lateral view.** A: BMNH 1972.10.10.1 (holotype), adult male, 880 mm TL, Seychelles; B: SAIAB 6092, juvenile female, 275 mm TL, South Africa; C: USNM 220585, juvenile male, 245 mm TL, the U.S.A. Scale bars: 50 mm.

**Table 2. External measurements of *C. asper* expressed as percentage of total length (%TL).**

| | *Cirrhigaleus asper* | | | | | | | | | | | | | | | | | |
|---|---|---|---|---|---|---|---|---|---|---|---|---|---|---|---|---|---|---|
| | WIO | | | | | | | | WCAO | | | | SWAO | | | | *x* | SD |
| | H | P1 | P2 | P3 | n | | | | n | | | | n | | | | | |
| TL (mm) | 880.0 | 847.0 | 865.0 | 1002.0 | 12 | 253.0 | – | 1120.0 | 2 | 1000.0 | – | 1023.0 | 9 | 560.0 | – | 1300.0 | 816.7 | 327.1 |
| PCL | 81.3 | 81.7 | 80.7 | 80.6 | 12 | 76.4 | – | 84.8 | 2 | 79.7 | – | 81.5 | 9 | 76.9 | – | 94.5 | 80.5 | 3.7 |
| PD2 | 63.3 | 64.9 | 64.0 | 63.5 | 12 | 60.0 | – | 68.3 | 2 | 63.0 | – | 64.0 | 9 | 59.8 | – | 70.9 | 63.2 | 2.9 |
| PD1 | 32.5 | 32.5 | 34.9 | 34.5 | 12 | 30.4 | – | 35.2 | 2 | 31.8 | – | 36.0 | 9 | 30.2 | – | 33.1 | 33.0 | 1.5 |
| SVL | 52.3 | 56.6 | 51.7 | 53.4 | 12 | 45.1 | – | 61.6 | 2 | 50.8 | – | 52.0 | 9 | 51.0 | – | 57.5 | 53.6 | 3.3 |
| PP2 | 48.1 | 51.0 | 49.2 | 50.2 | 12 | 42.8 | – | 58.3 | 2 | 47.9 | – | 50.0 | 9 | 47.4 | – | 54.7 | 50.7 | 3.2 |
| PP1 | 21.8 | 21.3 | 21.7 | 20.8 | 12 | 19.8 | – | 23.4 | 2 | 15.0 | – | 20.3 | 9 | 18.3 | – | 22.3 | 20.9 | 1.8 |
| HDL | 22.4 | 21.9 | 22.6 | 21.1 | 12 | 20.3 | – | 24.3 | 2 | 22.0 | – | 23.0 | 9 | 19.7 | – | 22.6 | 22.0 | 1.3 |
| PG1 | 18.4 | 17.0 | 18.8 | 17.6 | 12 | 17.4 | – | 20.4 | 2 | 17.6 | – | 19.0 | 9 | 16.5 | – | 18.5 | 18.2 | 1.0 |
| PSP | 11.3 | 11.0 | 11.2 | 10.9 | 12 | 10.0 | – | 14.3 | 2 | 9.6 | – | 11.1 | 9 | 9.4 | – | 11.5 | 11.2 | 1.1 |
| POB | 6.5 | 6.5 | 6.3 | 6.0 | 12 | 6.5 | – | 8.4 | 2 | 5.9 | – | 6.0 | 9 | 6.0 | – | 6.5 | 6.6 | 0.6 |
| PRN | 4.3 | 4.9 | 4.8 | 4.6 | 12 | 4.2 | – | 5.7 | 2 | 2.9 | – | 3.1 | 9 | 3.6 | – | 4.7 | 4.6 | 0.7 |
| POR | 8.1 | 7.9 | 8.8 | 8.2 | 12 | 7.3 | – | 11.1 | 2 | 5.3 | – | 6.9 | 9 | 6.1 | – | 8.3 | 8.1 | 1.3 |
| INLF | 4.5 | 4.0 | 4.3 | 4.4 | 12 | 3.8 | – | 4.9 | 2 | 2.8 | – | 4.1 | 9 | 3.5 | – | 4.9 | 4.2 | 0.5 |
| MOW | 8.1 | 8.2 | 8.3 | 8.2 | 12 | 7.4 | – | 8.8 | 2 | 8.3 | – | 9.0 | 9 | 7.5 | – | 9.2 | 8.2 | 0.4 |
| ULA | 1.6 | 1.6 | 1.6 | 1.4 | 12 | 1.5 | – | 2.8 | 2 | 2.1 | – | 2.1 | 9 | 1.0 | – | 1.9 | 1.7 | 0.3 |
| INW | 3.6 | 3.7 | 3.8 | 3.7 | 12 | 3.5 | – | 4.8 | 2 | 3.2 | – | 3.4 | 9 | 3.3 | – | 3.9 | 3.7 | 0.3 |
| INO | 8.2 | 8.3 | 8.5 | 8.1 | 12 | 8.1 | – | 10.1 | 2 | 8.2 | – | 8.7 | 9 | 7.3 | – | 8.5 | 8.4 | 0.6 |
| EYL | 4.5 | 4.5 | 4.2 | 4.3 | 12 | 3.6 | – | 6.0 | 2 | 2.3 | – | 4.1 | 9 | 2.0 | – | 5.2 | 4.3 | 1.2 |
| EYH | 0.8 | 1.3 | 0.9 | 0.9 | 12 | 1.1 | – | 2.5 | 2 | 0.9 | – | 1.0 | 9 | 0.8 | – | 1.4 | 1.3 | 0.4 |
| SPL | 1.2 | 1.4 | 1.2 | 1.3 | 12 | 1.3 | – | 1.7 | 2 | 1.3 | – | 1.4 | 9 | 1.0 | – | 1.4 | 1.3 | 0.2 |
| GS1 | 2.1 | 2.3 | 1.9 | 2.2 | 12 | 1.7 | – | 2.7 | 2 | 1.8 | – | 2.6 | 9 | 1.6 | – | 2.7 | 2.2 | 0.3 |
| GS5 | 2.4 | 2.7 | 2.6 | 2.9 | 12 | 2.1 | – | 3.2 | 2 | 2.3 | – | 3.0 | 9 | 1.9 | – | 2.6 | 2.5 | 0.4 |
| IDS | 23.1 | 21.8 | 21.9 | 21.3 | 12 | 19.1 | – | 24.1 | 2 | 22.5 | – | 24.0 | 9 | 19.6 | – | 24.6 | 22.0 | 1.8 |
| DCS | 9.5 | 9.3 | 8.3 | 8.2 | 12 | 7.4 | – | 10.1 | 2 | 7.5 | – | 10.0 | 9 | 8.6 | – | 10.8 | 9.2 | 1.0 |
| PPS | 26.1 | 28.2 | 21.1 | 22.6 | 12 | 17.6 | – | 33.0 | 2 | 22.5 | – | 29.0 | 9 | 24.5 | – | 31.5 | 26.2 | 3.8 |
| PCA | 23.5 | 22.3 | 21.0 | 20.9 | 12 | 20.0 | – | 24.6 | 2 | 22.0 | – | 25.0 | 9 | 20.5 | – | 24.4 | 22.3 | 1.4 |
| D1L | 14.8 | 14.8 | 13.8 | 14.2 | 12 | 12.3 | – | 15.8 | 2 | 15.5 | – | 16.7 | 9 | 13.9 | – | 15.5 | 14.3 | 1.1 |
| D1A | 13.7 | 12.4 | 11.8 | 12.3 | 12 | 11.2 | – | 13.4 | 2 | 13.7 | – | 14.5 | 9 | 10.8 | – | 12.8 | 12.2 | 0.9 |
| D1B | 9.4 | 8.9 | 8.6 | 8.7 | 12 | 7.0 | – | 10.0 | 2 | 9.5 | – | 9.6 | 9 | 8.2 | – | 10.0 | 8.8 | 0.9 |
| D1H | 8.6 | 8.5 | 9.5 | 8.2 | 12 | 7.0 | – | 9.6 | 2 | 9.1 | – | 9.9 | 9 | 7.7 | – | 9.2 | 8.5 | 0.7 |
| D1I | 5.3 | 6.0 | 5.8 | 5.6 | 12 | 4.2 | – | 6.4 | 2 | 6.2 | – | 6.3 | 9 | 4.7 | – | 6.2 | 5.6 | 0.5 |
| D1P | 10.1 | 8.0 | 9.2 | 9.4 | 12 | 6.9 | – | 10.2 | 2 | 10.9 | – | 11.8 | 9 | 9.0 | – | 10.3 | 9.1 | 1.2 |
| D1ES | 5.7 | 3.4 | 6.1 | 5.8 | 9 | 1.9 | – | 6.3 | 1 | 7.6 | – | 7.6 | 6 | 3.7 | – | 6.1 | 4.7 | 1.6 |
| D1BS | 1.1 | 1.3 | 1.2 | 1.1 | 11 | 0.6 | – | 1.2 | 2 | 1.3 | – | 1.4 | 9 | 0.8 | – | 1.5 | 1.1 | 0.2 |
| D2L | 14.6 | 13.4 | 14.6 | 9.3 | 12 | 12.1 | – | 14.8 | 2 | 13.6 | – | 15.2 | 9 | 11.4 | – | 15.3 | 13.4 | 1.3 |
| D2A | 13.9 | 12.8 | 13.5 | 7.8 | 12 | 11.5 | – | 13.8 | 2 | 13.9 | – | 14.2 | 9 | 10.8 | – | 14.5 | 12.7 | 1.4 |
| D2B | 9.5 | 9.3 | 9.4 | 8.3 | 12 | 7.2 | – | 9.7 | 2 | 8.6 | – | 9.1 | 9 | 7.5 | – | 10.1 | 8.7 | 0.8 |
| D2H | 8.5 | 8.2 | 8.6 | 3.9 | 12 | 7.0 | – | 9.0 | 2 | 9.3 | – | 9.3 | 9 | 7.5 | – | 8.6 | 7.9 | 1.0 |
| D2I | 5.1 | 4.4 | 5.3 | 1.5 | 12 | 4.4 | – | 6.1 | 2 | 4.8 | – | 5.4 | 9 | 3.9 | – | 5.4 | 4.9 | 0.8 |
| D2P | 6.9 | 6.5 | 7.9 | 3.9 | 12 | 5.8 | – | 8.0 | 2 | 7.8 | – | 8.0 | 9 | 6.8 | – | 7.7 | 6.9 | 0.9 |
| D2ES | 7.3 | 4.7 | 3.4 | 5.3 | 11 | 4.3 | – | 7.2 | 2 | 6.3 | – | 7.6 | 8 | 5.2 | – | 6.7 | 5.7 | 1.1 |
| D2BS | 1.0 | 1.0 | 0.9 | 1.0 | 12 | 0.8 | – | 1.2 | 2 | 1.1 | – | 1.2 | 9 | 0.9 | – | 1.2 | 1.0 | 0.1 |
| P1A | 14.4 | 12.5 | 13.9 | 13.2 | 12 | 12.2 | – | 16.1 | 2 | 15.6 | – | 16.0 | 9 | 12.2 | – | 15.2 | 13.9 | 1.1 |
| P1I | 8.4 | 7.4 | 8.8 | 8.6 | 12 | 8.1 | – | 9.4 | 2 | 8.6 | – | 9.0 | 9 | 7.9 | – | 11.7 | 8.7 | 0.7 |

(*Continued*)

**Table 2.** (Continued)

| | *Cirrhigaleus asper* | | | | | | | | | | | | | |
|---|---|---|---|---|---|---|---|---|---|---|---|---|---|---|
| | **WIO** | | | | | | | | **WCAO** | | | **SWAO** | | ***x*** | **SD** |
| | **H** | **P1** | **P2** | **P3** | ***n*** | | ***n*** | | ***n*** | | | | | | |
| P1B | 4.3 | 4.6 | 4.3 | 4.6 | 12 | 3.5 – 5.7 | 2 | 5.4 – 5.9 | 9 | 4.2 – 5.5 | | | | 4.8 | 0.5 |
| P1P | 10.5 | 8.7 | 12.0 | 9.3 | 12 | 9.7 – 12.9 | 2 | 10.8 – 12.0 | 9 | 8.8 – 10.7 | | | | 10.4 | 1.0 |
| P2L | 12.2 | 11.2 | 12.0 | 10.8 | 12 | 10.2 – 12.4 | 2 | 11.5 – 12.0 | 9 | 9.3 – 12.7 | | | | 11.2 | 0.8 |
| P2I | 6.3 | 5.6 | 5.9 | 3.7 | 12 | 4.1 – 6.4 | 2 | 5.6 – 7.6 | 9 | 4.7 – 6.7 | | | | 5.6 | 0.9 |
| CDM | 19.8 | 18.4 | 20.1 | 20.1 | 12 | 18.8 – 23.2 | 2 | 22.5 – 23.0 | 9 | 17.7 – 21.7 | | | | 20.6 | 1.5 |
| CPV | 11.6 | 10.9 | 11.4 | 11.2 | 12 | 7.2 – 12.4 | 2 | 11.8 – 13.0 | 9 | 9.4 – 13.0 | | | | 11.1 | 1.2 |
| CFW | 7.8 | 7.1 | 7.5 | 7.6 | 12 | 6.8 – 13.0 | 2 | 8.0 – 8.2 | 9 | 6.9 – 7.6 | | | | 7.6 | 1.1 |
| HANW | 7.7 | 6.9 | 7.0 | 7.3 | 12 | 7.8 – 10.2 | 2 | 7.2 – 7.9 | 9 | 6.0 – 9.2 | | | | 8.0 | 1.1 |
| HAMW | 12.5 | 13.4 | 12.8 | 12.6 | 12 | 11.6 – 13.3 | 2 | 12.1 – 12.5 | 9 | 10.8 – 12.5 | | | | 12.3 | 0.7 |
| HDW | 14.0 | 13.8 | 14.4 | 15.5 | 12 | 12.3 – 22.1 | 2 | 14.3 – 26.0 | 9 | 11.1 – 23.4 | | | | 16.4 | 4.2 |
| TRW | 9.4 | 10.1 | 12.8 | 13.8 | 12 | 8.6 – 24.5 | 2 | 20.5 – 30.5 | 9 | 12.5 – 24.6 | | | | 15.9 | 5.5 |
| ABW | 8.0 | 9.0 | 8.7 | 11.6 | 12 | 8.0 – 22.7 | 2 | 14.7 – 21.0 | 9 | 9.5 – 21.4 | | | | 13.4 | 4.8 |
| HDH | 11.5 | 9.8 | 12.1 | 12.1 | 12 | 8.4 – 15.8 | 2 | 11.7 – 15.0 | 9 | 8.3 – 13.1 | | | | 11.0 | 1.9 |
| TRH | 12.1 | 8.9 | 12.2 | 13.7 | 12 | 8.6 – 16.7 | 2 | 12.7 – 21.0 | 9 | 8.4 – 15.1 | | | | 12.3 | 2.9 |
| ABH | 12.9 | 9.6 | 12.5 | 10.7 | 12 | 7.1 – 15.8 | 2 | 11.2 – 20.0 | 9 | 10.0 – 17.5 | | | | 12.5 | 2.7 |
| CLO | 4.2 | 4.0 | – | – | 3 | 1.2 – 1.7 | – | – 3.5 | 5 | – 4.3 | | | | 3.2 | 1.3 |
| CLI | 6.9 | 6.8 | – | – | 3 | 2.2 – 4.1 | – | – 6.7 | 5 | – 7.3 | | | | 5.7 | 2.0 |
| ANFL | 1.5 | 1.3 | 1.2 | 1.3 | 9 | 1.0 – 2.4 | – | – 1.1 | 4 | – 1.3 | | | | 1.3 | 0.4 |

Range values for other material are provided by region for comparisons. TL is expressed in millimeter. Mean and standard deviation include all the specimens in which measurement were taken. H: holotype, BMNH 1972.10.10.1; P1: paratype, BMNH 1972.10.10.2; P2: paratype, BMNH 1972.10.10.3; P3: paratype, BMNH 1972.10.10.4. *n*: number of specimens; *x*: mean; SD: standard deviation; SWAO: Southwestern Atlantic Ocean; WCAO: Western Central Atlantic Ocean; WIO: Western Indian Ocean.

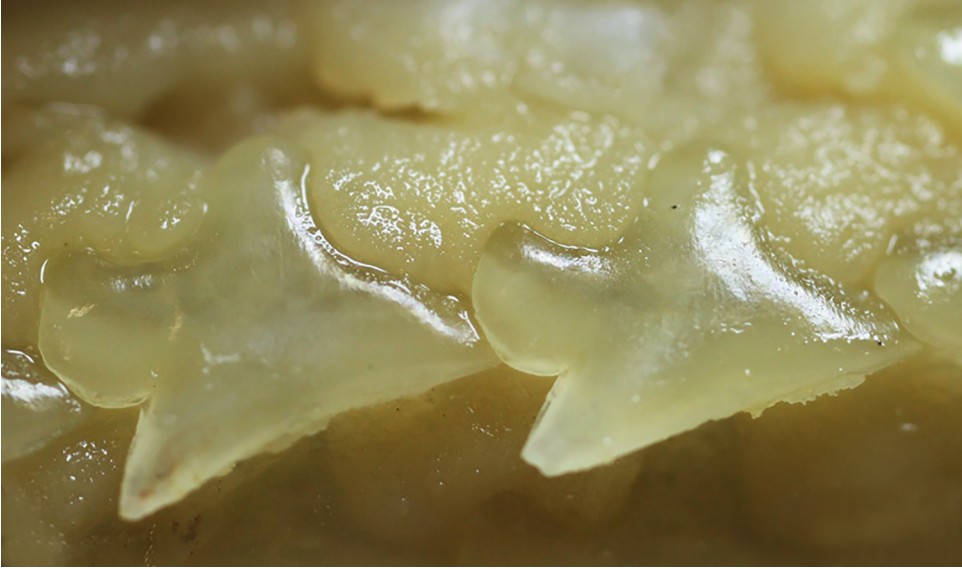

**Fig 22. Upper teeth (labial view) of the paratype of *C. asper*, BMNH 1972.10.10.2, adult male, 847 mm TL.**

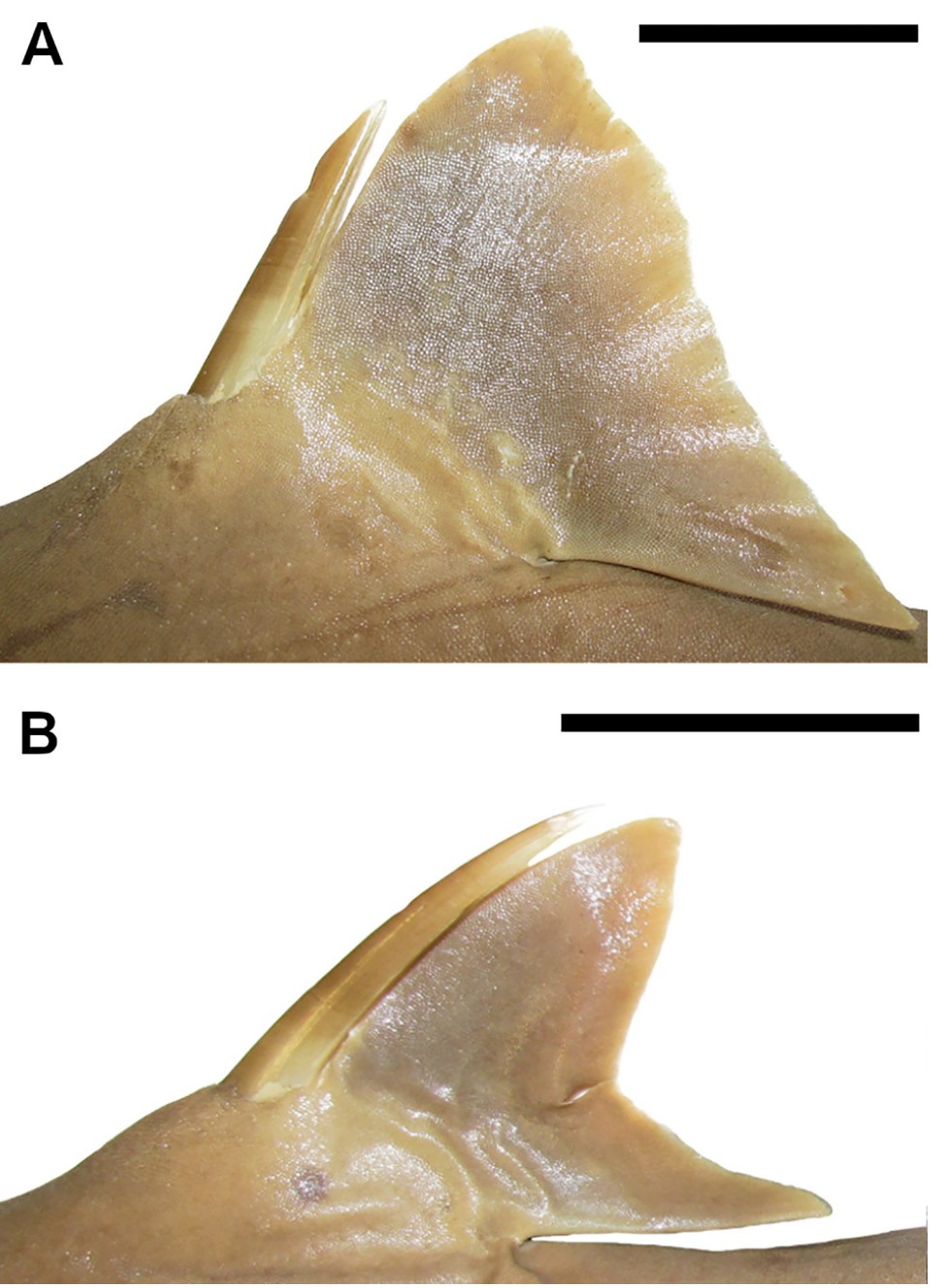

**Fig 23.** First (A) and second (B) dorsal fins of *C. asper*. A,B: BMNH 1972.10.10.1, adult male, 880 mm TL. Scale bars: 20 mm.

dorsal-fin apex narrow and free rear tip pointed; first dorsal fin conspicuously tall, its height 1.3 (1.0–1.5) times greater than preorbital length and 1.6 (1.4–2.1) times greater than inner margin length of first dorsal fin. First dorsal-fin spine thick (its base width 1.1%, 0.6%–1.5% TL) and markedly elongate, its length 5.7% (1.9%–7.6%) of TL; first dorsal-fin spine not reaching the fin apex, its length 0.7 (0.2–0.8) times height of first dorsal.

Interdorsal space almost equal to prepectoral length, corresponding to 1.1 (0.8–1.6) times the latter and 2.4 (2.0–3.0) times larger than dorsal-caudal space. Origin of second dorsal fin

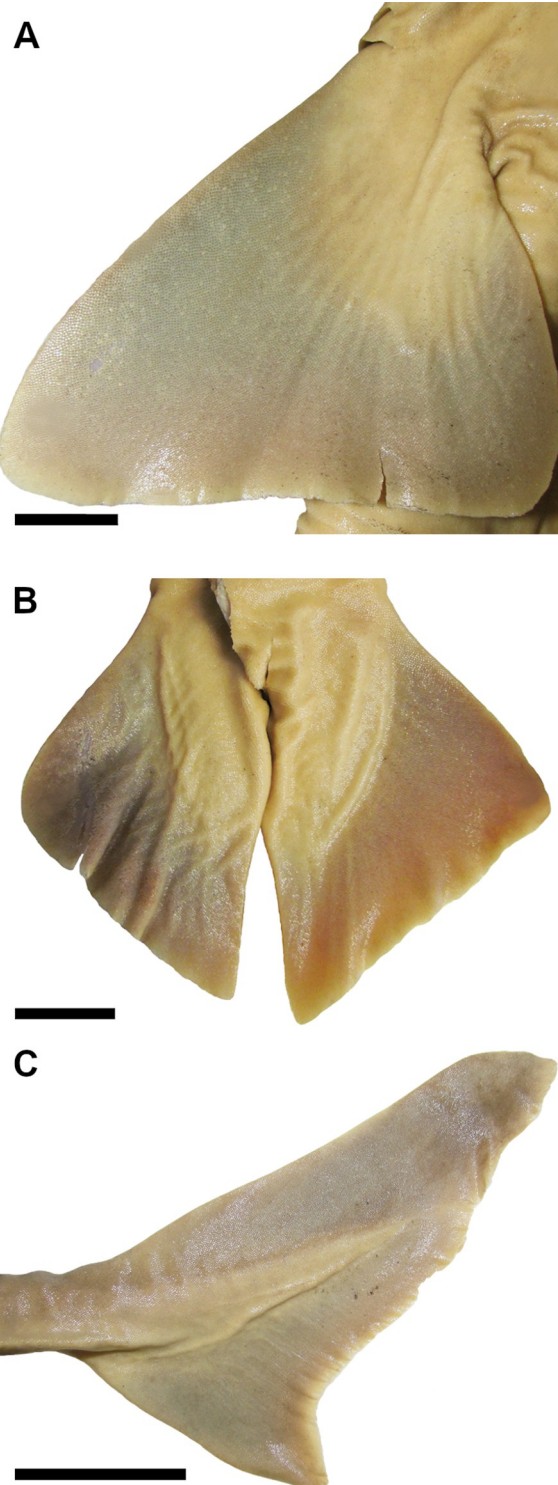

**Fig 24.** Pectoral (A), pelvic (B) and caudal (C) fins of the paratype of *C. asper* in ventral (A,B) and lateral views (C). A, B: BMNH 1972.10.10.3, adult female, 865 mm TL; C: BMNH 1972.10.10.1, adult male, 880 mm TL. Scale bars: 20 mm.

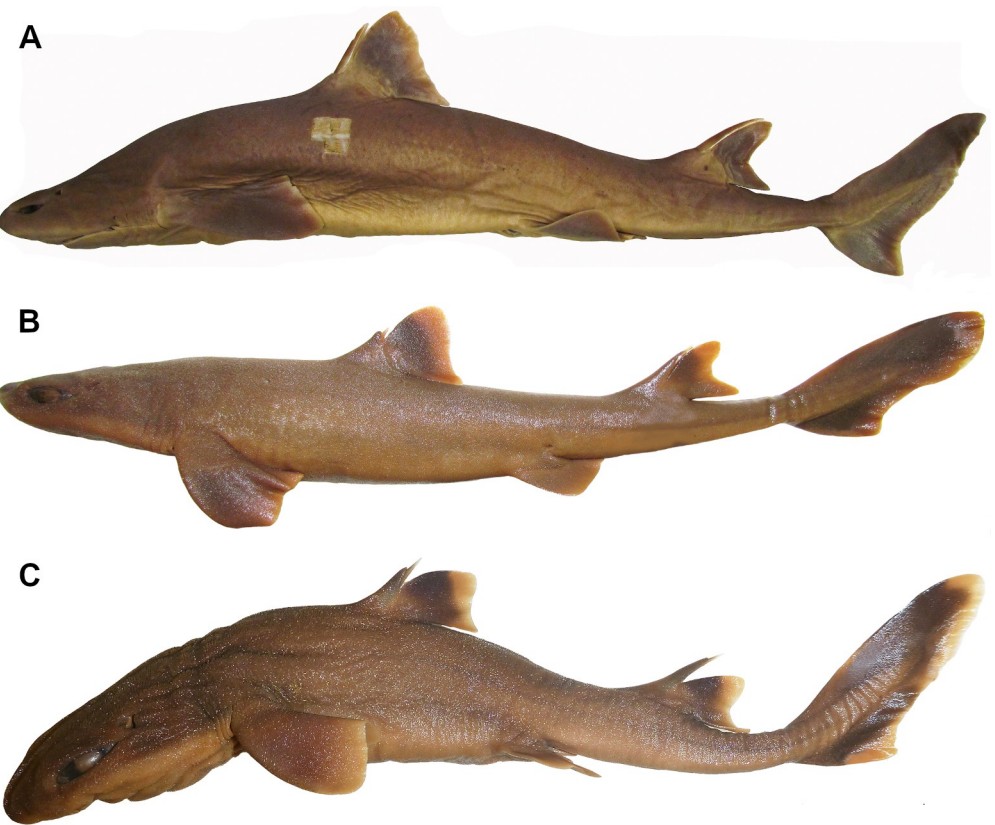

**Fig 25. Clasper (right side) of the paratype of *C. asper*, BMNH 1972.10.10.2, adult male, 847 mm TL in dorsal view.** Abbreviations: ap: apopyle; cg: clasper groove; hp: hypopyle; p2: pelvic fin; rh: rhipidion.

over vertical traced at free rear tips of pelvic fins. Second dorsal fin with anterior margin convex and large, its length 13.9% (7.8%–14.5%) TL; second dorsal-fin posterior margin markedly falcate and elongate, its length 6.9% (3.9%–8.0%) TL; second dorsal-fin apex conspicuously slender and lobe-like, and free rear tip pointed. Second dorsal-fin spine thick (its base width 1.0%, 0.8%–1.2% TL), and elongate, its length 7.3% (3.4%–7.6%) TL and not reaching second dorsal-fin apex with its length 0.9 (0.4–1.4) times height of second dorsal fin; second dorsal-fin spine slightly larger than first one (its length 1.3, 0.6–2.2 times length of first dorsal spine).

Pectoral fin conspicuously wide distally and square-like (Fig 24A); pectoral-fin anterior margin convex and elongate, its length 1.7 (1.2–1.9) times larger than pectoral-fin inner margin length, and corresponding at least to one-half the distance between pectoral and pelvic fins; pectoral-fin inner margin markedly convex; pectoral-fin posterior margin straight and large, its length 10.5% (8.7%–12.9%) TL; apex and free rear tip of pectoral fins conspicuously rounded and broad, not lobe-like and almost reaching the same length. Pectoral-pelvic distance 1.1 (0.7–1.5) times pelvic-caudal space; the latter 1.0 (0.8–1.3) times interdorsal space. Pelvic fins in the midline between first and second dorsal fin, although nearest to second dorsal fin in some paratypes and non-type specimens. Pelvic fins markedly broad with anterior margin somewhat convex (Fig 24B), posterior margin straight; pelvic-fin apex rounded and conspicuously wide, and free rear tips rounded and lobulated, although more slender in males.

Claspers in adults slender and flattened ventrally (Fig 25); claspers very short its inner length 1.1, (0.4–1.2) times pelvic inner margin length, transcending pelvic free rear tips; clasper groove very large, vertical, placed dorsally; apopyle and hypopyle with narrow apertures,

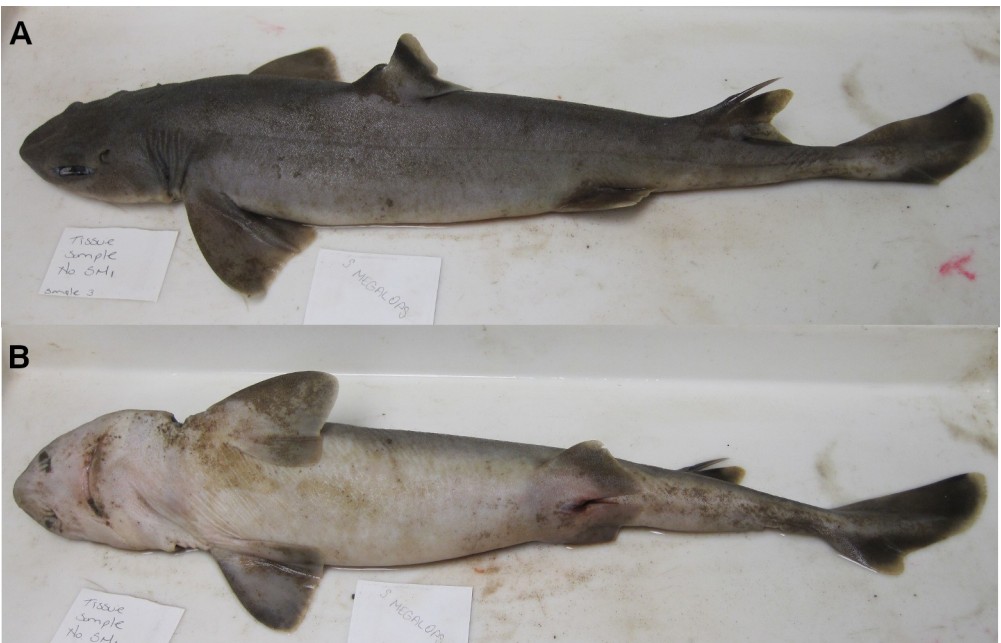

**Fig 26.** Fresh specimen of *C. asper* in lateral (A) and ventral (B) views, SAIAB 186460, juvenile female, 456 mm TL, South Africa. Reprinted from NRF-SAIAB under a CC BY license, with permission NRF-SAIAB, 2010.

located proximal and distally in the clasper groove, respectively; rhipidion flap-like, markedly narrow and short (not reaches distal end of clasper), attached mediodistally to clasper.

Caudal peduncle very short with dorsal-caudal space 9.5% (7.4%–10.8%) TL with lateral keel prominent and thick since insertion of second dorsal fin to behind origin of caudal fin; upper and lower precaudal pits absent. Caudal fin (Fig 24C) rather rectangular with dorsal caudal margin convex anteriorly and turning straight until the dorsal caudal tip; length of dorsal caudal margin 0.9 (0.8–1.0) times head length and 1.7 (1.7–3.2) times larger than length of preventral caudal margin; dorsal caudal tip pointed; upper post-ventral caudal margin concave, although straight near the dorsal caudal tip; lower post-ventral caudal margin straight; preventral caudal margin convex and short, its length 1.8 (1.4–3.1) times larger than inner length of pelvic fin; caudal fin slightly continuous between lobes and markedly broad at fork, its width 7.8% (6.8%–13.0%) TL.

*Dermal denticles.* Dermal denticles tricuspid and heart-shaped (Fig 12I–12L), conspicuously broad at the crown and imbricated; denticles with its width as large as its length; median cusp prominent, pointed and elongate; lateral cusps very short to inconspicuous; median ridge straight, tall, short, markedly thick anteriorly and thin posteriorly; conspicuous anterior furrow and often two small ridges on each side; median ridge projecting anteriorly beyond the crown base; one or two lateral ridges on each side, straight to slightly convex, tall and short, reaching the tip of crown edge; lateral furrow very profound and wide, placed aside each lateral ridge.

*Colouration.* In fresh material (Fig 26), body light grey dorsally and laterally up to the origin of pectoral fin, light grey laterally, and white ventrally. Pectoral and pelvic fins grey dorsally and whitish ventrally with light grey blotches at the edges and posterior margins white. Dorsal fins light grey and dark grey at the base of the dorsal-fin spines; dorsal-fin posterior margin broadly white from apex to free rear tips; submarginal bar broadly black. Dorsal-fin spines brownish grey, whitish at the tip. Caudal fin light grey, whitish at the vertebral column region;

postventral caudal margins white with ventral caudal tip broadly white; upper caudal blotch broadly black; lower subterminal bar broadly black at the lower caudal lobe. In preserved specimens, body brownish-grey dorsally and white ventrally; pectoral, pelvic, dorsal and caudal fins darker than the rest of the body. Pectoral fins with inner and posterior margins homogenously white. Pelvic fins pale ventrally with inner and posterior margins broadly white. Dorsal fins with posterior margins broadly white from apex to free rear tips; dorsal fins lighter at the fin base near the dorsal-fin spine. Dorsal-fin spines brown, whitish at its base and tip. Caudal fin also brown with white postventral caudal margins, except at the dorsal caudal tip; ventral caudal tip broadly white; dorsal caudal margin white, except near the dorsal caudal tip; ventral caudal lobe with white basal marking; caudal stripe light brown.

*Vertebral counts*. Monospondylous vertebrae 52 (49–52); precaudal vertebrae 87 (85–90); total vertebrae 116 (115–119).

*Geographical distribution*. It occurs in the Western Indian Ocean (Seychelles to South Africa and south to St. Paul Islands) and Western Atlantic Ocean (Texas, USA to Rio de Janeiro, Brazil), including the Caribbean Sea (Venezuela [108] and Mexico [110]). *Cirrhigaleus asper* is often caught between 91–750 meters depth. Its occurrence in Hawaii, USA is not confirmed as material from this region was not assessed in the present study (Fig 15).

## Remarks

*Intraspecific variations*. Regional variations in morphometrics are observed in *C. asper* when only adult specimens are compared, especially between populations of the Western Central Atlantic Ocean (WCAO) and the other two populations in the Western Indian Ocean (WIO) and South-Western Atlantic Ocean (SWAO). For instance, prenarial length (4.2%–5.0% TL for WIO *vs*. 2.9%–3.1% TL for WCAO *vs*. 3.6%–4.6% TL for SWAO). WCAO also exhibit the greatest variations, regarding internarial width, upper labial furrow length, length of first dorsal-fin posterior margin, length of first dorsal-fin spine, height of second dorsal fin, and length of dorsal caudal margin. These specimens also have greater range of values than the WIO specimens for length of preventral caudal margin, pectoral-fin base length, and length of first dorsal-fin inner margin. SWAO specimens vary from those of the WCAO on first dorsal-fin length, length of second dorsal-fin posterior margin, pectoral-fin posterior margin length, and caudal fork width.

This species also exhibits apparent ontogenetic variations in external morphometrics within the WIO population. Changes with growth are observed for precaudal length, pre-second dorsal length, pre-spiracular length, preoral length, distance between nostrils and upper labial furrow, eye length, length of first dorsal fin base, dorsal caudal margin length and clasper length, and length of the nasal barbels. These variations, however, are not observed in the SWAO population and it is unknown for the WCAO population as the current study did not examine young specimens collected from this region.

***Cirrhigaleus australis* White, Last & Stevens, 2007 [3].** Southern Mandarin dogfish

*Cirrhigaleus barbifer*: Garrick and Paul 1971 (in part): 1–13 (revision, description, illustrated; New Zealand) [4]; Last and Stevens 1994 (in part): 48, 68 (cited, description; New Zealand) [112]; Compagno and Niem 1998 (in part): 1203–1224 (listed, cited, illustrated; New Zealand) [77]; Nakabo 2002: 155 (listed; Southern Japan, Ryukyu Islands) [80]; Compagno et al. 2005 (in part): 73 (description; New Zealand) [6]; White et al. 2006: 66, 319 (cited, listed, illustrated; New Zealand) [86]; Nakabo 2013: 194 (listed; New Zealand) [81].

*Squalus barbifer*: Fourmanoir and Rivaton 1979 (in part): 436 (listed; New Zealand) [5].

*Cirrhigaleus australis* White, Last and Stevens, 2007: 19–30 (original description, illustrated; type by original designation; off Bicheno, Tasmania, Australia, New Zealand) [3]; Ebert 2013:

53–54 (listed, cited; Australia, New Zealand) [12]; Naylor et al. 2012: 59 (cited; Australia, New Zealand) [15]; Ebert et al. 2013: 74, 81 (cited, description; Southwest Pacific Ocean) [10]; Kempster et al. 2013: 1–4 (cited; Southern Australia) [7]; Yang et al. 2014: 1–2 (cited; Australia, New Zealand) [113]; Duffy and Last 2015: 125, 127 (listed, description; Australia, New Zealand) [82]; Weigmann 2016: 902 (listed; South-western Pacific Ocean) [88].

*Type material*. CSIRO H 5789–01 (holotype), adult female, 970 mm TL, East of Bicheno, Tasmania, Australia, 41˚55'S,148˚37'E, 360–414 meters depth. Collected on 18 May 2002 by unknown commercial fishermen; AMS I 19154–001 (paratype), juvenile female, 705 mm TL, off Brush Island, New South Wales, Australia, 35˚34'S,150˚45'E, 493 meters depth, collected on 6 July 1976 aboard F.R.V. *Kapala*; AMS I 27022–001 (paratype), adult female, 1205 mm TL, Northeast of Sydney, New South Wales, Australia, 33˚00'S,152˚00'E, 640 meters depth, collected on March 1986 by Ian Ross aboard R/V *Vincenzann*; AMS I 42891–001 (paratype), adult female, 1085 mm TL, Southeast of Green Cape, New South Wales, Australia, 37˚30'S,150˚30'E, 400 meters depth, collected on 3 November 2003 by R. "Smokey" Fanthan aboard R/V *Catherine J.*

*Type locality*: Bicheno, Tasmania, Australia.

*Other material examined (27 specimens)*: AMS I 45670–001, juvenile male, 630 mm TL, Britannia Seamount, New South Wales, Australia, 28˚22'S,155˚37'E; CSIRO H 7042–01, juvenile male, 510 mm TL, Browns Mount, New South Wales, Australia, 34˚02'S,151˚39'E; CSIRO H 7042–02, juvenile female, 639 mm TL, locality same as CSIRO 7042–01; CSIRO H 7042–03, juvenile female, 634 mm TL, locality same as CSIRO 7042–01; CSIRO H 7042–04, juvenile female, 605 mm TL, locality same as CSIRO 7042–01; CSIRO H 7048–01, adult male, 993 mm TL, east of Tweed Heads, New South Wales, Australia, 28˚17'S,153˚53'E; CSIRO H 7064–01, juvenile female, 705 mm TL, Northeast of Yamba, New South Wales, Australia, 29˚11'S,153˚52'E; NMNZ P 5105, juvenile female, 930 mm TL, between Mayor and White Islands, North Island, New Zealand, 37˚25'S,176˚40'E; NMNZ P 5163, adult female, 1045 mm TL, 10 miles off White Island, North Island, New Zealand, 37˚22.32'S,177˚12.83'E; NMNZ P 5204, adult female, 1195 mm TL, off Mayor Island, North Island, New Zealand, 37˚35'S,176˚40'E; NMNZ P 5205, five neonate females, 80–87 mm TL; four neonate males, 83–85 mm TL, same locality as NMNZ P 5204; NMNZ P 7366, adult female, 1272 mm TL, Mayor Island, North Island, New Zealand, 37˚20'S,176˚20'E; NMNZ P 7367, adult female, 1230 mm TL, Mayor Island, North Island, New Zealand, 37˚19'S,176˚15'E; NMNZ P 7681, adult male, 1040 mm TL, Tauranga, North Island, New Zealand, 37˚41'S,176˚10'E; NMNZ P 8030, adult male, 980 mm TL, South Auckland, North Island, New Zealand, 37˚39'S,176˚13.5'E; NMNZ P 17635, adult female, 866 mm TL, Hikurangi Trough, Hawke's Bay, North Island, New Zealand, 39˚58.65'S,178˚8.20'E; NMNZ P 28732, two adult females, 1128–1189 mm TL, New Zealand; NMNZ P 34452, adult male, 990 mm TL, Kermadec Islands, New Zealand, 28˚50.80'S,177˚50.400'W; NMNZ P 34817, juvenile male, 790 mm TL, Kermadec Islands, New Zealand, 33˚4.900'S,179˚34.300'W; NMNZ P 34821, adult male, 971 mm TL, Kermadec Islands, New Zealand, 32˚23.40'S,179˚13.60'W; NMNZ P 38074, adult male, 1020 mm TL, Southern Norfolk Ridge, New Zealand, 32˚40'S,167˚37'W; NMNZ P 42489, juvenile male, 710 mm TL, Tony B Seamount, West of Norfolk Ridge, New Zealand, 34˚31.850'S,168˚47.500'E; NMNZ P 42734, adult female, 1110 mm TL, off "The Faces", Kaikoura, New Zealand, 42˚28'S,173˚46'E; NMNZ P 43052, neonate female, 77 mm TL, same locality as NMNZ P 5204; NMNZ P 46806, juvenile female, 760 mm TL, Three King Islands, Southwest Norfolk Ridge, New Zealand, 34˚36.700'S,168˚56.300'E.

*Diagnosis*. *Cirrhigaleus australis* is distinguished from its congeners by having body light grey dorsally in preserved material (*vs*. dark grey in *C. barbifer vs*. brown in *C. asper*). It is clearly distinct from *C. asper* by having nasal barbel moustache-like and conspicuously

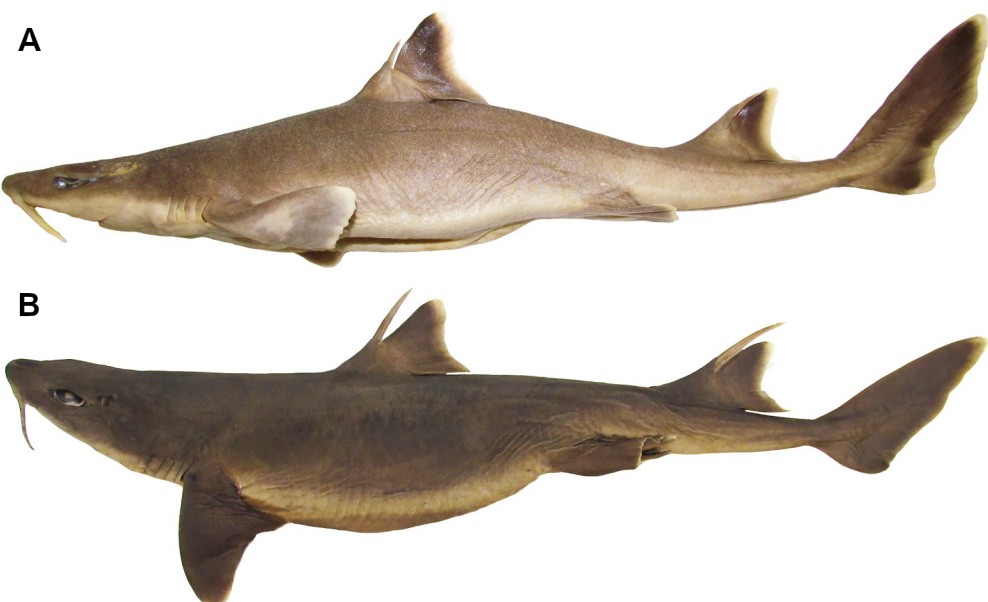

**Fig 27. *Cirrhigaleus australis* in lateral view.** A: CSIRO H 7042–04, juvenile female, 605 mm TL, Australia; B: NMNZ P 34821, adult male, 971 mm TL, New Zealand.

elongate, its length 4.4%–6.2% TL, usually reaching the mouth (*vs.* nasal barbel non mous-tache-like and conspicuously short, its length 1.0%–1.5% TL, never reaching the mouth in *C. asper*), and dermal denticles bat-like with narrow median ridge (*vs.* heart-shaped with broad median ridge in *C. asper*). It further differs from *C. asper* by smaller length of upper labial fur-row, its length 0.9%–1.4% TL (*vs.* 1.4%–2.1% TL). *Cirrhigaleus australis* is separated from *C. barbifer* by having teeth with narrow apron (*vs.* broad in *C. barbifer*), and dermal denticles with one cusplet lateral-posteriorly on each side (*vs.* two cusplets in *C. barbifer*) (Fig 27).

*Description.* Morphometric and meristic data are shown in Table 3 and S3 and S4 Tables.

*External morphology.* Body trihedral, strongly robust and humped in all its extension, con-spicuously arched from posterior margin of the eye to pelvic fin insertion with belly noticeable ventrally (more slender when juveniles than in adults); body equally wide from head to abdo-men with head width 1.0 (0.9–1.2) times trunk width, and 1.0 (1.0–2.1) times abdomen width; body deepest at trunk and abdomen with head height 0.7 (0.6–1.5) times trunk height, and 0.7 (0.7–1.5) times abdomen height. Head very small, its length 20.0% (18.2%–23.1%) TL, flat-tened and narrow anteriorly, humped and broader posteriorly near the mouth; head width at nostrils 7.1% (6.4%–7.7%) TL, and at mouth 11.8% (10.4%–13.6%) TL. Eyes oval and elongate, its length 3.3% (3.1%–4.6%) TL and corresponding to 2.1 (2.1–5.5) times its height; anterior margin of eye convex, posterior margin slightly notched, prominent dorsally. Spiracles cres-cent and wide, its length 0.8 (0.8–1.8) times eyes height, placed lateral-posteriorly and above the eyes. Gill slits vertical and slightly concave, tall with height of fifth gill slit 2.3% (1.9%–2.7%) TL, corresponding to 1.3 (0.9–1.4) times first gill slit height. Snout very short, preorbital length 6.5% (5.8%–7.2%) TL and conspicuously rounded at the tip; nostrils equally near to snout tip and upper labial furrow, its prenarial length 0.9 (0.9–1.4) times distance from nostrils to upper labial furrow; anterior nasal flap bilobate with second lobe markedly elongate and moustache-like as nasal barbels, reaching anterior margin of mouth in adults (young speci-mens have nasal barbels markedly beyond posterior margin of the mouth); anterior nasal flap length 5.0% (4.4%–7.1%) TL, corresponding to 4.0 (2.6–5.1) times in head length.

**Table 3. External measurements of *C. australis* expressed as percentage of total length (%TL).**

| | H | P1 | P2 | P3 | n | Australia | | | n | New Zealand | | | x | SD |
|---|---|---|---|---|---|---|---|---|---|---|---|---|---|---|
| | | | | | | | | | | | | | | |
| TL (mm) | 970.0 | 705.0 | 1205.0 | 1085.0 | 7 | 505.0 | – | 993.0 | 17 | 710.0 | – | 1272.0 | 925.1 | 220.5 |
| PCL | 98.8 | 78.9 | 81.9 | 81.7 | 7 | 77.7 | – | 80.0 | 17 | 78.5 | – | 86.2 | 80.5 | 2.1 |
| PD2 | 61.3 | 61.8 | 64.1 | 63.1 | 7 | 59.4 | – | 61.8 | 17 | 61.0 | – | 68.4 | 63.0 | 2.1 |
| PD1 | 32.0 | 31.5 | 33.2 | 30.9 | 7 | 29.7 | – | 32.5 | 17 | 30.3 | – | 43.3 | 32.3 | 2.4 |
| SVL | 54.1 | 51.1 | 59.3 | 56.2 | 7 | 51.5 | – | 56.6 | 17 | 53.2 | – | 59.5 | 55.3 | 2.2 |
| PP2 | 50.5 | 49.1 | 54.8 | 53.0 | 7 | 47.9 | – | 53.1 | 17 | 49.7 | – | 55.0 | 51.6 | 1.8 |
| PP1 | 19.1 | 19.3 | 19.9 | 20.1 | 7 | 17.3 | – | 20.5 | 17 | 18.2 | – | 22.4 | 19.8 | 1.0 |
| HDL | 20.0 | 21.3 | 20.2 | 22.6 | 7 | 18.2 | – | 22.0 | 17 | 18.9 | – | 23.1 | 20.6 | 1.1 |
| PG1 | 17.1 | 18.4 | 16.4 | 17.7 | 7 | 15.8 | – | 17.7 | 17 | 15.9 | – | 18.7 | 17.1 | 0.8 |
| PSP | 10.1 | 11.1 | 9.5 | 9.9 | 7 | 10.1 | – | 11.5 | 17 | 9.9 | – | 11.4 | 10.5 | 0.6 |
| POB | 6.5 | 7.2 | 6.0 | 5.8 | 7 | 5.9 | – | 7.2 | 17 | 6.0 | – | 7.3 | 6.5 | 0.4 |
| PRN | 4.0 | 4.6 | 3.9 | 4.1 | 7 | 4.2 | – | 4.9 | 17 | 4.0 | – | 5.0 | 4.5 | 0.3 |
| POR | 7.3 | 8.0 | 6.9 | 7.8 | 7 | 7.7 | – | 8.6 | 17 | 7.0 | – | 8.3 | 7.7 | 0.5 |
| INLF | 4.3 | 4.5 | 4.0 | 4.5 | 7 | 4.1 | – | 4.9 | 17 | 4.0 | – | 4.9 | 4.4 | 0.3 |
| MOW | 7.6 | 7.9 | 6.6 | 7.4 | 7 | 7.5 | – | 8.5 | 17 | 7.2 | – | 8.8 | 7.8 | 0.4 |
| ULA | 0.9 | 1.0 | 0.9 | 0.9 | 7 | 1.2 | – | 1.6 | 17 | 1.0 | – | 1.4 | 1.2 | 0.2 |
| INW | 3.6 | 3.6 | 3.8 | 3.9 | 7 | 3.5 | – | 4.1 | 17 | 3.4 | – | 4.0 | 3.7 | 0.2 |
| INO | 7.8 | 8.5 | 7.5 | 8.0 | 7 | 7.8 | – | 8.4 | 17 | 7.2 | – | 8.5 | 7.9 | 0.3 |
| EYL | 3.3 | 3.9 | 3.3 | 3.1 | 7 | 3.7 | – | 4.4 | 17 | 3.3 | – | 4.6 | 3.9 | 0.4 |
| EYH | 1.5 | 1.8 | 1.1 | 1.0 | 7 | 1.1 | – | 1.6 | 17 | 0.7 | – | 1.6 | 1.3 | 0.3 |
| SPL | 1.3 | 1.7 | 1.3 | 1.6 | 7 | 1.3 | – | 1.8 | 17 | 1.1 | – | 1.6 | 1.4 | 0.2 |
| GS1 | 1.8 | 2.3 | 1.7 | 1.8 | 7 | 2.0 | – | 2.3 | 17 | 1.6 | – | 2.3 | 2.1 | 0.2 |
| GS5 | 2.3 | 2.4 | 2.3 | 2.2 | 7 | 2.0 | – | 2.6 | 17 | 1.9 | – | 2.7 | 2.3 | 0.2 |
| IDS | 22.3 | 22.7 | 23.2 | 21.8 | 7 | 21.0 | – | 23.0 | 17 | 21.2 | – | 26.5 | 22.8 | 1.2 |
| DCS | 7.5 | 8.0 | 8.0 | 8.6 | 7 | 7.8 | – | 9.1 | 17 | 6.9 | – | 9.1 | 8.0 | 0.6 |
| PPS | 29.4 | 28.4 | 30.3 | 28.1 | 7 | 26.8 | – | 32.0 | 17 | 26.8 | – | 35.0 | 29.9 | 2.2 |
| PCA | 19.8 | 20.0 | 21.1 | 21.2 | 7 | 20.7 | – | 23.8 | 17 | 17.6 | – | 22.9 | 20.7 | 1.1 |
| D1L | 14.4 | 13.6 | 14.6 | 15.3 | 7 | 13.5 | – | 15.0 | 17 | 13.6 | – | 15.5 | 14.4 | 0.6 |
| D1A | 12.2 | 13.9 | 13.6 | 13.7 | 7 | 11.7 | – | 14.6 | 17 | 11.9 | – | 14.4 | 13.3 | 0.7 |
| D1B | 8.4 | 8.6 | 8.2 | 8.8 | 7 | 7.7 | – | 9.2 | 17 | 8.4 | – | 9.8 | 8.7 | 0.5 |
| D1H | 9.8 | 9.7 | 8.9 | 10.9 | 7 | 9.5 | – | 10.2 | 17 | 9.2 | – | 11.1 | 9.9 | 0.5 |
| D1I | 6.3 | 4.9 | 6.2 | 6.8 | 7 | 5.3 | – | 6.5 | 17 | 5.2 | – | 6.6 | 5.8 | 0.4 |
| D1P | 10.6 | 9.6 | 9.8 | 11.4 | 7 | 9.4 | – | 11.3 | 17 | 9.0 | – | 11.9 | 10.2 | 0.8 |
| D1ES | 5.7 | 5.2 | – | 4.7 | 6 | 3.1 | – | 6.4 | 11 | 3.7 | – | 6.8 | 4.9 | 1.1 |
| D1BS | 0.9 | 1.0 | 1.1 | 0.9 | 6 | 0.6 | – | 0.9 | 17 | 0.5 | – | 1.1 | 0.9 | 0.1 |
| D2L | 14.9 | 13.9 | 12.4 | 15.5 | 7 | 13.8 | – | 15.8 | 17 | 13.3 | – | 15.8 | 14.5 | 0.8 |
| D2A | 14.2 | 14.5 | 13.5 | 14.8 | 7 | 13.6 | – | 14.8 | 17 | 13.1 | – | 14.9 | 14.2 | 0.4 |
| D2B | 9.6 | 9.2 | 10.0 | 9.3 | 7 | 9.0 | – | 10.1 | 17 | 8.8 | – | 10.5 | 9.6 | 0.5 |
| D2H | 9.5 | 8.8 | 9.4 | 9.7 | 7 | 8.7 | – | 9.5 | 17 | 8.3 | – | 10.8 | 9.1 | 0.5 |
| D2I | 5.5 | 4.7 | 4.5 | 6.3 | 7 | 4.8 | – | 5.8 | 17 | 4.4 | – | 5.6 | 5.1 | 0.4 |
| D2P | 9.1 | 8.8 | 7.2 | 8.5 | 7 | 8.4 | – | 10.6 | 17 | 7.2 | – | 9.1 | 8.4 | 0.7 |
| D2ES | 6.4 | 7.6 | 5.9 | 6.7 | 7 | 5.5 | – | 8.1 | 14 | 4.4 | – | 9.3 | 6.6 | 1.2 |
| D2BS | 1.1 | 1.1 | 1.0 | 1.1 | 7 | 0.8 | – | 1.2 | 17 | 0.8 | – | 1.3 | 1.0 | 0.1 |
| P1A | 15.2 | 14.8 | 16.1 | 15.5 | 7 | 14.3 | – | 16.9 | 17 | 14.1 | – | 16.7 | 15.5 | 0.8 |
| P1I | 8.5 | 8.1 | 8.9 | 8.9 | 7 | 8.1 | – | 9.5 | 17 | 7.5 | – | 8.8 | 8.4 | 0.5 |
| P1B | 5.0 | 4.7 | 4.5 | 5.1 | 7 | 4.4 | – | 4.9 | 17 | 4.2 | – | 5.8 | 4.8 | 0.4 |

*(Continued)*

**Table 3.** (Continued)

| | H | P1 | P2 | P3 | n | Australia | | | n | New Zealand | | | x | SD |
|---|---|---|---|---|---|---|---|---|---|---|---|---|---|---|
| | | | | | | *Cirrhigaleus australis* | | | | | | | | |
| P1P | 10.2 | 11.7 | 11.2 | 11.8 | 7 | 10.8 | – | 12.4 | 17 | 8.3 | – | 12.1 | 10.9 | 1.0 |
| P2L | 12.9 | 11.7 | 13.6 | 13.3 | 7 | 11.1 | – | 13.1 | 17 | 11.9 | – | 14.8 | 12.9 | 0.7 |
| P2I | 6.0 | 5.3 | 6.7 | 5.7 | 7 | 5.3 | – | 6.8 | 17 | 5.0 | – | 8.3 | 6.0 | 0.8 |
| CDM | 21.6 | 21.8 | 19.2 | 21.2 | 7 | 20.1 | – | 21.8 | 16 | 18.0 | – | 21.3 | 20.2 | 1.1 |
| CPV | 11.7 | 11.4 | 10.5 | 12.2 | 7 | 10.0 | – | 12.4 | 17 | 9.4 | – | 11.2 | 10.7 | 0.8 |
| CFW | 7.6 | 8.0 | 7.5 | 7.9 | 7 | 7.2 | – | 8.6 | 17 | 7.4 | – | 8.4 | 7.8 | 0.3 |
| HANW | 7.1 | 7.5 | 6.5 | 7.2 | 7 | 6.5 | – | 7.7 | 17 | 6.4 | – | 7.4 | 7.0 | 0.4 |
| HAMW | 11.8 | 12.6 | 10.4 | 11.6 | 7 | 11.5 | – | 13.6 | 17 | 11.2 | – | 13.2 | 12.0 | 0.7 |
| HDW | 15.9 | 15.1 | 22.8 | 21.4 | 7 | 14.0 | – | 15.5 | 17 | 13.1 | – | 16.8 | 15.4 | 2.1 |
| TRW | 16.5 | 12.9 | 21.0 | 23.3 | 7 | 12.3 | – | 14.8 | 17 | 11.5 | – | 16.4 | 14.3 | 2.6 |
| ABW | 15.5 | 9.6 | 15.8 | 16.1 | 7 | 8.0 | – | 14.5 | 17 | 7.3 | – | 12.6 | 10.9 | 2.5 |
| HDH | 12.8 | 11.3 | 12.0 | 16.1 | 7 | 9.6 | – | 11.8 | 17 | 8.2 | – | 12.0 | 10.8 | 1.5 |
| TRH | 17.5 | 12.2 | 12.9 | 10.9 | 7 | 10.6 | – | 12.8 | 17 | 6.3 | – | 17.7 | 11.7 | 2.5 |
| ABH | 17.4 | 14.0 | 15.4 | 10.9 | 7 | 9.3 | – | 13.8 | 17 | 7.8 | – | 17.0 | 12.6 | 2.7 |
| CLO | – | – | – | – | 3 | 1.2 | – | 4.4 | 7 | 1.2 | – | 10.0 | 3.6 | 2.6 |
| CLI | – | – | – | – | 3 | 3.5 | – | 7.2 | 7 | 3.2 | – | 8.8 | 5.6 | 2.1 |
| ANFL | 5.0 | 5.9 | 5.2 | 4.4 | 6 | 5.3 | – | 7.1 | 17 | 4.5 | – | 6.2 | 5.6 | 0.7 |

Range values for other material are provided. TL is expressed in millimeter. H: holotype, CSIRO H 5789–01; P1: paratype, AMS I 19154–001; P2: paratype, AMS I 27022–001; P3: AMS I 42891–001. n: number of specimens; x: mean; SD: standard deviation.

Short distance from mouth to snout tip, preoral length 7.3% (6.9%–8.6%) TL. Mouth strongly arched and wide, its width 1.9 (1.6–2.0) times prenarial length, and 2.2 (1.8–2.3) times broader than internarial space. Upper labial furrow markedly short, its length 0.9% (0.9%–1.6%) TL, with thick fold; lower labial furrow elongate, lacking a fold. Teeth similar in both jaws, compressed labial-lingually, conspicuously broad and low (Fig 28); teeth of upper jaw much smaller than those of lower jaw; teeth unicuspid with cusp pointed, thin, cylindrical and short, directed laterally; mesial cutting edge somewhat convex, although diagonally; mesial and distal heels rounded; apron narrow and short, larger on lower teeth than upper teeth. Three series of functional teeth on both jaws; upper jaw with 12–12 (13–13) teeth rows; lower jaw with 10–10 (11–11) teeth rows.

Dorsal fins conspicuously upright and vertical, both equally tall with first dorsal fin height 1.0 (0.9–1.2) times height of second dorsal fin; first dorsal fin as large as second dorsal fin, its length 1.0 (0.9–1.2) times length of second dorsal fin. Origin of first dorsal fin beyond vertical traced at the pectoral-fin free rear tips in adults. First dorsal fin markedly slender at fin web, apex rounded, and broad on its base, base length 8.4% (7.7%–9.8%) TL; first dorsal-fin anterior margin convex, raked and large, its length 12.2% (11.7%–14.6%) TL; first dorsal-fin posterior margin concave and elongate, its length 10.6% (9.0%–11.9%) TL; first dorsal-fin inner margin short, its length 6.3% (4.9%–6.8%) TL with pointed free rear tip; first dorsal fin height 1.6 (1.4–2.0) times greater than first dorsal-fin inner margin length, and 1.5 (1.3–1.9) times preorbital length (Fig 29). First dorsal-fin spine straight, markedly elongate, although never reaching the fin apex, its length 5.7% (3.1%–6.8%) TL, corresponding to two-fourth the height of first dorsal fin. Interdorsal distance 1.2 (1.0–1.3) times pre-pectoral length, and 3.0 (2.3–3.5) times greater than dorsal-caudal space.

Origin of second dorsal fin posterior to vertical traced at pelvic-fin free rear tips. Second dorsal fin slender at the fin web, apex rounded and lobulate, broad at its base, base length 9.6%

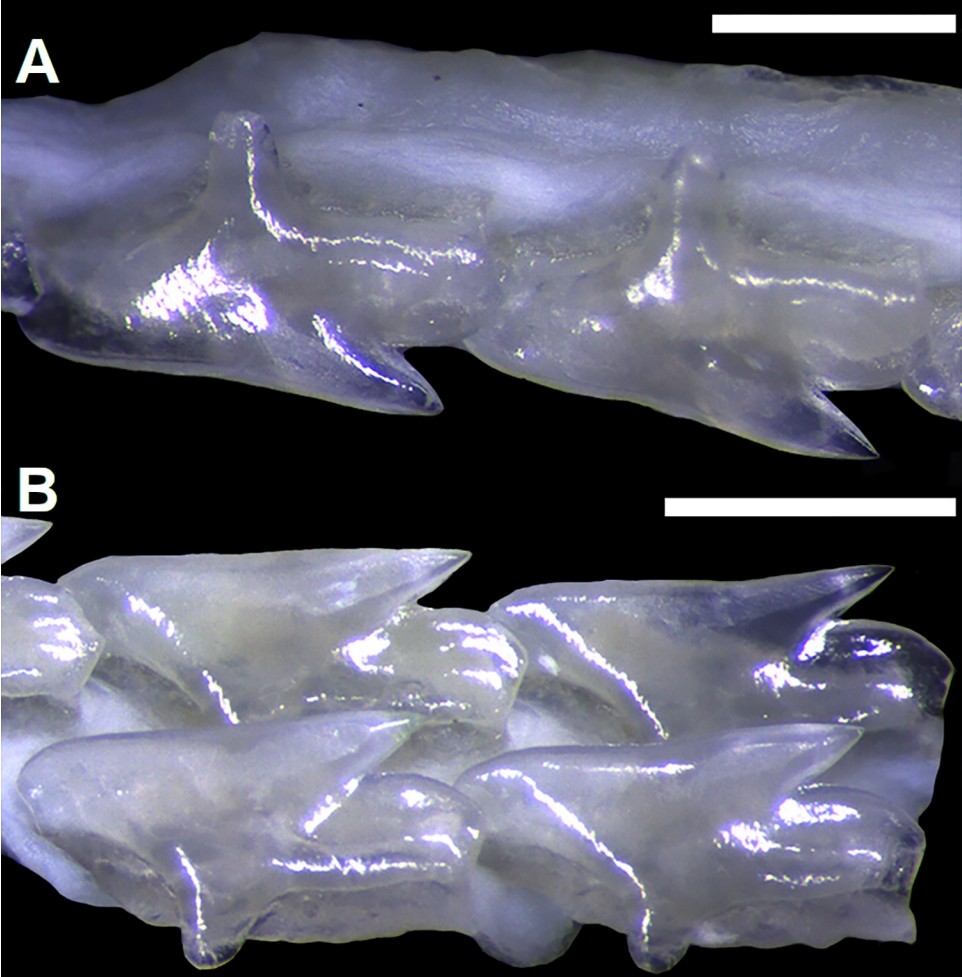

**Fig 28.** Upper (A) and lower (B) teeth (labial view) of *C. australis*, AMS I 45670–001, juvenile male, 630 mm TL. Scale bars: 1 mm.

(9.0%–10.5%) TL; second dorsal-fin anterior margin convex and large, its length 14.2% (13.5%–14.9%) TL; second dorsal-fin posterior margin strongly concave, falcate (half-moon shaped in juveniles); second dorsal-fin height 1.7 (1.5–2.2) times greater than second dorsal-fin inner margin length; second dorsal-fin inner margin also short, its length 5.5% (4.4%–6.3%) TL. Second dorsal-fin spine convex and broad, its base width 1.1% (0.8%–1.3%) TL, conspicuously elongate, its length 6.4% (4.4%–9.3%) TL; second dorsal-fin spine corresponding to 1.1 (0.7–2.1) times length of first dorsal-fin spine and three-forth the height of second dorsal fin, exceeding the fin apex in juveniles only.

Pectoral fins narrow (Fig 30A), although broader posteriorly than anteriorly, its base length 5.0%, (4.4%–5.8%) TL; pectoral-fin anterior and inner margins markedly convex; pectoral-fin posterior margin slightly concave, its length never greater than the trunk height when adpressed laterally, corresponding to 0.6 (0.6–1.7) times trunk height; pectoral-fin apex and free rear tips rounded and wide (sometimes lobulate in paratypes) with apex transcending horizontal line traced at pectoral-fin free rear tip; pectoral-fin anterior margin length 1.5 (1.2–1.9) times pectoral-fin posterior margin length, and 1.8 (1.6–2.0) times pectoral-fin inner margin length.

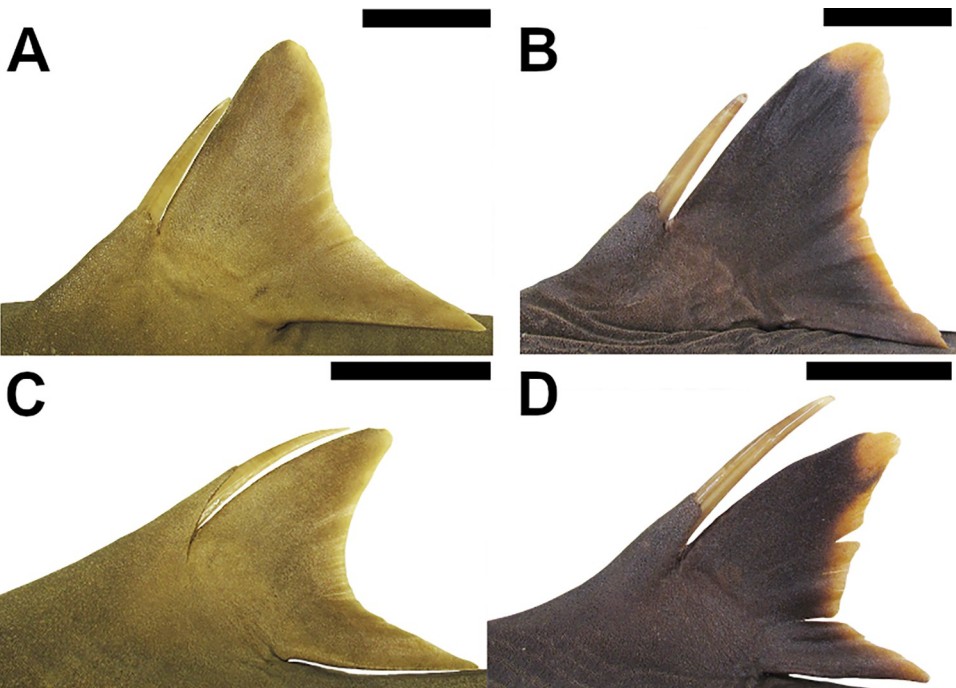

**Fig 29.** First (A,B) and second (C,D) dorsal fins of *C. australis*. A,C: AMS I 42891–001 (paratype), adult female, 1085 mm TL, Australia; B,D: NMNZ P 28732, adult female, 1189 mm TL, New Zealand. Scale bars: 20 mm.

Pelvic fins (Fig 30B) nearest to second dorsal fin than first dorsal fin; length of pelvic fin 1.7 (1.6–2.1) times interdorsal space. Pelvic fins conspicuously broad and pentagonal; pelvic-fin anterior margin slightly convex; pelvic-fin inner margin straight; pelvic-fin posterior margin somewhat concave; pelvic-fin apex markedly rounded; pelvic-fin free rear tips triangular and lobe-like; pelvic length 1.1 (1.0–1.5) times length of preventral caudal margin. Claspers (Fig 31) in adults flattened ventrally and markedly wide, narrower distally; claspers very short its inner length 0.5–1.3 times pelvic inner margin length, exceeding pelvic-fin free rear tips; clasper groove elongate, vertical, placed dorsally; apopyle with narrow aperture, located anteriorly in the clasper groove; hypopyle with broad aperture, placed distally in the clasper groove, prior to the rhipidion; rhipidion flap-like, slender and large, attached mediodistally to clasper.

Pelvic-caudal space 0.9 (0.8–1.0) times interdorsal space. Caudal peduncle thick in cross-section and conspicuously short with dorsal-caudal distance 7.5% (6.9%–9.1%) TL, corresponding to 3.0 (2.3–3.5) times smaller than interdorsal space; lateral precaudal keel prominent almost as a fold, since second dorsal fin insertion to backwards origin of caudal fin; upper and lower precaudal pits absent. Caudal fin (Fig 30C) conspicuously rectangular; dorsal caudal margin straight and elongate, its length 1.1 (0.9–1.2) times head length and 1.8 (1.7–2.2) times greater than length of preventral caudal margin; dorsal caudal tip rounded; upper post-ventral caudal margin convex, although evidently straight proximally; lower post-ventral caudal margin convex; preventral caudal margin somewhat straight and small, its length 2.0 (1.3–2.3) times greater than length of pelvic-fin inner margin; ventral caudal tip rounded; transition between upper and lower caudal lobes continuous; caudal fork markedly broad, its width 7.6% (7.2%–8.6%) TL.

*Dermal denticles.* Dermal denticles (Fig 12D–12H) tricuspid and bat-like, imbricate, large and conspicuously broad at the crown; dermal denticles with its width equivalent to its length; cusps pointed and located posteriorly at the crown margin; median cusp much larger than

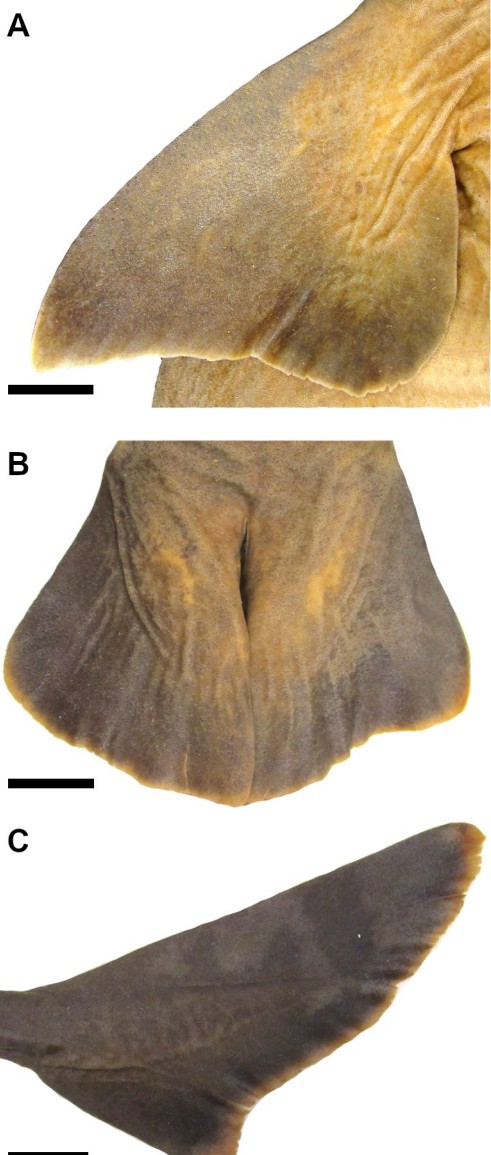

**Fig 30.** Pectoral (A), pelvic (B) and caudal (C) fins of *C. australis* (NMNZ P 28732, adult female, 1189 mm TL) in ventral (A,B) and lateral (C) views. Scale bars: 20 mm.

lateral ones; median ridge elongate, thin, tall and straight, slightly transcending anterior base of crown; lateral ridges thin, short and low, markedly convex proximally; median furrow small and shallow, placed proximally over median ridge; lateral furrow short and deep, located aside each lateral ridge; one secondary cusp (or cusplet) sometimes present on each side of posterior margin of the crown; cusplets much smaller than median cusp.

*Colouration.* Body light grey dorsal and laterally, white ventrally (Fig 27). Nasal barbels white, and slightly grey lateral-proximally. First dorsal fin dark grey with posterior margin broadly white from its apex to free rear tip, submarginal bar light grey, white spots in the anterior margin near its apex. Second dorsal fin dark grey with posterior margin broadly white until its midline, and submarginal bar light grey. Dorsal-fin spines white, greyish proximal and distally, although whitish at the tips. Pectoral fins grey dorsally and light grey ventrally, with

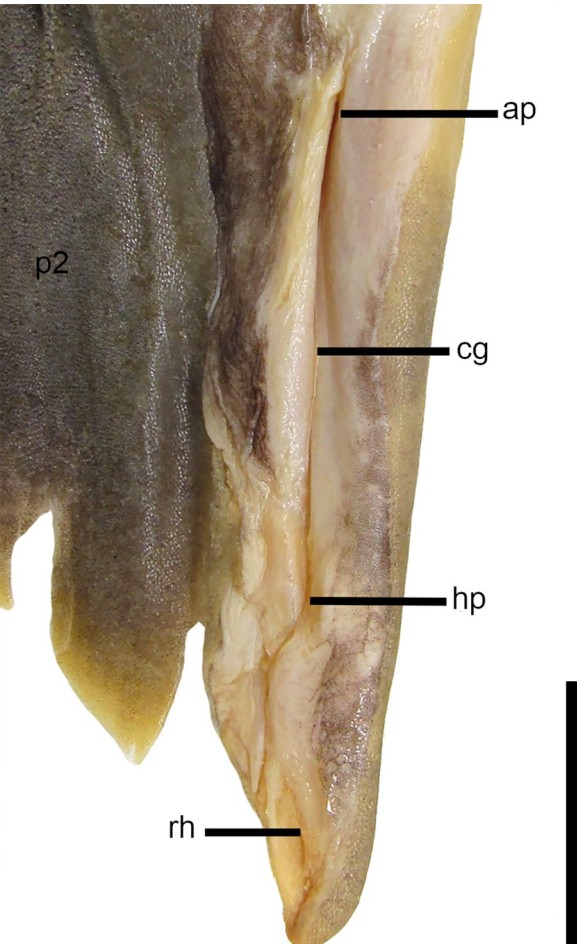

**Fig 31. Clasper (left side) of *C. australis*, NMNZ P 38074, adult male, 1020 mm TL in dorsal view.** Abbreviations: ap: apopyle; cg: clasper groove; hp: hypopyle; p2: pelvic fin; rh: rhipidion. Scale bar: 20 mm.

both inner and posterior margins discreetly whitish at the edge. Pelvic fins also grey, lighter ventrally, with posterior margin sparsely whitish. Caudal fin dark grey, whitish medially; post-ventral caudal margins white but not uniform, broadly white in the dorsal and ventral caudal tips; subterminal bar blackish; black caudal stripe absent. In recent collected specimens, large dark grey blotches are observed ventrally in both pectoral and pelvic fins.

*Vertebral counts*. Monospondylous vertebrae 51 (49–53); precaudal vertebrae 85 (85–90); total vertebrae 117 (113–119).

*Geographical distribution*. Cirrhigaleus australis occurs in the South-western Pacific Ocean and it is endemic to New Zealand and Southern Australia from Tweed Heads in New South Wales to Tasmania. It is caught between 91–1103 meters depth, and more often between 540–580 meters (Fig 15).

**Remarks.** *Intraspecific variations*. Snout and nasal barbels are smaller in adults than in juveniles of *C. australis* from Australia (AUS) but not in the New Zealand (NZ) population (prenarial length 3.9%–4.2% TL for adults *vs*. 4.5%–4.9% TL for juveniles; length of anterior margin of nostrils 4.4%–5.3% TL for adults *vs*. 5.9%–7.1% TL for juveniles). Clasper length also varies with maturity in both populations (clasper outer length 1.2%–1.5% TL for juveniles *vs*. 4.4% in adults for AUS; 3.7%–10.0% and 1.2%–1.7% for NZ, respectively; clasper inner

length 3.5%–3.9% TL in juveniles *vs*. 7.2% TL in adults for AUS; 4.3%–8.8% and 3.2%–3.6% for NZ). Other variations are observed here for the first time, including position of origin of first dorsal fin varies from over or just anterior to the vertical traced at the pectoral-fin free rear tips in juveniles while it is behind it in adults.

*C. australis* exhibits morphometric and morphological variations between the New Zealand (NZ) and Australian (AUS) populations, such as: head longer and flattened dorsally in the NZ specimens (*vs*. shorter and arched in the AUS specimens); snout more obtuse (*vs*. snout rounded); dorsal fins broad at the fin web with straight posterior margin (*vs*. dorsal fins thin at fin web with strongly concave posterior margins); second dorsal-fin spine transcending second dorsal-fin apex (*vs*. not transcending second dorsal-fin apex); narrower abdomen when adults (7.3%–12.6% TL for NZ *vs*. 13.1%–16.1% TL for AUS, respectively).

It is also noticed that specimens from New Zealand exhibit intermediary characteristics, concerning body colour, dentition and shape of dermal denticles, shape of dorsal fins, length of dorsal-fin spines related to dorsal-fin apex, and origin of first dorsal fin related to pectoral-fin free rear tips, suggesting that more than one morphological group of *Cirrhigaleus* occurs in this region. One group has greyish body, origin of first dorsal fin prior to a vertical line traced at pectoral-fin free rear tips, first dorsal-fin spine never reaching first dorsal-fin apex, and caudal fin with posterior caudal tip narrow. The second group has dark brown body, origin of first dorsal fin behind a vertical line traced at pectoral-fin free rear tips, first dorsal-fin spine transcending first dorsal-fin apex, and caudal fin with posterior caudal tip broad. These specimens also have greater range of values of external measurements (e.g., precaudal length; pre-first and pre-second dorsal length; interdorsal space; dorsal-caudal distance; pectoral-pelvic distance; pelvic-caudal distance; size of first and second dorsal fins; length of dorsal-fin spines) and vertebral counts, which give additional support for this hypothesis.

## Internal morphological description of *Cirrhigaleus* species

*Neurocranium* (Table 4) (Fig 2A and 2B; Fig 4A and 4B). In *C. barbifer*, neurocranium very thick, broader anteriorly across nasal capsules than posteriorly across postorbital processes (width across nasal capsules 68.2% CL *vs*. width across postorbital processes 62.2% CL in *C. barbifer*; 62.5%–70.5% CL and 55.1%–71.9% CL in *C. asper*; 68.0% and 59.7% CL in *C. australis*); neurocranium narrower in the interorbital region (its width 37.5% CL in *C. barbifer*, 30.6%–34.3% CL in *C. asper* and 33.1% CL in *C. australis*) and posteriorly in the occipital region (width across opistotic processes 41.6% CL in *C. barbifer*, 39.8%–43.6% CL in *C. asper* and 43.0% CL in *C. australis*). Precerebral fossa very deep distally, and shallow proximally. Prefrontal fontanelle markedly narrow. Rostrum spoon-like and very short (its length 39.9% CL in *C. barbifer*, 29.0%–34.1% CL in *C. asper* and 38.7% CL in *C. australis*), uniform on its extension with somewhat cylindrical lateral cartilages, rounded at the tip; rostral keel prominent, although markedly short (its length 22.6% CL), not reaching the anterior margin of the nasal capsules. Nasal capsules conspicuously large, oval dorsally and rounded ventrally, slightly oblique; many large nasal foramina evident lateral and ventrally in each nasal capsule, rarely in dorsal view; nasal capsule cracks present ventrally. Subnasal fenestra oval, slender and small on each side of the rostral keel at the base of prefrontal fontanelle. Ethmoidal region strongly narrow; epiphysial pit large and rounded, located anterior-dorsally just posteriorly to the prefrontal fontanelle with thick anterior crest. Ectethmoidal chamber markedly narrow ventrally, and antorbital cartilage tapered, continuous and convex at its distal margin. Subethmoidal region conspicuously short and slender, its width 15.2% CL in *C. barbifer*, 17.0%–18.4% CL in *C. asper* and 17.8% CL in *C. australis*); subethmoidal ridge prominent.

**Table 4. Cranial measurements of *C. barbifer*, *C. asper* and *C. australis* expressed as percentage of total length of the neurocranium (% CL).**

| Measurements | | *C. barbifer* | | *C. asper* | | | | | *C. australis* | |
|---|---|---|---|---|---|---|---|---|---|---|
| | | N | *x* | N | Range | | *x* | SD | N | *x* |
| 1 | total length (mm) | 1 | 71.7 | 6 | 37.6 – | 123.5 | 99.3 | 31.5 | 1 | 78.2 |
| 2 | postcerebral length | 1 | 60.8 | 6 | 67.1 – | 73.1 | 69.8 | 2.0 | 1 | 62.9 |
| 3 | precerebral fenestra length | 1 | 39.9 | 6 | 29.0 – | 34.1 | 30.8 | 2.1 | 1 | 38.7 |
| 4 | precerebral fenestra width | 1 | 14.9 | 6 | 14.5 – | 20.7 | 17.3 | 2.1 | 1 | 14.6 |
| 5 | width across nasal capsules | 1 | 68.2 | 6 | 62.5 – | 70.5 | 65.1 | 3.2 | 1 | 68.0 |
| 6 | interorbital width | 1 | 37.5 | 6 | 30.6 – | 34.3 | 31.9 | 1.7 | 1 | 33.1 |
| 7 | width across preorbital processes | 1 | 60.8 | 6 | 57.2 – | 64.5 | 59.7 | 2.7 | 1 | 53.5 |
| 8 | postorbital process length | 1 | 13.4 | 6 | 11.2 – | 14.4 | 13.2 | 1.2 | 1 | 10.0 |
| 9 | width across postorbital processes | 1 | 62.2 | 6 | 55.1 – | 71.9 | 65.2 | 5.5 | 1 | 59.7 |
| 10 | distance between orbital processes | 1 | 34.0 | 6 | 36.0 – | 43.8 | 39.9 | 2.8 | 1 | 32.2 |
| 11 | distance across opistotic processes | 1 | 41.6 | 6 | 39.8 – | 43.6 | 41.7 | 1.6 | 1 | 43.0 |
| 12 | width across hyomandybular facets | 1 | 50.6 | 5 | 50.4 – | 56.0 | 59.6 | 2.3 | 1 | 48.3 |
| 13 | nasobasal length | 1 | 66.4 | 6 | 66.6 – | 75.5 | 70.6 | 4.0 | 1 | 65.7 |
| 14 | rostral keel length | 1 | 22.6 | 6 | 16.9 – | 30.4 | 22.0 | 5.5 | 1 | 22.8 |
| 15 | subethmoidean width | 1 | 15.2 | 6 | 17.0 – | 18.4 | 17.6 | 0.5 | 1 | 17.8 |
| 16 | basal angle width | 1 | 25.5 | 6 | 17.8 – | 29.4 | 24.9 | 3.8 | 1 | 18.8 |
| 17 | basal plate length | 1 | 51.5 | 6 | 44.0 – | 50.3 | 47.2 | 2.2 | 1 | 41.6 |
| 18 | basal plate width | 1 | 25.0 | 6 | 21.8 – | 30.5 | 25.8 | 2.9 | 1 | 20.3 |
| 19 | width across first cartilaginous process | 1 | 36.0 | 5 | 35.4 – | 42.8 | 43.3 | 2.7 | 1 | 33.5 |
| 20 | width across second cartilaginous process | – | – | 6 | – – | – | – | – | | |
| 21 | maximum sagital length | 1 | 22.6 | 6 | 20.5 – | 26.5 | 22.6 | 2.3 | 1 | 22.9 |
| 22 | foramen magnum width | 1 | 7.9 | 6 | 7.6 – | 15.5 | 10.8 | 2.5 | 1 | 4.1 |
| 23 | subethmoidal ridge length | – | – | 5 | 26.5 – | 76.4 | 38.3 | 21.4 | – | – |

CL: expressed in millimeter. N: number of specimens; *x*: mean; SD: standard deviation.

Cranial roof rather flattened with superficial longitudinal sulcus, carrying 10 tiny foramina of the branch superficial ophthalmic of the trigeminal and facial nerves (V–VII) plus preorbital canal; the latter conspicuously large and rounded, placed in front of the series of foramina for the ophthalmic nerve (VII); ethmoidal canal large, located well anteriorly in the base of the nasal capsule; *profundus* canal for the deep ophthalmic of the trigeminal nerve (V) with two apertures, one dorsal, very small and rounded, and another lateral, small and rounded, located in the interorbital wall just before the preorbital canal; dorsal aperture of the *profundus* canal placed between the ethmoidal canal and preorbital canal. Interorbital region concave with supraorbital crest slender and C-shaped; distance between orbital processes very small, its length 34.0% CL in *C. barbifer*, 36.0%–43.8% CL in *C. asper* and 32.2% CL in *C. australis*); pre-orbital process inconspicuous; postorbital process triangular and small, its length 13.4% CL in *C. barbifer*, 11.2%–14.4% CL in *C. asper* and 10.0% CL in *C. australis*); width across postorbital processes slightly greater than the width across preorbital processes (the former 62.2% CL and the latter 60.8% CL in *C. barbifer*, 55.1%–71.9% CL and 57.2%–64.5% CL in *C. asper*, and 59.7% and 53.5% CL in *C. australis*).

Preorbital wall convex and short. Interorbital wall profound with eye stalk small and wide, carrying distal disc, and located more posteriorly; foramen optic (II) very large and rounded, placed more ventral-anteriorly near the preorbital wall; foramen trochlear (IV) dorsally near

the series of foramina of the branch superficial ophthalmic of the trigeminal nerve (V–VII); foramen oculomotor (III) just above the eye stalk while foramen abducens (VI) is near its base; trigeminal (V) and facial (VII) nerves with a common aperture, the foramen prooticum, posteriorly in the orbital wall; foramen prooticum also opens lateral- posteriorly to a branch hyomandibular of facial nerve (VII) in the hyomandibular facet through two small apertures. Basal angle prominent (its width 25.5% CL in *C. barbifer*, 17.8%–29.4% CL in *C. asper* and 18.8% CL in *C. australis*), located ventral-posteriorly in the interorbital wall for supporting the orbital articulation; foramen for an efferent of the pseudobranchial artery small and rounded, placed dorsally in the basal angle.

Otic region with endolymphatic fossa hexagonal and profound, carrying two anterior endolymphatic foramina and two posterior perilymphatic foramina, with similar sizes; otic capsule pentagonal and wide on each side; anterior semicircular canal inconspicuous; posterior and lateral semicircular canals well prominent on each otic capsule; the latter placed laterally in the otic capsule above the hyomandibular facet; otic crest very small and prominent, placed posteriorly between endolymphatic fossa and foramen magnum; sphenopterotic ridge subtle; opisthotic process somewhat pointed; hyomandibular facet shallow; lateral auditory groove markedly deep in the anterior edge of the lateral otic wall; inconspicuous lateral commissure.

Occipital region with occipital condyles small and triangular, ventrally; foramen magnum broad (its width 7.9% CL in *C. barbifer*, 7.6%–15.5% CL in *C. asper* and 4.1% CL in *C. australis*); foramen for vagus nerve (X) also large aside the foramen magnum; glossopharyngeal base broad, thick, and subtriangular, carrying glossopharyngeal foramen (IX) with two apertures; one aperture rounded and large placed posterior-dorsally, and a second aperture small and oval, located lateral-ventrally. Basal plate flattened and elongate (its length 51.5% CL in *C. barbifer*, 44.0%–50.3% CL in *C. asper* and 41.6% CL in *C. australis*), narrower anteriorly (its width 25.0% CL in *C. barbifer*, 21.8%–30.5% CL in *C. asper* and 41.6% CL in *C. australis*) than posteriorly in the glossopharyngeal base; basitrabecular processes been-shaped, prominent and very narrow, directed lateral-posteriorly; a single cartilaginous process, short and rounded, placed laterally in each side of the subotic shelf, width across cartilaginous processes 36.0% CL in *C. barbifer*, 35.4%–42.8% CL in *C. asper* and 33.5% CL in *C. australis*; a single foramen for carotid artery, rounded, placed mesial-anteriorly between basitrabecular processes and first cartilaginous processes; foramen for orbital artery, oval and small, located laterally near the base of each cartilaginous process.

*Cirrhigaleus asper* and *C. australis* have neurocranium of morphological condition similar to that of *C. barbifer*. *Cirrhigaleus asper*, however, differs from these species by having foramen for carotid artery with two apertures (*vs.* a single aperture for *C. barbifer* and *C. australis*), anterior semicircular canal prominent (*vs.* inconspicuous for *C. barbifer* and *C. australis*), and hyomandibular facet profound (*vs.* very shallow for *C. barbifer* and *C. australis*). Foramina of the branch superficial ophthalmic of the trigeminal nerve (V–VII) varies from 8–11 in *C. asper* while in *C. australis* varies from 7–9 foramina. *C. asper* exhibits intraspecific variation of the supraethmoidal processes, a paired short and cylindrical process located at the dorsoposterior edge of the prefrontal fontanelle. These processes may be absent (Indian Ocean population, including the holotype of *C. asper*) or present (Western Atlantic population) within the species. The supraethmoidal processes are also absent in both *C. barbifer* and *C. australis*. *Cirrhigaleus barbifer* and *C. australis* also have epiphyseal pit large and rounded with single and independent aperture, and it is not fused to the dorsal base of the prefrontal fontanelle as it was illustrated in [22]. *Cirrhigaleus asper* differs from *C. barbifer* and *C. australis* by: shorter precerebral fenestra length (39.9% CL in *C. barbifer* *vs.* 38.7% CL in *C. australis* *vs.* 29.0%–34.1% CL in *C. asper*); more elongate distance between orbital processes (34.0% CL in *C. barbifer* *vs.* 32.2% CL in *C. australis* *vs.* 36.0%–43.8% CL in *C. asper*); length of basal plate (51.5%

CL in *C. barbifer vs.* 41.6% CL in *C. australis vs.* 44.0%–50.3% CL in *C. asper*). *Cirrhigaleus asper* also has broader subethmoidal chamber than in *C. barbifer* (subethmoideal width 17.0%–18.4% CL in *C. asper vs.* 15.2% CL in *C. barbifer*), and narrower interorbital width (30.6%–34.3% CL in *C. asper vs.* 37.5% CL in *C. barbifer*).

*Pectoral apparatus*. Pectoral girdle (Fig 6) constituted by a ventral transverse element, the coracoid bar, and a pair of scapula that is continuous to the coracoid bar, ending into a pointed tip, the scapular processes. Coracoid bar together with the scapula forms the scapulocoracoid cartilage. Scapulocoracoid is U-shaped, somewhat sinuous and cylindrical in general morphology, and placed transversely in relation to the body axis with scapular processes directed dorsal-posteriorly. Coracoid bar (Fig 7A–7F) straight and horizontal with its anterior margin conspicuously pointed and triangular, forming a prominent dorsal fossa for the insertion of the hypobranchial longitudinal muscles; ventral posterior margin of coracoid bar conspicuously pointed and convex on its midline; subrectangular and narrow prominence observed lateral-ventrally on each side of the coracoid bar. Posteriorly, a pair of lateral fossa supports the origin of the *parietalis pars epaxonica* muscle on the coracoid bar; a robust and triangular process, also called caudal process or posterior process of coracoid bar, is evident on the hindmost part of each fossa that together with the base of the articular process supports the origin of *pterygii ventralis* muscle. Scapula placed more dorsal-posteriorly, very broad ventrally and tapering cylindrically to its dorsal extremities where it turns into a pointed scapular process; its anterior margin markedly convex and posterior margin concave; there is a conspicuous and expanded lateral-anterior fossa on each side of scapula for origin of the *pterygii cranialis* muscle; this anterior fossa also carries a large and rounded foramen for the pectoral fin nerve, the foramen diazonale; foramen diazonale with small posterior aperture, placed more laterally between the two articular regions; foramen for pectoral artery located laterally just above the mesocondyle; scapula with prominent posterior ridge, placed more dorsally near scapular process. Scapular process somewhat cylindrical, tapering to its dorsal tip and directed medially to the body horizontal axis; scapular process attached proximally to the scapular process by thick connective tissue. Posteriorly, the *pterygii dorsalis* fossa, placed more distally above the articular region of the pectoral girdle, supports the origin of the *pterygii dorsalis* muscle in the scapula.

Propterygium divided into two pieces, one proximal, well elongate and conical, slenderer distally, and one distal, rectangular and cylindrical, shorter than the proximal one; propterygium carries one series of segmented radials in *Cirrhigaleus barbifer*. Mesopterygium triangular, much broader distally than proximally, carrying 8 series of segmented radials. Metapterygium triangular, thin, and elongated, narrower than the mesopterygium, carrying 8 series of segmented radials; metapterygium axis with a rectangular element, carrying one series of radials and another lateral and subtriangular element with three small radials attached to it. Pectoral radials segmented into three elements as well; distal radials longer than the proximal and mesial radials in propterygium and mesopterygium, while in metapterygium is shorter than the proximal and mesial radials; first series of radials in the mesopterygium is segmented into two elements only.

*Cirrhigaleus asper* and *C. australis* share similar morphology of the pectoral apparatus of that of *C. barbifer*. *Cirrhigaleus australis* and *C. asper* bear same number of pectoral radials for each pectoral element as those of *C. barbifer*. Articular region between pectoral girdle and fin comprises by two distinct regions in *C. asper* and *C. australis*, the procondyle and a meso-metacondyle. Procondyle elliptical, oblique and well prominent, located lateral-dorsally in the scapula base, articulating with the propterygium; meso-metacondyle rounded and very large, placed posterior-ventrally in the scapula base with two distinct areas for articulating with the mesopterygium and metapterygium; the largest area articulates with the mesopterygium and it is located more lateral-posteriorly in the scapula; the second area is smaller and located more posterior-medially in the scapula for articulation with the metapterygium.

*Pelvic apparatus*. Pelvic girdle (Fig 8A–8F) with pubosichiadic bar transverse, large and narrow with anterior and posterior margins straight, although slightly convex medially; two small foramina for pelvic nerves evident laterally in the pubosichiadic bar, although anterior-most foramen is located right in the edge of lateral prepelvic process. Pelvic fin with anterior pelvic basal (= first enlarged radial) subtriangular, small and conspicuously broad with two series of radials associated to it (first radial is markedly broad and segmented); basipterygium elongate, cylindrical and thin, slightly broader on its proximal end than on its distal end; pelvic radials large, cylindrical and segmented into proximal and distal elements, the former much larger than the distal elements; 16–18 total pelvic radials for *C. barbifer* and 18 total pelvic radials for *C. australis*. Females of *C. barbifer*, *C. asper* and *C. australis* have one intermediate segment, small, barrel-shaped, attached proximally to the distal end of basipterygium, and a modified pelvic radial attached to the intermediate segment. In *C. barbifer* and *C. asper* this latter cartilage is rectangular, wide, bifurcated and cylindrical distally while in *C. australis* is ray-like, thin and cylindrical. These three species of *Cirrhigaleus* have two regions of pelvic articulation, one lateral and one medial. First region consists of a latero-posterior pelvic condyle in the puboischiadic bar that is prominent, rounded and elongate. It articulates laterally the puboischiadic bar to the anterior pelvic basal of the pelvic fin. Second articulating region is located at the medial posterior margin of puboischiadic bar and is comprised by two articular surfaces, one condyle and a pelvic facet. Pelvic facet is flattened and rectangular, placed ventromedially for articulating medially to the basipterygium. The second condyle is smaller than first condyle and it is located more dorsolaterally than the facet. The second condyle articulates to the dorsal portion of the basipterygium. The articular surface in the basipterygium is comprised by a small and rounded hollow for articulating with the second condyle.

Cirrhigaleus asper and C. australis differ from C. barbifer by having anterior foramen for pelvic nerve not completely enclosed in the lateral prepelvic process. This foramen is placed more posteriorly in C. australis while is more anteriorly in C. asper. Anterior pelvic basal of pelvic fin is rectangular and much larger in C. asper and C. australis than in C. barbifer. Four series of radials thin, cylindrical and segmented (except first one) are shown in C. asper while C. australis and C. barbifer exhibit two series of radials broad in which first series is wider than second one. Cirrhigaleus asper differs from C. barbifer and C. australis by having puboischiadic bar with anterior margin conspicuously convex medially, and posterior margin concave with paired convexity medially.

*Cartilages of the claspers*. Claspers (Fig 10) with two intermediate segments, barrel-shaped and thick connecting basipterygium to axial cartilage. First intermediate segment of *C. barbifer* is smaller than second one. *Cirrhigaleus australis* and *C. asper* apparently also have two intermediate segments that articulate basipterygium of the pelvic fin in females. Beta cartilage in *C. barbifer* is conspicuously stout and placed laterally over distal edge of basipterygium, first and second intermediate segments and proximal edge of axial cartilage transcending its mesial process. Axial cartilage almost straight and very elongate (its length equal to length of basipterygium of pelvic fin) with mesial process. Dorsal marginal cartilage strongly thick, although short (its length less than one-half length of axial cartilage), placed laterally over axial cartilage. Dorsal terminal cartilage thick, somewhat blade-like and elongate (its length one-half length of axial cartilage), located distally and connect proximally to dorsal marginal cartilage and axial cartilage. Dorsal terminal 2 cartilage leaf-like and conspicuously slender, although wider proximally, and large (its length one-third greater than length of dorsal terminal cartilage, although not reaching tip of dorsal terminal cartilage), connected proximal-medially to dorsal marginal cartilage and medial-distally to dorsal terminal 2 cartilage. Ventral marginal cartilage subtriangular and conspicuously broad distally, very elongate (it transcends midline of axial cartilage), attached to distal edge of axial cartilage; ventral marginal cartilage has folded plate markedly

profound in which accessory terminal 3 cartilage is inserted. Accessory terminal 3 cartilage blade-like and heavy, although somewhat slender and cylindrical distally. Ventral terminal cartilage spatula-like, convex at its tip and laterally, markedly large (its length twice length of ventral marginal cartilage), attached proximally to ventral marginal cartilage. Accessory ventral marginal 2 cartilage present, located over proximal edge of ventral terminal cartilage, rectangular in shape and elongate (its length one-half length of ventral terminal cartilage).

## Taxonomic considerations within Cirrhigaleus

Three species of barbel-bearing dogfish sharks are recognized in this study with occurrences in limited deep-sea regions of all oceans: *C. barbifer*, *C. australis* and *C. asper*. *Cirrhigaleus barbifer* occurs from North to South Pacific Ocean between Japan, Taiwan, New Caledonia, Western Australia and Indonesia as noticed in [1, 3–5, 7, 86]. This species is also reported in Vanuatu [5] and Taiwan [87] but examination of specimen vouchers and/or genetic analysis were not assessed in the present study to corroborate these assumptions. *Cirrhigaleus australis* is endemic to Southern Australia and New Zealand waters in the Pacific Ocean as suggested in [3, 82] but in contrast to [12] for the Indian Ocean. *Cirrhigaleus asper* occurs in the Western Indian and North to South Western Atlantic Oceans as noticed in [2, 8, 12, 83, 99, 100, 101, 114]. Its occurrence in the Hawaiian Islands and in the Central Pacific Ocean [88, 103, 107] is still doubtful.

*Cirrhigaleus* species are easily differentiated by characteristics of external morphology as provided in their diagnosis and key to species. Dermal denticles are tricuspid and conspicuously broad with cusplets on each side in *C. barbifer* and *C. australis*, which it is in disagreement with [3] who only noticed this character for the Australian species. *Cirrhigaleus barbifer*, however, differs from *C. australis* in number of cusplets (two cusplets *vs*. one cusplet, respectively). *Cirrhigaleus asper* lacks cusplets. Other characters of the dermal denticles that help to separate these species are its shape (bat-like in *C. barbifer* and *C. australis vs*. heart-shaped in *C. asper*), width of median ridge (narrow in *C. barbifer* and *C. australis vs*. broad in *C. asper*), and lateral cusp (conspicuous in *C. barbifer* and *C. australis vs*. inconspicuous in *C. asper*). Dental formula of *C. barbifer* is congruent with *C. asper* and *C. australis*, although the some specimens of this species may present one intermediate tooth on upper and lower jaws (*vs*. absence of intermediate tooth). Dentition differentiates *C. barbifer* from congeners regarding width of apron that is much broader in this species. Body colouration is also helpful for taxonomic separation, which is in agreement with [3, 7] for *C. barbifer* and *C. australis*. These authors reported differences related to second dorsal-fin spine length, length of preventral caudal margin, eye length, and length of upper labial furrow between *C. australis* and *C. barbifer*, which are not supported in our study [3]. Further separated these species based on length of first dorsal-fin spine, second dorsal fin base length, height of second dorsal fin, prenarial length, interdorsal space, and length of pectoral-fin posterior margin, although these differences are not observed here. Vertebral counts overlap between *C. barbifer* and *C. australis* and it is consistent with observations of [3, 4, 7].

Morphometric characters are useful for separating species of *Cirrhigaleus* but it is more apparent when only adult specimens are compared due to strong intraspecific and ontogenetic variations. Growth changes are not consistent to all species of *Cirrhigaleus*, revealing that their taxonomy and morphology are much more complex. For instance, length of anterior margin of nostrils and length of claspers vary with growth in *C. australis* and *C. asper* while these are constant in *C. barbifer*. Pre-second dorsal length and prespiracular length vary from juveniles to adults in *C. barbifer* and *C. asper* while these characters are continuous in *C. australis*. *Cirrhigaleus barbifer* vary with growth regarding pre-vent length, prepelvic length, pectoral-pelvic

space, length of first and second dorsal-fin inner margins, height of first and second dorsal fins, length of second dorsal-fin posterior margin, length of second dorsal-fin spine, length of pectoral-fin anterior margin, and length of pelvic fin. These variations are not observed for *C. australis* and *C. asper*. In contrast to *C. barbifer* and *C. asper*, *C. australis* exhibits ontogenetic variations in prenarial length. Precaudal length, distance between nostril and upper labial furrow, eye length, base length of first dorsal-fin, and dorsal caudal margin length vary in *C. asper* while these characters are constant in *C. barbifer* and *C. australis*.

[3,4] previously reported morphometric growth changes in *C. barbifer* and *C. australis*, respectively. The former authors took into account measurements provided in [84] for the holotype of *P. barbulifer* to represent a juvenile female with 555 mm TL. Examination of this specimen reveals that its total length corresponds to 730 mm TL (and not 555 mm TL) and that this information has been erroneously reproduced elsewhere since [84] which possibly interfered in the results of [4] regarding growth changes. Despite of it, we support that height of first and second dorsal fins, and length of pectoral-fin anterior margin vary ontogenetically in *C. barbifer* as in [4] but these characters increase with growth which is in contrast to their study. Variations in preoral length, eye length, and length of dorsal caudal margin for *C. barbifer* are not supported in the current study, which is incongruent with [4]. [3] noticed variations in pre-branchial length, prepelvic length, space between pelvic and caudal fins, distance from pectoral to pelvic fins, eye length, interorbital space, and length of second dorsal spine between juveniles and adults of *C. australis*, which is in disagreement with the current analysis. Ontogenetic variations are more expressive in *C. barbifer* than in *C. asper* and *C. australis*. Morphological analysis further reveals intraspecific regional variations for all *Cirrhigaleus* species as described under their remarks, which contribute to the taxonomical complexity of the group that was previously speculative in [3, 12, 88]. Existence of possible separate morphotypes of *C. barbifer* in Indonesia/New Caledonia/Vanuatu/Taiwan, and *C. asper* in the USA and Brazil should be investigated using an integrative taxonomic approach by incorporating molecular markers and morphological characters of newly collected material.

## Identification key to species of Cirrhigaleus

1. –Anterior margin of nostrils with nasal barbel conspicuously elongate (4.4%–6.4% TL), moustache-like and thin, often reaching anterior margin of mouth; upper labial furrow short, its length 0.9%–1.5% TL; second dorsal fin very tall, its height 8.4%–10.5% TL; pelvic fins large, its length 11.9%–14.8% TL; dermal denticles bat-like with lateral cusps conspicuous at posterior margin of the crown and narrow median ridge . . . . . . . . . . . . . . . . . . . . . . ... . . . . . . . . ...2

 –Anterior margin of nostrils with nasal barbel short (1.0%–1.5% TL), non moustache-like and thick, slightly transcending posterior margin of nostrils; upper labial furrow large, its length 1.4%%–2.1% TL; second dorsal fin very low, its height 3.9%–9.3% TL; pelvic fins small, its length 10.8%–12.7% TL; dermal denticles heart-shaped with lateral cusps inconspicuous to weak at posterior margin of the crown and median ridge broad. . . . . . . . . . . . . . . . . . . . .*C. asper*

 2. –Body dark brown; anterior margin of nostrils 5.4%–6.4% TL; dermal denticles often with two cusplets posteriorly on each side of the crown . . . . . . . . . . . . . . .. . . . . . . .*C. barbifer*

 –Body light grey; anterior margin of nostrils 4.5%–6.2% TL; dermal denticles with one cusplet posteriorly on each side of the crown . . . . . . . . . . . . . . . . . . . ... . . . . . . .... . ..*C. australis*

## Supporting information

**S1 Table. Matrix summarizing quantitative characters used in the phylogenetic study.** 1, monospondylous vertebral counts; 2, diplospondylous vertebral counts; 3, upper tooth row counts; 4, lower tooth row counts. Absolute values and normalized ones (in parentheses) are

given.
(DOCX)

**S2 Table. Matrix of discrete morphological characters utilized in the phylogenetic analysis.**
(DOCX)

**S3 Table. Tooth counts for *Cirrhigaleus barbifer*, *C. asper* and *C. australis*.** Range values are provided by regions for *C. barbifer* and *C. asper* for comparisons. Abbreviations: AUS: Australia; HT: holotype; IND: Indonesia; n: number of specimens; NC: New Caledonia; NT: neotype; NZ: New Zealand; PT: paratypes; WAO: Western Atlantic Ocean; WIO: Western Indian Ocean.
(XLSX)

**S4 Table. Vertebral counts for species of *Cirrhigaleus*.** *: neotype of *C. barbifer*; **: holotype.
(XLSX)

**S1 File. List of non-ambiguous synapomorphies of clades and terminal taxa based on the most-parsimonious cladogram resulting from implied weighting (*k* = 3).**
(DOCX)

**S2 File. List of morphological character transformation based on the most-parsimonious cladogram resulting from combined datasets and under extended implied weighting (SEP, *k* = 3).**
(DOCX)

**S3 File. Complete data matrix including quantitative and qualitative characters analyzed in.**
(XLSX)

**S4 File. Complete data matrix including quantitative and qualitative characters analyzed in.**
(TXT)

**S5 File. A list of examined preserved material is provided below and include species of the genera *Squalus*, *Dalatias* and *Isistius* that were used for comparative purposes.** Preserved examined material of *Cirrhigaleus* species are given in full under the section 'Taxonomic account'. Skeletal material for character analysis and polarity is given for both ingroups and outgroups of the current analysis.
(DOCX)

## Acknowledgments

We thank M. McGrouther and S. Reader (AMS), P. Last and A. Graham (CSIRO), C. Roberts and A. Stewart (NMNZ), O. Crimmen, J. Maclaine and R. Britz (NHM), T. Kawai and F. Tashiro (HUMZ), O. Gon, R. Bills, N. Mazungula and A. Gura (SAIAB), D. Clark, M. Bougaardt and M Lisher (iSAM), T. Iwamoto, D. Catania, J Fong and L. Rocha (CAS), P. Pruvost (MNHN), A. Palandacic (NMW), M. Britto (MNRJ), U. Gomes (UERJ), M. Gonzalez (NUPEC), J. Williams (USNM), S. Kullander, B. Delling, E. Ahlander and M. Noren (NRM) for their curatorial and technical assistance. Mônica T. P. Ragazzo (USP) and M.R. Carvalho (formerly USP) are thanked for their suggestions and academic support on this manuscript. Special thanks to K. Sakamoto (ZUMT) and D. Catania (CAS) for assisting with type material, A. Ray (AMS), Sandra Raredon (USNM), J. Fong (CAS), J. Pogonoski (CSIRO), and C. Struthers (NMNZ) for providing radiographs, W. White (CSIRO) and D. Catania (CAS) for

sending material on loan for dissection. Thanks to A. Paterson (SAIAB) for institutional support of this project. Special thanks also to F. Tashiro (HUMZ) and C. Struthers (NMNZ) for sending photos and data of specimen vouchers, and P. Roque and F. Hazin (UFRPE) for providing photos and collecting data of fresh specimens. We acknowledge that opinions, findings and conclusions or recommendations expressed in this publication, partly supported by NRF funds, are those of the authors, and that the NRF accepts no liability whatsoever in this regard.

## Author Contributions

**Conceptualization:** Sarah Viana.

**Data curation:** Sarah Viana.

**Formal analysis:** Sarah Viana, Karla D. A. Soares.

**Funding acquisition:** Sarah Viana.

**Investigation:** Sarah Viana, Karla D. A. Soares.

**Methodology:** Sarah Viana, Karla D. A. Soares.

**Project administration:** Sarah Viana.

**Validation:** Sarah Viana, Karla D. A. Soares.

**Visualization:** Sarah Viana, Karla D. A. Soares.

**Writing – original draft:** Sarah Viana.

**Writing – review & editing:** Karla D. A. Soares.

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
