## [Decision Letter · Decision Letter 0]

17 Nov 2022

PONE-D-22-29095Untangling the systematic mystery behind the roughskin spurdog Cirrhigaleus asper (Merrett, 1973) (Chondrichthyes: Squaliformes), with phylogeny of Squalidae and a key to speciesPLOS ONE

Dear Dr. Viana,

Thank you for submitting your manuscript to PLOS ONE. After careful consideration, we feel that it has merit but does not fully meet PLOS ONE’s publication criteria as it currently stands. Therefore, we invite you to submit a revised version of the manuscript that addresses the points raised during the review process.

We look forward to receiving your revised manuscript.

Kind regards,

Jürgen Kriwet

Academic Editor

PLOS ONE

“This study was funded by the following: Fundação de Amparo à Pesquisa do Estado de São Paulo (FAPESP 2011/18861–7; 2013/11621–6; 2014/26503–1), STFLV, http://www.fapesp.br/en/; Conselho Nacional de Desenvolvimento Científico e Tecnológico (CNPq 158773/2011–0), STFLV, http://cnpq.br/;  Geddes Collection Visiting Fellowship (2013), Australian Museum, STFLV. All funders had important roles in the study design, data collection and analysis, and decision to publish.”

3. We note that you have referenced (ie. Bewick et al. [5]) which has currently not yet been accepted for publication. Please remove this from your References and amend this to state in the body of your manuscript: (ie “Bewick et al. [Unpublished]”) as detailed online in our guide for authors

5. We note that Figure 25 in your submission contain copyrighted images. All PLOS content is published under the Creative Commons Attribution License (CC BY 4.0), which means that the manuscript, images, and Supporting Information files will be freely available online, and any third party is permitted to access, download, copy, distribute, and use these materials in any way, even commercially, with proper attribution. For more information, see our copyright guidelines: http://journals.plos.org/plosone/s/licenses-and-copyright.

a. You may seek permission from the original copyright holder of Figure 25 to publish the content specifically under the CC BY 4.0 license.

Additional Editor Comments:

Dear authors:

Thank you very much for submitting this excellent study to Plos One. Two reviewers commented on your manuscript and both agree with your results and only have some minor comments and/or suggestion, which I would like to aks to consider carefully. I agree with the second reviewer that it is unforunate that you did not provide the complete data matrix. I would suggest to include the full matrix in the manuscript and also a nexus file in the supplementary information, so that it is possible to recapitulate your analysis.

Kind regards,

Jürgen Kriwet

Reviewers' comments:

Reviewer's Responses to Questions

**Comments to the Author**

1. Is the manuscript technically sound, and do the data support the conclusions?

Reviewer #1: Yes

Reviewer #2: Yes

2. Has the statistical analysis been performed appropriately and rigorously? 

Reviewer #1: Yes

Reviewer #2: Yes

3. Have the authors made all data underlying the findings in their manuscript fully available?

Reviewer #1: Yes

Reviewer #2: No

4. Is the manuscript presented in an intelligible fashion and written in standard English?

Reviewer #1: Yes

Reviewer #2: Yes

5. Review Comments to the Author

Reviewer #1: Dear editors, authors,

It was my pleasure to read and review the paper by Viana & Soares entitled ‘Untangling the systematic mystery behind the roughskin spurdog Cirrhigaleus asper (Merrett, 1973) (Chondrichthyes: Squaliformes), with phylogeny of Squalidae and a key to species’.

This topic is of taxonomic importance, and the paper itself is well organised, written and apparently prepared with much care as relatively little typo’s and mistakes in cross-references to Figs etc. occur. It is also noticeable that authors are experts in this field.

Nevertheless, please allow me to suggest the following improvements (on headlines):

- Titel: avoid the word ‘mystery’ (perhaps better: dilemma), and be more specific about

the key that is for the genus, whereas the title only mentions a single species and the

family (now confusing).

- Introduction: visualise the main hypotheses by simplified trees in an additional Fig.

- Morp. phyl. reconstruction: there is a problem in the lettering of the Clades, either

in the text of the MS, or in the Fig 13 itself (I guess in the latter). This needs special

attention and if needed correction throughout the entire MS.

- Key: in terms of presentation, I think it would be better to place this key at the final end

of the paper (i.e. after ‘Taxonoic considerations within Cirrhigaleus’), somewhat as a

conclusion.

- Figures: perhaps better to put the lettering of all panels in CAPITALS (A, B, C…) for

better contrast with abbreviations in Figs, and better contrast with Figs of external

references. Also check for scale bars.

- Smaller comments: for several smaller comments (e.g., rows vs. files, holotype info C.

barbifer, etc. etc.), please check the annotated MS carefully (see attached).

Wishing you all the best with finalising the MS,

Best regards,

Frederik

Reviewer #2: Overall a really good work with tons of information that can be put on larger context within Neoselachians. My only concern with the work is that the authors did not provide their full matrix, only the discrete features were provided. Consequently there is no way evaluate their results. Furthermore, I would suggest to the authors to provide their matrices in nexus file to facilitate the revision of their coding, as their matrix seems to present some non-informative characters (these features do not provide grouping information, but can inflate the CI and RI of the tree).

I have added some comments in a PDF of the paper (see comments with acrobat or other PDF readers, the browser one wont display them).

6. PLOS authors have the option to publish the peer review history of their article (what does this mean?). If published, this will include your full peer review and any attached files.

Reviewer #1: No

Reviewer #2: **Yes: **Eduardo Villalobos Segura

---

## [Author Response · Author response to Decision Letter 0]

11 Jan 2023

Dear Editor and Reviewes,

Thank you for all the time spent during the review of this manuscript. We have addressed most of the comments/sugestions. Please see our answers under the file "Response to Reviewers". 

Kind regards,

Sarah

---

## [Decision Letter · Decision Letter 1]

27 Jan 2023

PONE-D-22-29095R1Untangling the systematic dilemma behind the roughskin spurdog Cirrhigaleus asper (Merrett, 1973) (Chondrichthyes: Squaliformes), with phylogeny of Squalidae and a key to Cirrhigaleus speciesPLOS ONE

Dear Dr. Viana,

Thank you for submitting your manuscript to PLOS ONE. After careful consideration, we feel that it has merit but does not fully meet PLOS ONE’s publication criteria as it currently stands. Therefore, we invite you to submit a revised version of the manuscript that addresses the points raised during the review process.

We look forward to receiving your revised manuscript.

Kind regards,

Jürgen Kriwet

Academic Editor

PLOS ONE

Journal Requirements:

Additional Editor Comments:

Dear authors:

Thank you very much for considerng the comments by the reviewers and also for your patience. Reviewer 2 has raised an important aspect concerning character 16. Please check this comment and address it shortly in your manuscript. Please also check Figures 14 and 15 as there quality seems to be very low and improve the quality if necessary.

Please re-submit the manuscript once you have done the required improvements. I will check it (without another reviewer round) and make asap a final decision.

Kind regards,

Jürgen

Reviewers' comments:

Reviewer's Responses to Questions

**Comments to the Author**

1. If the authors have adequately addressed your comments raised in a previous round of review and you feel that this manuscript is now acceptable for publication, you may indicate that here to bypass the “Comments to the Author” section, enter your conflict of interest statement in the “Confidential to Editor” section, and submit your "Accept" recommendation.

Reviewer #1: All comments have been addressed

Reviewer #2: All comments have been addressed

2. Is the manuscript technically sound, and do the data support the conclusions?

Reviewer #1: (No Response)

Reviewer #2: Yes

3. Has the statistical analysis been performed appropriately and rigorously? 

Reviewer #1: (No Response)

Reviewer #2: N/A

4. Have the authors made all data underlying the findings in their manuscript fully available?

Reviewer #1: (No Response)

Reviewer #2: (No Response)

5. Is the manuscript presented in an intelligible fashion and written in standard English?

Reviewer #1: (No Response)

Reviewer #2: (No Response)

6. Review Comments to the Author

Reviewer #1: (No Response)

Reviewer #2: Char. 16. While I agree, the objective of the present work is not to address the homology of structures between batoids and sharks or in group shark relations. By naming a character, for example, Antorbital cartilages in the neurocranium, some sense of homology is implied in the name, the limitations on the generalisations on the observations need to be adequately clarified in their character description, so the reader interprets no untested homology. While I do not expect a full anatomical revision, I would hope that the authors include a short statement mentioning that while they are using the term Antorbital cartilage, no homology between this observed feature and the one present on batoids is implied by them as this need to be further tested. The same goes for characters 9 and 10. Furthermore, if they are following Shirai (1992 and 1996) in the codification of character 16 for this paper need to be cited to add context to their character description, same goes for Maisey's works (1982, 1983).

7. PLOS authors have the option to publish the peer review history of their article (what does this mean?). If published, this will include your full peer review and any attached files.

Reviewer #1: **Yes: **Frederik H. Mollen (Elasmobranch Research, Belgium)

Reviewer #2: **Yes: **Eduardo Villalobos Segura

---

## [Author Response · Author response to Decision Letter 1]

9 Feb 2023

Dear Dr Kriwet,

Thank you for considering our manuscript for publication at PLOSOne and for your time spent and inputs provided during the review process. 

We have addressed all the comments provided by you and the two reviewers. Answers to each comment and details about the changes made to the manuscript are provided below in blue. 

Journal Requirements:

We have reviewed accordingly and we have not made any changes. We did not cite any paper that has been retracted. 

Additional Editor Comments:

Dear authors:

Thank you very much for considerng the comments by the reviewers and also for your patience. Reviewer 2 has raised an important aspect concerning character 16. Please check this comment and address it shortly in your manuscript. Please also check Figures 14 and 15 as there quality seems to be very low and improve the quality if necessary.

Please re-submit the manuscript once you have done the required improvements. I will check it (without another reviewer round) and make asap a final decision.

Kind regards,

Jürgen

We have addressed the issues raised with character 16 and the quality of the Figures 14 and 15.

Reviewers' comments:

Reviewer's Responses to Questions

Comments to the Author

1. If the authors have adequately addressed your comments raised in a previous round of review and you feel that this manuscript is now acceptable for publication, you may indicate that here to bypass the “Comments to the Author” section, enter your conflict of interest statement in the “Confidential to Editor” section, and submit your "Accept" recommendation.

Reviewer #1: All comments have been addressed

Reviewer #2: All comments have been addressed

2. Is the manuscript technically sound, and do the data support the conclusions?

Reviewer #1: (No Response)

Reviewer #2: Yes

3. Has the statistical analysis been performed appropriately and rigorously? 

Reviewer #1: (No Response)

Reviewer #2: N/A

4. Have the authors made all data underlying the findings in their manuscript fully available?

Reviewer #1: (No Response)

Reviewer #2: (No Response)

5. Is the manuscript presented in an intelligible fashion and written in standard English?

Reviewer #1: (No Response)

Reviewer #2: (No Response)

6. Review Comments to the Author

Reviewer #1: (No Response)

Reviewer #2: Char. 16. While I agree, the objective of the present work is not to address the homology of structures between batoids and sharks or in group shark relations. By naming a character, for example, Antorbital cartilages in the neurocranium, some sense of homology is implied in the name, the limitations on the generalisations on the observations need to be adequately clarified in their character description, so the reader interprets no untested homology. While I do not expect a full anatomical revision, I would hope that the authors include a short statement mentioning that while they are using the term Antorbital cartilage, no homology between this observed feature and the one present on batoids is implied by them as this need to be further tested. The same goes for characters 9 and 10. Furthermore, if they are following Shirai (1992 and 1996) in the codification of character 16 for this paper need to be cited to add context to their character description, same goes for Maisey's works (1982, 1983).

We have changed the text accordingly to clarify description of Character 16. Please read as: “16) Antorbital cartilage…. No homology between this character and the one similarly described for non-squalean sharks and batoids in Shirai (1992) is implied in the present study as this requires further investigation. Character codification follows Shirai (1992, 1996).”

However, again we did not do the same for character 9 and 10 (rostral keel) as the ‘caudal internasal keel’ (also cited as ‘median internasal keel’ in other studies such as Maisey 1982, 1983) seems to be homologous to the subethmoidal ridge in Squaliformes as per Shirai’s (1992) and the present study. The character rostral keel does not correspond to this structure. As we clarified in the first round of the review process, the rostral keel is an anterior-ventral extension of the subethmoidal ridge.. 

7. PLOS authors have the option to publish the peer review history of their article (what does this mean?). If published, this will include your full peer review and any attached files.

Do you want your identity to be public for this peer review? For information about this choice, including consent withdrawal, please see our Privacy Policy.

Reviewer #1: Yes: Frederik H. Mollen (Elasmobranch Research, Belgium)

Reviewer #2: Yes: Eduardo Villalobos Segura

---

## [Decision Letter · Decision Letter 2]

21 Feb 2023

Untangling the systematic dilemma behind the roughskin spurdog Cirrhigaleus asper (Merrett, 1973) (Chondrichthyes: Squaliformes), with phylogeny of Squalidae and a key to Cirrhigaleus species

PONE-D-22-29095R2

Dear Dr. Viana,

We’re pleased to inform you that your manuscript has been judged scientifically suitable for publication and will be formally accepted for publication once it meets all outstanding technical requirements.

Kind regards,

Jürgen Kriwet

Academic Editor

PLOS ONE

Additional Editor Comments (optional):

Reviewers' comments:

Reviewer's Responses to Questions

**Comments to the Author**

1. If the authors have adequately addressed your comments raised in a previous round of review and you feel that this manuscript is now acceptable for publication, you may indicate that here to bypass the “Comments to the Author” section, enter your conflict of interest statement in the “Confidential to Editor” section, and submit your "Accept" recommendation.

Reviewer #2: All comments have been addressed

2. Is the manuscript technically sound, and do the data support the conclusions?

Reviewer #2: Yes

3. Has the statistical analysis been performed appropriately and rigorously? 

Reviewer #2: Yes

4. Have the authors made all data underlying the findings in their manuscript fully available?

Reviewer #2: Yes

5. Is the manuscript presented in an intelligible fashion and written in standard English?

Reviewer #2: Yes

6. Review Comments to the Author

Reviewer #2: I would like to thank the authors for addressing the issues raised on previous revision rounds and congratulate them on their work. I consider that the manuscript is publishable.

7. PLOS authors have the option to publish the peer review history of their article (what does this mean?). If published, this will include your full peer review and any attached files.

Reviewer #2: **Yes: **Eduardo Villalobos Segura

---

## [Editor Report · Acceptance letter]

24 Feb 2023

PONE-D-22-29095R2 

Untangling the systematic dilemma behind the roughskin spurdog *Cirrhigaleus asper* (Merrett, 1973) (Chondrichthyes: Squaliformes), with phylogeny of Squalidae and a key to *Cirrhigaleus* species 

Dear Dr. Viana:

I'm pleased to inform you that your manuscript has been deemed suitable for publication in PLOS ONE. Congratulations! Your manuscript is now with our production department. 

Kind regards, 

on behalf of

Dr. Jürgen Kriwet 

Academic Editor

PLOS ONE